# Ice nucleation abilities of soot particles determined with the Horizontal Ice Nucleation Chamber

Fabian Mahrt[1], Claudia Marcolli[1], Robert O. David[1], Philippe Grönquist[3,4], Eszter J. Barthazy Meier[2], Ulrike Lohmann[1], and Zamin A. Kanji[1]

[1]Department of Environmental System Sciences, Institute for Atmospheric and Climate Science, ETH Zurich, 8092 Zurich, Switzerland
[2]Scientific Center for Optical and Electron Microscopy, ETH Zurich, 8093 Zurich, Switzerland
[3]Department of Civil, Environmental and Geomatic Engineering, Institute for Building Materials, ETH Zurich, 8093 Zurich, Switzerland
[4]Department of Functional Materials, Applied Wood Materials, Empa, 8600 Dubendorf, Switzerland

**Correspondence:** F. Mahrt (fabian.mahrt@env.ethz.ch) and Z. A. Kanji (zamin.kanji@env.ethz.ch)

**Abstract.** Ice nucleation by different types of soot particles is systematically investigated over the temperature range from 218 to 253 K relevant for both mixed-phase (MPCs) and cirrus clouds. Soot types were selected to represent a range of physicochemical properties associated with combustion particles. Their ice nucleation ability was determined as a function of particle size using relative humidity ($RH$) scans in the Horizontal Ice Nucleation Chamber (HINC). We complement our ice nucleation results by a suite of particle characterization measurements, including determination of particle surface area, fractal dimension, temperature dependent mass loss, water vapor sorption and inferred porosity measurements. Independent of particle size, all soot types reveal absence of ice nucleation below and at water saturation in the MPC regime ($T > 235$ K). In the cirrus regime ($T \leq 235$ K), soot types show different freezing behaviour depending on particle size and soot type, but the freezing is closely linked to the soot particle properties. Specifically, our results suggest that if soot aggregates contain mesopores (pore diameters of $2-50\,\mathrm{nm}$) and have sufficiently low water-soot contact angles, they show ice nucleation activity and can contribute to ice formation in the cirrus regime at $RH$ well below homogeneous freezing of solution droplets. We attribute the observed ice nucleation to a pore condensation and freezing (PCF) mechanism. Nevertheless, soot particles without cavities of the right size and/or too high contact angles nucleate ice only at or well above the $RH$ required for homogeneous freezing conditions of solution droplets. Thus, our results imply that soot particles able to nucleate ice via PCF, could impact the microphysical properties of ice clouds.

## 1 Introduction

Soot, mainly composed of highly agglomerated carbon spherules, is a by-product of incomplete combustion of biomass and fossil fuels and represents a major anthropogenic pollutant. Globally, emissions of soot are estimated to reach 7500 $\mathrm{Ggy^{-1}}$

(uncertainty range: $2000 - 29000 \text{ Ggy}^{-1}$, Bond et al., 2013), with a direct source in the upper troposphere from aviation emissions, and thus are of high relevance for climate (Ramanathan and Carmichael, 2008). Soot aerosols generally denote complex internal mixtures of black carbon (BC) and associated organic matter (OM, Petzold et al., 2013), and here we use the term *soot* to encompass internal mixtures of both. Soot aggregates usually have diameters on the nanoscale, ranging from individual primary carbonaceous spherules to large, fractal-like aggregates (Adachi et al., 2007). The primary particle diameter itself can vary from around $10 \text{ nm}$ to several tens of nanometers, depending on the combustion source. The diverse physicochemical properties of soot aerosols make an analytical assessment of their environmental effects challenging. Soot particles can act as ice nucleating particles (INPs) in cirrus clouds (DeMott, 1990; DeMott et al., 1999) and MPCs as well as in aircraft contrail formation (Seinfeld, 1998; Boucher, 1999; Popovicheva et al., 2004; Schumann, 2005; Kärcher et al., 2007; Heymsfield et al., 2010; Schumann and Heymsfield, 2017; Kärcher, 2018) and thus affect cloud properties such as emissivity, lifetime, albedo and cloud coverage. The pathways of ice formation have recently been reviewed by Vali et al. (2015) and Kanji et al. (2017). For instance, when acting as INPs in supercooled liquid and/or MPCs, aerosol particles can trigger glaciation of the clouds resulting in efficient precipitation formation causing a reduction in cloud lifetime (Lohmann, 2002). However, this so-called glaciation effect is rather uncertain due to unknowns in primary ice formation with aerosol species such as soot (Lohmann, 2002). Additionally, particle properties such as coatings of BC with organic substances can reduce or inhibit their ice nucleation abilities (Hoose et al., 2008) thus adding further uncertainties to the contribution of soot to ice nucleation. Understanding the ice nucleation mechanism of soot aerosols is crucial in order to describe the fate of these particles in the atmosphere and resolve the uncertainties associated with aerosol-cloud interactions (Lohmann, 2015; Fan et al., 2016; Lohmann and Feichter, 2005). This will ultimately enhance our understanding of the anthropogenic influence on clouds.

Numerous laboratory studies have investigated the ice nucleation ability of soot (e.g. Garten and Head, 1964; DeMott, 1990; Diehl and Mitra, 1998; Gorbunov et al., 1998; DeMott et al., 1999; Gorbunov et al., 2001; Suzanne et al., 2003; Popovicheva et al., 2004; Möhler et al., 2005a, b; Dymarska et al., 2006; Kanji and Abbatt, 2006; DeMott et al., 2009; Fornea et al., 2009; Koehler et al., 2009; Kanji et al., 2011; Crawford et al., 2011; Chou et al., 2013; Brooks et al., 2014; Kulkarni et al., 2016; Schill et al., 2016; Charnawskas et al., 2017; Demirdjian et al., 2009; Kireeva et al., 2009; Häusler et al., 2018) and a review was recently provided by Ullrich et al. (2017). However, these studies have revealed a large variability in ice nucleation characteristics of soot particles, indicating that the ice formation ability of soot remains poorly understood. For instance, DeMott et al. (1999) used commercially available lamp black soot (Degussa) to test the ice nucleation ability of soot in the cirrus regime ($T \leq 235 \text{ K}$), using a continuous flow diffusion chamber (CFDC). They found ice formation below water saturation for $T < 231 \text{ K}$. However, since ice nucleation was observed very close to water saturation, whether the ice formation occurred through deposition nucleation or water sorption and subsequent freezing could not be determined. This finding is in direct contrast to Möhler et al. (2005a), who used soot produced by a graphite spark generator (GSG) and tested its ice nucleation ability using the AIDA (Aerosol Interactions and Dynamics in the Atmosphere) chamber. They found bare (uncoated) GSG soot to nucleate ice in the deposition mode for $T < 233 \text{ K}$, with ice formation onset observed well below homogeneous freezing conditions, even though their reported activated fractions made up only $0.3 \%$ of the total aerosol population. Investigation of

ice nucleation on propane fuel particles by Möhler et al. (2005b) revealed a dependence of the ice nucleation ability on the OM content of the soot particles. They showed that ice nucleation at 207 K on low OM soot (16 % by mass) was more efficient compared to high OM content (40 % by mass) particles. While the ice nucleation onset for the low OM soot was similar to that observed by Möhler et al. (2005a) for $H_2SO_4$ coated GSG soot, the high OM soot required homogeneous freezing conditions.

The cause for this was interpreted as a suppression of deposition nucleation by the increased organic content covering the carbon spherules. A later study by Crawford et al. (2011) on the ice nucleation of propane soot particles came to the same conclusion by varying OM content. Specifically, they found that only the soot particles with the lowest OM content (5 % by mass) were able to heterogeneously nucleate ice in the deposition mode (at $T = 226$ K), while higher OM content soot (30 % and 70 % by mass) required water supersaturation. Friedman et al. (2011) performed ice nucleation experiments on size

selected soot particles produced by a propane burner at $T = 233$ K, 243 K and 253 K. They did not detect heterogeneous ice formation of bare soot particles before the formation of cloud droplets. At the same time they found that organic coatings resulted in an increased hydrophilicity, but not in a significant change in the ice nucleation ability of the particles. Koehler et al. (2009) tested the ice nucleation ability of five different soot types with different physicochemical properties in the cirrus regime. They reported that the capacity of the soot particles to take up water strongly influenced their ice nucleation

characteristics. Using the hydration property classification described in Popovicheva et al. (2008), they found hygroscopic soot, with water uptake caused by water-soluble material on the soot aggregates, to freeze only homogeneously, similar to any other hygroscopic particles. In contrast, hydrophobic and hydrophilic soots, associated with little or no water-soluble material both froze heterogeneously. The authors therefore concluded that water affinity alone is insufficient to predict the ice nucleation ability of soot and that other particle characteristics such as porosity and surface polarity complicate predictions

for the heterogeneous freezing of soot particles. A direct intercomparison of the ice nucleation results discussed above is challenging due to the complex characteristics of soot particles. Yet, the heterogeneity of the reported ice nucleation results arise partly from the irreproducibility associated with combustion particle properties (Popovicheva et al., 2008) and from the different techniques used to probe ice nucleation.

In this study we systematically investigate the ice formation ability of six different soot samples. Testing the ice nucleation of

soot particles with different physicochemical properties using the same experimental procedure, allows for a direct comparison and improves our understanding of the ice formation characteristics of soot aerosol. Ice nucleation was investigated in the MPC ($T > 235$ K) and cirrus cloud temperature regime, covering the range between $218 - 253$ K, as a function of relative humidity ($RH$) and aerosol size, using the Horizontal Ice Nucleation Chamber (HINC, Lacher et al., 2017), a continuous flow diffusion chamber. Our measurements extend the limited measurements of particle size dependent ice nucleation characteristics

of soot aerosol, particularly in the cirrus regime. In addition, we characterize the particle properties and report measurements of morphological features, bulk heat sensitive fraction, hydrophilicity and porosity of the soot samples to interpret our INP measurements. By combining these auxillary measurements with our INP experiments, we draw fundamental conclusions on the relationship between particle properties and the ice formation mechanisms of soot particles for atmospherically relevant conditions.

## 2 Experimental methods and materials

### 2.1 Soot samples

Soot samples were chosen to represent a wide range of combustion aerosol physicochemical properties as proxies of atmospheric soot which could still differ from those particles studied here. A summary of the characteristics of the soot samples
used here is shown in Table 1. Below we briefly describe their properties and origin.

**FW200**: This commercially available carbon black (Orion Engineered Carbons GmbH, OEC, Frankfurt, Main, Germany) is produced in the gas black process, through the incomplete combustion of liquid hydrocarbons. A commercial carbon black was chosen to ensure reproducibility of the sample for future ice nucleation studies. The gas black method allows for production of particles with narrow primary particle size distributions with mean primary particle diameters of $\bar{d}_{pp} = 22$ nm, as found
by transmission electron microscopy (TEM, see Table 1). The smaller the primary particle size the higher the surface area, consistent with the high specific surface area, $a_{BET} = 526$ m$^2$g$^{-1}$, determined by N$_2$ adsorption using the Brunauer-Emmett-Teller (BET) method (Brunauer et al., 1938). Due to their slightly oxidized surface, gas blacks are acidic. Based on these characteristics, we choose FW200 soot with the goal to identify a potential laboratory surrogate for atmospherically aged soot.

**LB_OEC**: A lamp black carbon was obtained from OEC. Lamp black carbons generally have a broader primary particle
size distribution and a lower specific surface area compared to gas black soots. In the case of LB_OEC, produced by the lamp black process, the manufacturer specifies an average primary particle sizes of $\bar{d}_{pp} = 95$ nm, comparable to the $\bar{d}_{pp} = 119$ nm we derived by TEM (see Appendix C). Soot aerosols with a wide primary particle size distribution usually result in aggregates that span a relatively wide size range compared to soots with a narrow range of primary particle sizes. At the same time, the fraction of particles containing many primary particles should be limited when selecting a mobility diameter close to the
reported $\bar{d}_{pp}$. Nitrogen BET surface area was measured to be $24$ m$^2$g$^{-1}$. Similar soot has been used for previous ice nucleation studies (DeMott et al., 1999), rendering a direct comparison of our results feasible.

**LB_RC**: Another amorphous lamp black soot, lamp black Rublev Colours (RC), was purchased from Natural Pigments LLC, (Willits, California, USA). It is also produced by the lamp black process and we found an average primary particle size of $\bar{d}_{pp} = 152$ nm, by evaluating TEM images. Its specific surface area was measured to be $23$ m$^2$g$^{-1}$. This sample is directly
comparable to the lamp black soot purchased from OEC, due to their very similar physical properties, thus, allowing for a direct comparison of the ice nucleation abilities of the same soot type stemming from different combustion sources.

**miniCAST soot**: Combustion aerosol was generated using a miniature combustion aerosol standard (miniCAST, Model: 4200, Jing Ltd., Zollikofen, Switzerland). The miniCAST produces soot aerosol in a co-flow diffusion flame, mixing two concentric, interleaved flows of propane (inner flow) and particle-free, VOC-filtered, synthetic oxidation air (outer flow). Changing
the fuel-air ratio varies the properties of the resulting aerosols, such as mean aggregate size, morphology and OM content over a wide range and in a reproducible fashion (Moore et al., 2014; Schnaiter et al., 2006; Mueller et al., 2015; Mamakos et al., 2013; Yon et al., 2015). It has been shown in the past that miniCAST soot can mimic the physicochemical properties of aircraft soot (Bescond et al., 2014). Here we test two soots that differ in OM content, namely a sample termed **mCAST black** produced under fuel lean conditions, with a C:O ratio of $0.21$ and $\bar{d}_{pp} = 31$ nm, as well as a more organic rich soot, termed **mCAST**

**brown**, with a flow C:O ratio of 0.23 and $\bar{d}_{pp} = 21$ nm. Details about the miniCAST operation conditions used here are given in Appendix B. The specific surface area using $N_2$ absorption was measured at 120 $m^2g^{-1}$ and 70 $m^2g^{-1}$ for mCASTblack and mCASTbrown, respectively.

**FS**: Lastly, FS, a fullerene soot, was purchased from Sigma Aldrich, Darmstadt, Germany. Fullerenes are hollow molecules
5 of carbon atoms, arranged in a highly symmetric manner (Krätschmer et al., 1990). This fullerene soot mainly consist of $C_{60}$ molecules with a minor fraction of $C_{70}$ (see Table 1). Fullerene-like soots are often used as surrogates for particles produced by emissions from diesel engines (Muller et al., 2005). The FS investigated here had a specific surface area of 265 $m^2g^{-1}$ and the derived primary particle diameter was 41 nm.

**Table 1.** Particle characteristics of soot types investigated in this study, where $\bar{d}_{pp}$ denotes the mean physical diameter of primary particles, as obtained from TEM evaluation of the number of primary particles indicated and $\sigma(d_{pp})$ the corresponding standard deviation; $a_{BET,N2}$ the BET specific surface area using $N_2$ adsorption at 77 K and $ML_{950}$ the mass loss at 950 °C measured by thermogravimetric analysis, as an upper limit of heat sensitive material associated with the soot. $D_{fm}$ denotes the fractal dimension derived from power-law fits of the form of Eq. (A4) to mass-mobility data.

| Type of soot | mCAST black | mCAST brown | FW200 | LB_OEC | LB_RC | FS |
|---|---|---|---|---|---|---|
| Source | miniCAST[a] | miniCAST[a] | Orion Engineered Carbons Colour black FW200 powder Gas black process | Orion Engineered Carbons Flammruss FR101 powder Lamp black process | Natural Pigments LLC Lamp black process | Product no.: 572497, Lot no. MKBB8240V |
| Form/Dispersion method | combustion | combustion | Powder/FBG[e] | Powder/FBG[e] | Powder/FBG[e] | Powder/FBG[e] |
| General classification | | | amorphous CAS-no.: 1333-86-4 | amorphous CAS-no.: 1333-86-4 | amorphous CAS-no.: 1333-86-4 | $C_{60}$ (78.7 %), $C_{70}$ (18.8 %), >$C_{70}$ (1.6 %), Fullerene oxides (0.9 %) |
| $\bar{d}_{pp}$ [nm] | 31 | 21 | 22; 13[b] | 119; 95[b] | 152; 95[f] | 41 |
| $\sigma(d_{pp})$ [nm] | 5.9 | 4.9 | 3.9 | 43 | 55.7 | 12.9 |
| Primary particles analyzed | 122 | 129 | 166 | 141 | 137 | 142 |
| $ML_{950}$ °C [%] | 15.62 | 18.58 | 31.43; 20[c] | 0.97 | 2.8 | 33.37 |
| $a_{BET,N2}$ [$m^2g^{-1}$][d] | 120 | 70 | 526 | 24 | 23 (20)[f] | 265 |
| $D_{fm}$ [-] | 1.86 | 2.31 | 2.35 | 2.64 | 2.83 | 2.5 |
| Ice nucleation studies | Möhler et al. (2005b) Crawford et al. (2011) Friedman et al. (2011) | | | Dymarska et al. (2006) DeMott et al. (1999) | see LB_OEC | |

[a] Series 4200, see Table B1 for detailed flow settings

[b] $\bar{d}_{pp}$ specified by manufacturer based on ASTM (American Society for Testing and Materials) International standard D3849

[c] $ML_{950}$ °C specified by manufacturer based on DIN 53552

[d] $N_2$ BET see Appendix A1

[e] aerosolized using the Fluidized Bed Aerosol Generator, Model 3400A, TSI Inc.

[f] specified by manufacturer, technique/method not specified

## 2.2 Experimental setup

The experimental setup can be divided into three main sections as shown in Fig. 1, namely aerosol generation, aerosol selection and ice nucleation, which are briefly described below.

**Aerosol generation:** Soot aerosols were generated using different techniques, as specified in Table 1. Dry-dispersion was used for samples available in powder form through a Fluidized Bed Aerosol Generator (FBG, Model 3400A, TSI Inc.), operated with a bed flow rate of $10$ $Lmin^{-1}$, a chain purge flow rate of $2$ $Lmin^{-1}$ along with a variable chain feed rate to achieve stable aerosol number concentrations. Particles from the FBG were subsequently passed to a $0.125$ $m^3$ stainless steel mixing volume, equipped with a fan to keep the particles suspended. Contrary, the miniCAST soot samples, were directly generated in the laboratory from the propane burner. Aerosol particles for ice nucleation experiments were sampled isokinetically from the miniCAST, using forward-pointing inlets centered at the outlet. The aerosol particles were then fed through a dilution

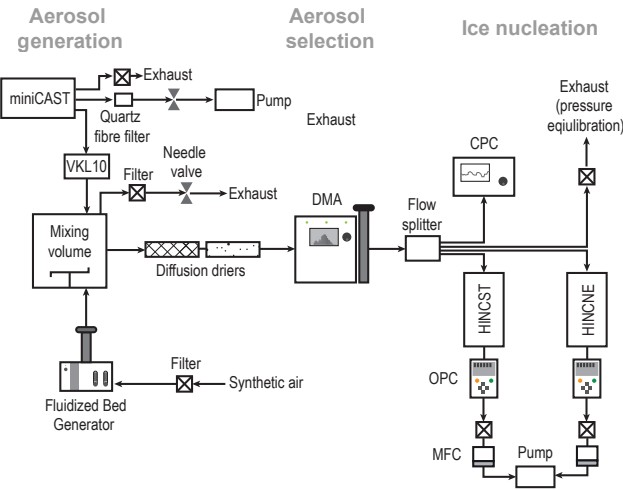

**Figure 1.** Schematic of the experimental setup comprising three different stages: aerosol generation, aerosol size selection and ice nucleation measurements. Arrows indicate the direction of the aerosol flow. Aerosol particles were either produced by the miniCAST burner or generated from the FBG. See text for details of instrument details and abbreviations.

system (Model VKL10, Palas) using particle-free, VOC-filtered air, prior to entering the mixing volume. mCAST samples collected for analysis were directly sampled from the miniCAST outlet, upstream of the VKL10, on $47$ $mm$ diameter quartz fiber filters (Tissuquartz Filters, Type 2500QAT-UP, Pall Inc.), using a $47$ $mm$ aluminum in-line filter holder. The filter holder was mounted at a distance of $10$ $cm$ downstream of the miniCAST exhaust pipe, using an air-cooled stainless-steel pipe and connected to a vacuum pump, operated at a constant flow rate of $20$ $Lmin^{-1}$. Soot aerosols were then carefully removed from the filters with a metal spatula for bulk particle analysis as specified in Sect. 2.4.

**Aerosol selection:** From the mixing volume particles were size selected before being sampled into HINC for ice nucleation analysis (see Supplementary Information (SI) Sect. S6). Employment of a mixing volume enhances coagulation, increasing the size of the soot aggregates and buffers any fluctuations in particle generation allowing for a constant particle number concentra-

tion during the ice nucleation experiments. From the mixing volume, particles were sent through a DMA (Differential Mobility Analyzer; Classifier 3080, with a 3081 column and a polonium radiation source, TSI Inc.), where they were size-selected based on their dry electrical mobility diameter, $d_m$ (see SI Table S1 for details of mobility sizes selected and associated flow settings within the DMA). The ice nucleation experiments presented below include multiple-charged particles, i.e. particles that are

larger than the electrical mobility size selected. The amount of double-charged particles is approximately 6 and 12 % for the 100 and 200 nm particles respectively and 15 % for the case of selecting mobility diameters 300 and 400 nm (Wiedensohler, 1988). Downstream of the DMA, the flow was split using a four-way flow splitter (Model 3708, TSI Inc.) to the various instruments, as depicted in Fig. 1. Particle number concentration was monitored using a CPC (Model 3776, TSI Inc.), operated in the low flow mode (0.3 Lmin$^{-1}$).

**Ice nucleation:** Two similar HINC chambers were connected to the flow splitter, with the last port being used for exhaust flow to avoid over-pressuring the setup. The two HINC chambers used encompass HINC*ST* and HINC*NE*, where the suffix *ST* denotes standard and *NE*, non-evaporation operation conditions, respectively. An evaporation section at the end of HINC*NE* can selectively be switched on or off and HINC*NE* used here is identical to the chamber described in Lacher et al. (2017), i.e. not making use of the evaporation section and as such, identical to HINC*ST*. To ensure that results obtained from both chambers

are comparable, the chambers were characterized prior to the soot experiments using aerosols with thermodynamically well defined ice formation properties and good agreement was found within instrumental uncertainty (see SI Fig. S16). For this reason, we generally refer to both instruments as HINC in the following and combine their data for our analysis. To avoid external contamination from entering the setup, the system was run with an over-pressure from the aerosol generation source to the splitter. The over-pressure was regulated with a needle-valve controlled exhaust, mounted to the mixing volume, where the pressure

was continuously monitored using a pressure sensor. This was possible as the total flow from the aerosol generation was approximately 12 Lmin$^{-1}$ in case of the FBG and 35 Lmin$^{-1}$ in case of the miniCAST. In addition, a stable correct flow through the DMA was ensured by checking the flow between the DMA and the flow splitter over regular intervals of approximately 3 h.

## 2.3    Ice nucleation experiments and data processing

Ice nucleation experiments with HINC were conducted in deposition and/or condensation mode between 218 K and 253 K, in 5 K steps. HINC is a continuous flow diffusion chamber (Rogers, 1988; Rogers et al., 1998), based on the design of the UT-CFDC (University of Toronto Continuous Diffusion Chamber; Kanji and Abbatt, 2009) and is described in detail by Lacher et al. (2017). Briefly, two parallel, horizontally oriented copper plates separated by a 20 mm polyviniylidene fluoride (PVDF; Angst+Pfister AG, Zurich, Switzerland) spacer are lined with borosilicate glass fiber filter paper (Type A/C, Pall Corporation)

which is wetted prior to cooling. Upon cooling of the walls (copper plates) to subzero temperatures, a smooth ice layer is formed on the wetted filter paper. The wall temperatures are controlled by two thermostats (LAUDA ProLine RP890C) to keep temperature fluctuations around 0.1 K and are monitored through four equally spaced thermocouples along each wall. When held at different temperatures, a linear temperature gradient is established between the upper (relatively warmer) and lower (relatively colder) wall, resulting in a parabolic water vapor supersaturation profile across the chamber, due to the nonlinear

relationship between saturation vapor pressure and temperature (Clausius-Clapeyron equation). Ice particles can be formed within the chamber by exposing the injected aerosol particles to conditions of $RH_i > 100$ %, where the subscript $i$ denotes evaluation with respect to ice, and cloud droplets can be formed for conditions $RH_w > 100$ %, where the subscript $w$ denotes evaluation with respect to water. Freezing experiments with salt particles have been performed and compared to theoretical predictions from the water activity based homogeneous freezing parametrization of solution droplets (Koop et al., 2000) to verify chamber performance (Lacher et al., 2017). The flow in HINC is maintained by two mass flow controllers (MFC, G-Series, MKS Instruments, Andover, USA), which control the sheath air and total flow. The sheath air flow ($F_{sheath}$) is made up of particle-free $N_2$ and confines sampled particles to a lamina at the center of the two walls, corresponding to a specific temperature and $RH$ while preventing interaction of the particles with the walls. The total flow through HINC ($F_{OPC}$) denotes the flow that is sampled by the optical particle counter at the chamber outlet. The aerosol containing sample flow ($F_{AP}$), drawn into HINC, is given by the difference of $F_{OPC}$ and $F_{sheath}$. The aerosol flow was chosen such that the ratio of $F_{AP}$ to $F_{sheath}$ was between 1:10-1:12, by adjusting $F_{sheath}$ and keeping $F_{OPC}$ fixed at 2.83 $\mathrm{Lmin}^{-1}$. Both $F_{AP}$ and $F_{sheath}$ are introduced into HINC at approximately room temperature conditions, however, $F_{sheath}$ is introduced at the beginning of HINC (prior to $F_{AP}$) and thus will reach steady state conditions of temperature and water vapor upon entering the chamber prior to joining the aerosol flow. $F_{AP}$ ($\approx 1/10^{th}$ of $F_{OPC}$) should equilibrate with the temperature and to the saturation conditions in HINC within $0.2 - 2$ s, as described by Kanji and Abbatt (2009) and Lacher et al. (2017), depending on the temperature in HINC. The particle residence time within HINC is purely a function of the position of the movable injector as described by Lacher et al. (2017), for a constant total flow ($F_{OPC}$). Assuming a perfectly parabolic velocity profile across the chamber, the aerosol particles are assumed to travel at the maximal velocity at the center of the profile, which is used to derive the particle residence time in the chamber, i.e. the time it takes a particle to cross the chamber. The residence time can be divided into the time it takes to nucleate an ice crystal (or activate a water droplet) and the subsequent growth time of the particle within the chamber. For all experiments presented here a particle residence time of $\tau \approx 16$ s was chosen. This is well above the maximum time needed for the airstream to reach steady state conditions within HINC ($0.2 - 2$ s) and allows the nucleated ice crystals to grow to sizes $> 1$ mum (ice detection threshold size) in diameter within the chamber. At the HINC outlet, an optical particle counter (OPC, Model GT-526S, MetOne) is used to detect particles. The OPC can count and size particles in the size range between 0.3 μm and 10 μm (optical diameter) and can be operated at six different, customizable size bins within this range. However, it does not have phase discrimination capability, as such discrimination between interstitial aerosol particles, cloud droplets and ice crystals is based purely on optical particle size. The OPC was operated in *normal* (cumulative) mode such that the number counts within each channel correspond to particles of that optical size and larger. Here, we choose the 1 μm size bin as the threshold to detect ice crystals in HINC, i.e. particles with optical diameters $> 1$ μm. This threshold size was set above the size of aerosol particles entering HINC but small enough to capture the onset of ice formation. Conditions at which water droplets grow to sizes larger to the ice crystal threshold size is referred to as water droplet survival (WDS) relative humidity and is a function of *T*, initial aerosol size and residence time in the chamber. Beyond that size (bin) ice crystals and cloud droplets cannot be distinguished any longer. Water droplet growth was calculated assuming pure diffusional growth based on Rogers and Yau (1989) for a given initial diameter corresponding to the size of the selected aerosol, and assuming the entire residence

time within HINC is available for diffusional growth, i.e. neglecting any nucleation time. Therefore, the calculated values represent an upper limit of droplet growth and thus a lower limit of the WDS relative humidity reported. In our case the WDS line is only slightly above water saturation due to the 1 μm OPC channel used to report ice nucleation (see Fig. 2). Data points occurring above the WDS line are not considered to represent heterogeneous ice nucleation. Nevertheless, we cannot exclude

that heterogeneous ice formation takes place simultaneously with or after droplet formation. For all experiments the OPC was operated at a logging interval of 5 s, such that the detected counts within a given size channel yield the cumulative counts detected during this period. While most of the samples were probed using the six channel MetOne OPC, some experiments were performed using a four channel LightHouse OPC (Model R3014) during instrument unavailability. However, the same settings and analyses were applied to ensure consistency in data treatment and comparable results. Ice nucleation abilities of

the soot aerosol were probed by so-called $RH$ ramps where the aerosol is exposed to increasing $RH$ at fixed $T$. To achieve this, the desired lamina (center) temperature is set and then the $RH$ is increased by cooling and warming the temperatures of the two walls at the same rates. Instrumental noise from HINC, including occasional frost particle detection, was determined with a particle-free flow in HINC at the beginning and the end of each $RH$ scan. The background counts, corresponding to a linear interpolation of the OPC counts detected during the particle-free periods, were subtracted from the data and all of our

reported values are above this background level, as described in detail in Lacher et al. (2017).

The fraction of aerosol particles that nucleated ice is called the activated fraction ($AF$), defined as the ratio of the number concentration of ice particles detected by the OPC in the 1 μm size channel, $n_{ice,CH>1\mu m}$, to the total number concentration of particles entering the chamber, $n_{tot}$, as determined by the CPC in parallel to HINC:

$$AF = \frac{n_{ice,CH>1\mu m}}{n_{tot}}. \tag{1}$$

The uncertainty range in $AF$ is $\pm 14$ %, resulting from a 10 % counting uncertainty in each of the OPC and CPC. The fractal-like nature of soot aggregates causes the structure of monodisperse aerosols to vary on a particle to particle basis (Park et al., 2004), affecting aerosol properties such as shape and surface area thus adding to the complexity when size-selecting soot aggregates. This resulted in larger aerosol particles (optical diameter $> 1$ μm) being detected in the ice channel making it necessary to aerosol-correct some of the $AF$ curves, when evaluating the 1 μm OPC channel. In these cases the mean $AF$

value detected by the OPC (channel) at low RH values, where no ice crystals are formed, was subtracted from each data point of the $AF$ curve, correcting for any false signal in $AF$ arising from large unactivated aerosol particles. Finally, all ice nucleation results presented below correspond to mean $AF$ values observed over a range of two to eight $RH$ scans performed at a given temperature. We constrained the ramp rate of our $RH$ scans to below 3 %min$^{-1}$, evaluating $RH$ with respect to water. The CFDC data was then linearly interpolated into bins of 0.25 % $RH_{\mathrm{w}}$, where the subscript $w$ denotes evaluation with respect to

water. Thus, each reported curve represents the arithmetic mean of all binned and interpolated $RH_{\mathrm{w}}$ scans performed at a given temperature for a given soot sample.

## 2.4 Auxillary measurement for sample characterization

A suite of auxiliary measurements were conducted to characterize the physicochemical properties of the tested soots, which ultimately contribute to their ice nucleation behavior. Details of the measurements can be found in Appendix A and are briefly described in the following. The fractal extent of the soot aggregates was determined through TEM and coupled DMA-CPMA
(Centrifugal Particle Mass Analyzer, Cambustion Ltd., Cambridge, UK ) measurements to assess the fractal nature of the soots. Therefore, the TEM sampler and the CPMA were operated directly downstream of the flow splitter depicted in Fig. 1, i.e. on (mobility) size selected aerosol particles. In addition, bulk soot properties were investigated by means of a thermogravimetric analyzer (TGA; Model Pyris 1 TGA, PerkinElmer) to determine the particle fraction that can be volatilized as a function of temperature, referred to hereafter as the mass loss at a given temperature. This was done to determine the presence of any
secondary material coating the carbon spherules of the soot aggregates. In order to determine the hydrophilicity and infer the porosity of the samples, gravimetric water vapor sorption isotherms of the bulk soot samples were measured by Dynamic Vapor Sorption (DVS, Model Advantage ET 1, Surface Measurement Systems Ltd., London, UK). Finally, the BET specific surface area of the bulk soot samples was determined from additional $N_2$ adsorption measurements ($a_{BET,N2}$). The $a_{BET,N2}$ values were used in combination with the water sorption measurements to derive equivalent water monolayer coverage.

## 3 Results and discussion

### 3.1 Overview of ice nucleation measurements

An overview of the ice nucleation experiments is given in Fig. 2. From this overview a few key features are inferred concerning the ice nucleation ability of the tested soot particles. A clear dependence of ice nucleation of soot on the temperature regime can be identified. For $T > 235$ K all soot types require water saturation for activation into ice crystals and/or cloud droplets.
The fact that the activation onset conditions for all soot types (independent of particle size) lie at $RH_w \geq$ WDS (dashed red line in Fig. 2), indicates an absence of heterogeneous freezing for $RH_w <$ WDS. However, we re-iterate that we cannot unambiguously distinguish between cloud droplets and ice crystals at $RH_w \geq$ WDS, given that cloud droplets can grow to 1 μm by water vapor diffusion and be detected as ice crystals. Furthermore, evaluating the ice nucleation data for the 5 μm OPC channel, where WDS is well above water saturation for the residence time of $\tau \approx 16$ s, reveals that activation onset only
takes place for $RH_w \geq$ WDS (see SI Sect. S3). Thus, we can conclude that no condensation/immersion freezing of soot takes place in the region 100 % $< RH_w <$ WDS for the MPC temperature regime. This interpretation is consistent with Koehler et al. (2009), who did not find ice nucleation at temperatures above 233 K. Conversely, in the cirrus temperature regime some soot particles nucleate ice below homogeneous freezing conditions. From this, we conclude that the unprocessed soot types investigated here can significantly contribute to ice nucleation only in the cirrus regime.
A strong increase in the freezing ability below the homogeneous nucleation temperature (HNT, 235 K) has been observed before for kaolinite (Welti et al., 2014) and suggests the involvement of a liquid water phase in the ice formation process, similar to the soot particles tested here.

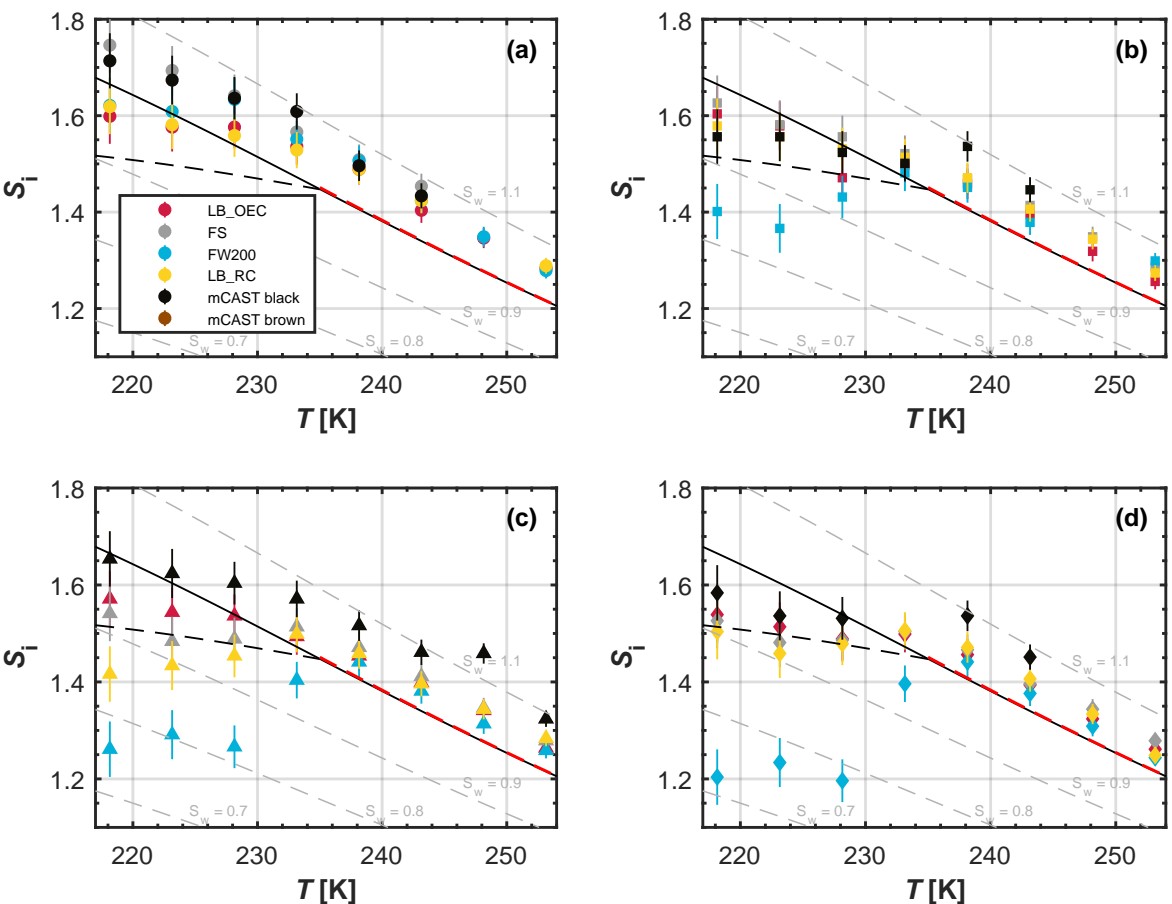

**Figure 2.** Onset saturation ratio with respect to ice ($S_i$) of freezing where $1\ \%$ of the aerosol particles are activated as a function of temperature for size-selected particles with mobility diameter of (a) $100$, (b) $200$ (c) $300$ and (d) $400\ nm$ (see SI Sect. S6). The black, solid lines denote water saturation using the water saturation pressure parameterization given by Murphy and Koop (2005); gray, dashed lines indicate offsets from this line in steps of $0.1$ to guide the eye. The black, dashed lines mark the homogeneous freezing threshold for supercooled solution droplets according to Koop et al. (2000). The red dashed line indicates WDS conditions. WDS appears close to water saturation for the $1\ \mu m$ OPC channel used. Each dot represents the mean of a minimum of two $RH$-scans performed with HINC. Temperature uncertainty is $0.1\ K$ and $S_i$ uncertainties, outlined in Appendix D1, are indicated as vertical bars. We note that mCAST black and brown do not activate to $1\ \%$ and are thus not plotted for some conditions (see SI Sect. S2 for complete $AF$ curves).

Moreover, a clear size dependence is observed, with larger soot aggregates showing activation at lower $RH$ at a given temperature. A size dependence is consistent with the INP requirements reported in Pruppacher and Klett (1997) proposing that INP typically have (physical) diameters $\geq 200\ nm$. While these findings are confirmed by more recent studies for mineral dust (e.g. Archuleta et al., 2005; Welti et al., 2009), ice nucleation measurements of size selected (e.g. Friedman et al., 2011; Kulkarni et al., 2016) or nearly monodisperse soot particles (DeMott, 1990) are limited, especially in the cirrus regime. The observation

that the $100$ nm soot particles only induced ice formation above homogeneous freezing conditions, suggests that particles of this size are not relevant for forming ice crystals at $T < 233$ K, irrespective of morphology and composition since homogeneous freezing of droplets will out-compete any heterogeneous nucleation by soot particles in the cirrus temperature regime. Besides a dependence on the minimum aerosol size, ice nucleation has been suggested to depend on the chemical composition of the INP (Pruppacher and Klett, 1997). This is consistent with our results, which only show differing ice nucleation abilities among the soot types for particles with sizes of $200$ nm or larger tested here. Nevertheless, the absence of freezing of our $100$ nm soot particles, at conditions below homogeneous freezing of solution droplets reveals that both particle size and chemical properties determine the ice nucleation ability of the investigated soots. To further elucidate which properties provide soots with ice nucleation activity, we focus here on the ice nucleation results of the most active $400$ nm particles in the cirrus regime. We show the complete activation curves in Fig. 3, in order to discuss differences in the ice nucleation mechanisms associated with the soot types with information gained from the auxiliary measurements performed on the soot samples.

## 3.2 Ice nucleation dependency on soot type

Figure 3 shows the $AF$ as a function of $RH_w$ for all investigated $400$ nm soot samples, covering the temperature range $218 - 233$ K. Given that the $RH_w$ range between homogeneous freezing condition and water saturation at $T = 233$ K spans only approximately $2$ % $RH_w$, which is within our instrumental uncertainty, we cannot exclude that water saturation conditions were reached for the observed freezing at $233$ K. Overall, soot particles require high ice supersaturations to nucleate ice, rendering them unlikely to contribute to heterogeneous freezing at $T = 233$ K within the residence time of HINC. An exception is FW200, which shows activation well below homogeneous freezing conditions for all temperatures shown in Fig. 3. This becomes more apparent at lower temperatures. At $T = 218$ K FW200 shows significant freezing at $RH$ as low as $RH_w = 70$ %. This heterogeneous freezing ability of FW200 is even more pronounced than that found by Koehler et al. (2009), who reported $AF$ of roughly $1$ % below homogeneous freezing conditions, when testing thermal oxidized soot, i.e. soot processed in a mixture of nitric and sulfuric acid, at $T = 233$ K. The lower ice nucleation onset of FW200 soot can be interpreted as a more active deposition nucleation process due to the larger surface area of the FW200 compared to the other soots (see Tables 1 and A1). However, the absence of ice nucleation below water saturation for $T > 233$ K is incompatible with a classical deposition nucleation (Welti et al., 2014). As mentioned above, the dependence on the HNT supports the interpretation that liquid water is involved in the freezing mechanism prior to reaching water saturation. This suggests a homogeneous freezing mechanism for the FW200 particles below water saturation, most likely not directly related to the particle surface area. Soot aerosols typically consist of agglomerated chains of spherical particles. They are prone to form cavities and/or pore-like features between the primary spherules and the individual chains, in which water can condense at water subsaturated conditions (inverse Kelvin effect, Marcolli, 2014), and can subsequently freeze homogeneously through a so-called pore condensation and freezing mechanism (Higuchi and Fukuta, 1966; Fukuta, 1966; Christenson, 2013; Marcolli, 2014). During PCF, water that is taken up in capillaries well below water saturation, can freeze homogeneously if the temperature is below the HNT and if the pore diameter is large enough to accommodate the critical ice germ. Freezing of pore water can subsequently trigger macroscopic ice growth, resulting in ice crystal formation. The steepness of the activation curves for FW200 shown in Fig. 3 resemble a

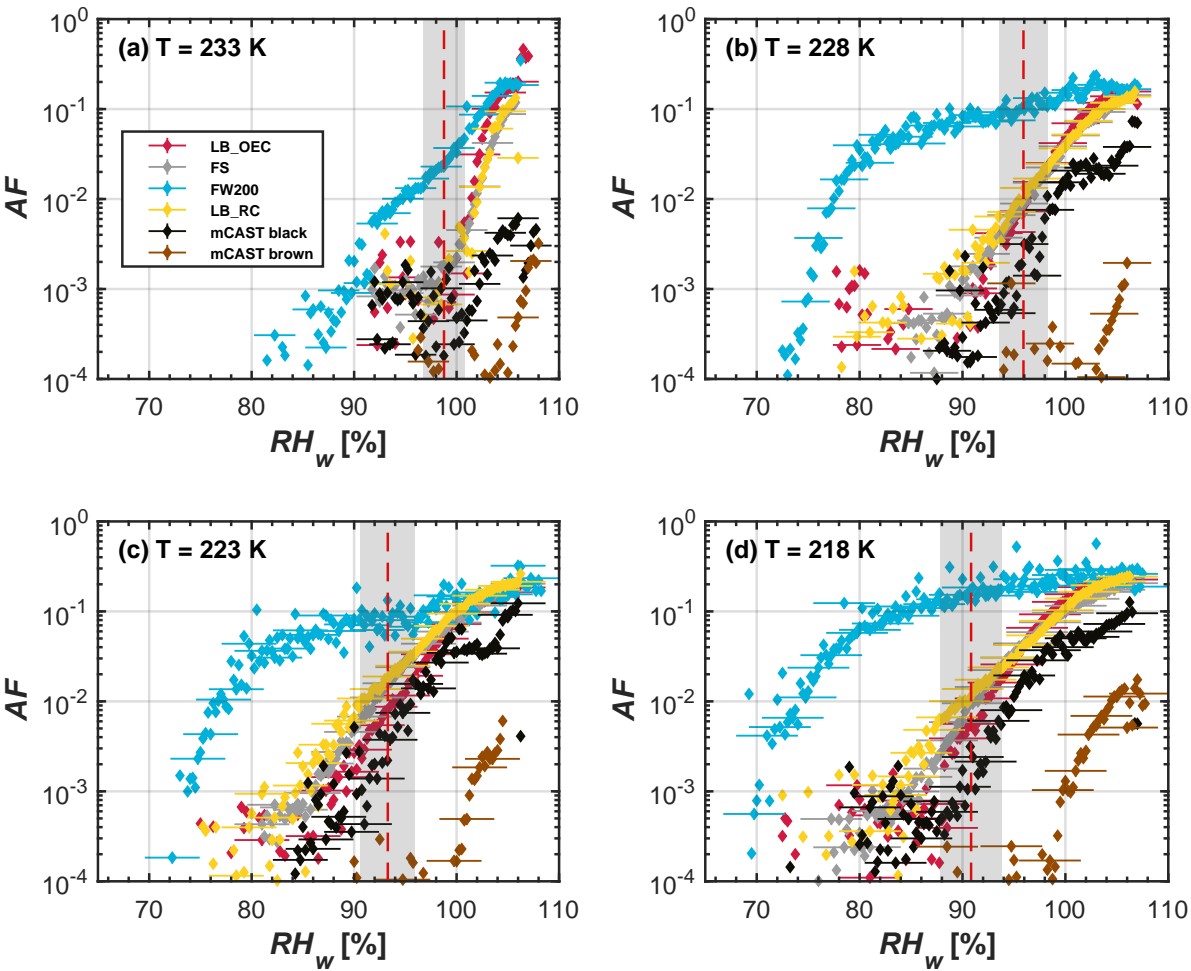

**Figure 3.** Mean $RH$ scans of $400$ nm mobility diameter soot particles, showing $AF$ as a function of $RH_w$ for different temperatures in the cirrus regime. Red dashed lines represent expected homogeneous freezing conditions according to Koop et al. (2000), and the gray shaded regions indicates the calculated $RH_w$ variation across the aerosol lamina in HINC. Uncertainties, given for every $5^{th}$ data point, are as in Fig. 2.

step-like activation when reaching a critical $RH$, consistent with a PCF type mechanism which requires the pores on the soot aggregate to fill and subsequently freeze homogeneously. In fact, the large specific surface area of FW200 itself (see Table 1 and A1), must be caused by concave and convex surfaces formed from the sintered primary particles and potentially indicates the presence of cavities. Additionally, aggregate porosity in the form of voids formed between sintered primary particles is supported by our TEM analysis (see Sect. 3.3). Yet, we note that none of the other soots tested shows a similarly clear ice nucleation behavior attributable to PCF, which should be more heterogeneous for the other soot types. Thus, the observed

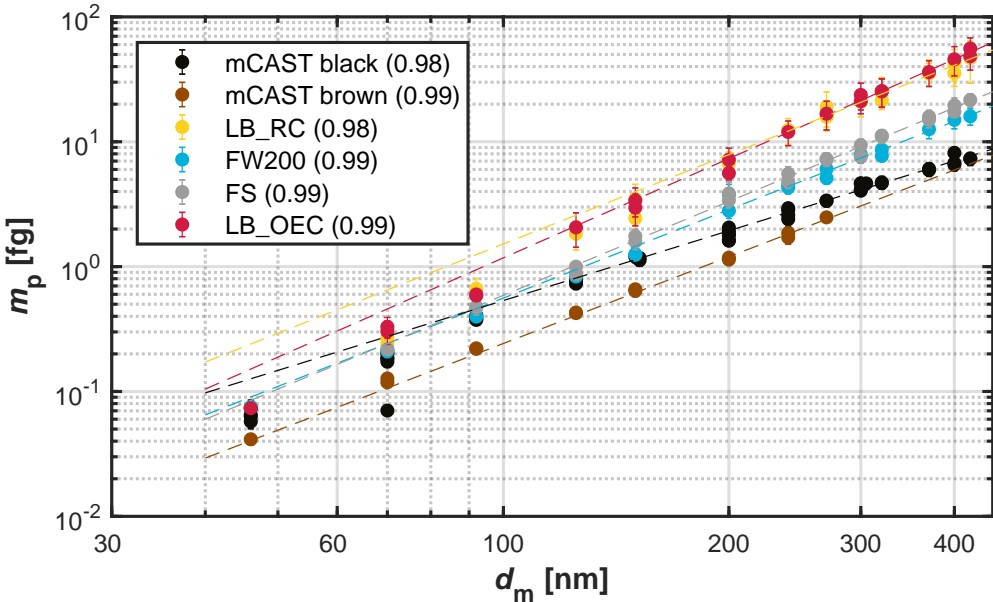

**Figure 4.** DMA-CPMA data for the different soot types: Electrical mobility diameter ($d_m$) vs. median particle mass ($m_p$) derived from fitting the resulting mass distribution with a log normal distribution. Error bars correspond to standard deviations of the individual measurements and dashed lines to the power law fits using eq.A4. The values in parenthesis give the $R^2$ of the fit.

difference must be related to the physical and chemical properties of the particles. We discuss these in the context of possible different freezing mechanisms between the soots in the following.

### 3.3 Soot particle fractal structure

Soot morphology was derived from DMA-CPMA measurements (see Appendix A4), and qualitatively from analysis of TEM

images (see Appendix C). The fractal dimensions, $D_{fm}$, of the different soot types are reported in Table 1 and the corresponding DMA-CPMA data is shown in Fig. 4. A $D_{fm}$ value of 1 corresponds to a straight chain-like structure, whereas $D_{fm} = 3$ indicates a compact sphere like structure. In general the values of the fractal dimensions lie within those observed in previous studies, such as the values reported for Diesel soot by Olfert et al. (2007), ranging from 2.2 to 2.9, or Abegglen et al. (2015), who observed values from 1.86 to 2.88 for particulate matter from aircraft turbine exhaust. Our DMA-CPMA measurements reveal

a similar mean fractal structure of the different soot types. However, we cannot identify a direct relationship of the fractal dimension to the ice nucleation efficiency of the soot types reported in Fig. 3 above.

Interestingly, the two lamp black soots (LB_OEC and LB_RC) show significantly higher masses for a given mobility diameter compared to the other soots. The higher mass for a selected size of the lamp black soots translates into larger effective densities for these soots, given as median particle mass divided by the equivalent volume of a sphere with this mobility di-

ameter. From our TGA mesurements we can exclude the contribution of secondary material to cause the higher masses of the

lamp black soots (see Sect. 3.4). The higher masses for the lamp black soots are more likely caused by larger primary particles making up these aggregates resulting from the wider size distribution of the primary particles. As noted above, the lamp black soots have a rather wide primary particle size distribution, causing the resulting soot aggregates to be composed of primary particles with different masses. This is also reflected in the relatively larger error bars of the lamp black soots in Fig. 4 and

5    further supported by our TEM analysis.

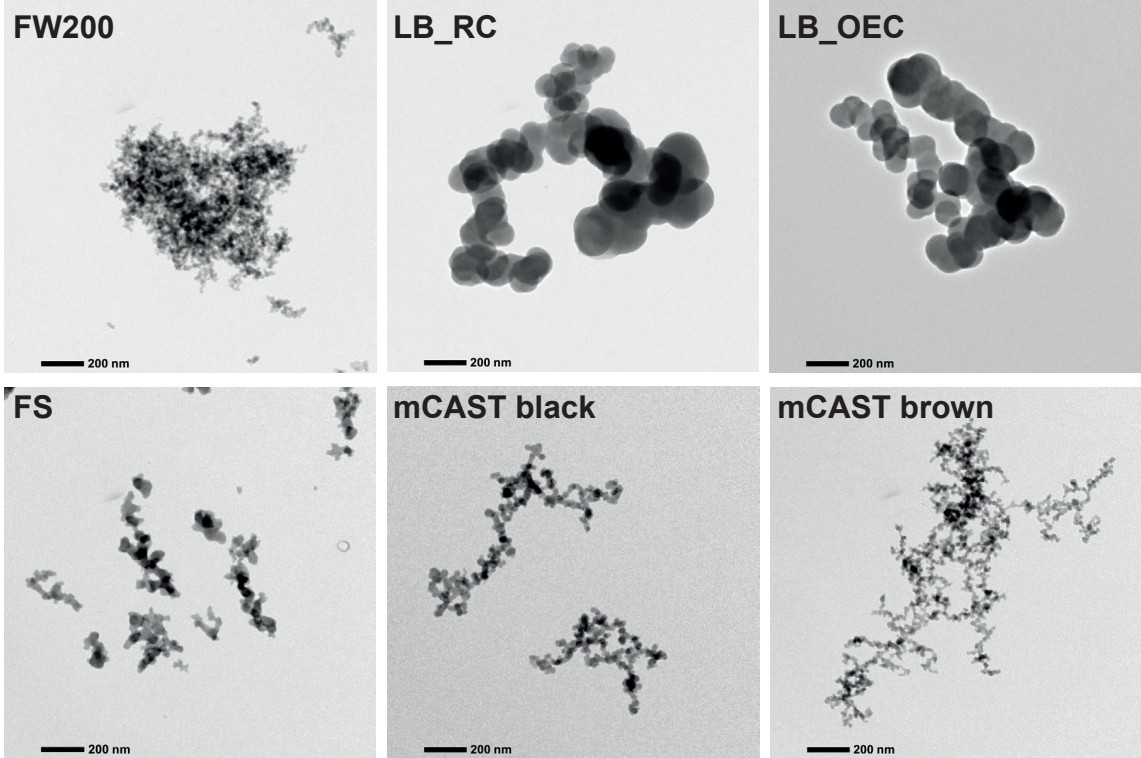

**Figure 5.** Exemplary TEM images of representative soot aggregates of the different soot types when selecting $400$ nm mobility diameter particles with the DMA.

In general our TEM results reveal large, fractal like aggregates composed of primary particles that form a variety of different micro structures. In Fig. 5 we show exemplary TEM images of the different soot types for $400$ nm size selected aggregates (see also SI Fig. S18). Images of both the LB_RC and LB_OEC reveal a range of differently sized primary particles making up the aggregates. This is further supported by the size distribution of primary particles shown in SI Fig. S17, for which between

10    10 to 50 aggregates and a minimum of 122 primary particles have been evaluated for each soot type. From the TEM images depicted in Fig. 5 we further note that the most ice active FW200 soot shows particularly densely clustered aggregates, whereas the other soots exhibit more branched, chain-like aggregates. It should be noted that these TEM images constitute only a 2D projection of the soot aggregates and can thus not be directly compared with the 3D fractal dimension derived from our CPMA-DMA measurements. We conclude from our TEM analysis that all soot types demonstrate aggregate structures that encompass

cavities and pores, which could enable a PCF freezing mechanism. Nevertheless, it is likely that the clear difference in primary particle size determines and/or strongly influences overall aggregate porosity. Soot aggregates of a given mobility size are composed of an increasing number of carbon spherules for decreasing primary particle sizes. As the number of spherules increases, the propensity for pores in an aggregate also increases due to the potential for pores between sintered spherules and/or through intra-aggregate cavities between the branches of the aggregate. As such, soot particles with smaller spherules are more likely to nucleate ice via a PCF mechanism due to the higher concentration of pores, resulting from the increased number of primary particles in these aggregates. Indeed, a recent study by David et al. (submitted) showed that macroscopic ice can only grow out of cylindrical pores if they are closely spaced. Therefore, it is possible that the enhanced ice nucleation capabilities of FW200 are caused by its small spherule size and the associated increased propensity for pores. However, it is important to note that the mCAST brown soot shows a similar spherule size distribution to the FW200 soot, but also a significantly more branched 2D projection of the aggregate and exhibits significantly lower ice nucleation ability compared to FW200 soot. Thus, the observed difference must be related to other physical and chemical properties of the particles, in addition to the morphology, which might be more heterogeneous for the other soot types. Next we discuss the presence of volatile material on the soot particles.

### 3.4 Temperature dependent mass loss

Besides carbon, soots usually contain OM which can include compounds with hydrophilic functional groups such as hydroxyls, carbonyls or carboxyls. Heating the soot samples leads to the loss of this OM, depending on the heat resistance of the compounds. A loss in mass encompasses the volatilization of associated OM and/or the thermal decomposition of OM at higher temperatures. The mass loss of the soots as a function of $T$, measured by TGA, is shown in Fig. 6. Here we focus on the mass decrease within three different temperature regimes (ranges labelled A, B, and C, respectively, in Fig. 6).

Highly volatile compounds, classified as volatile at temperatures below 200 °C, evaporate under inert gas flow (region A in Fig. 6). In the range below 100 °C FW200 shows the strongest reduction of mass of approximately 7 % (7.5 % and 6.5 % for samples with solid and dashed lines, respectively). We interpret the strong decrease in mass of FW200 to be caused by the loss of adsorbed/condensed water and/or low molecular weight organic substances. The observed mass decrease of approximately 9 %, when drying the FW200 sample for 1000 min at 25 °C by continuous $N_2$ flushing prior to the DVS measurement, supports the interpretation that the mass loss is due to evaporation of water (see SI Fig. S14). Besides, the mass loss rate is highest below 100 °C, likely due to evaporation of adsorbed water vapor and stays approximately constant between 100 °C and 400 °C (see SI Fig. S13). Overall, the strong mass decrease of FW200 below 200 °C may reflect the presence of strongly hydrophilic sites on this sample, where moisture can adsorb. The high affinity of FW200 to adsorb water is further reflected by the DVS experiments (see Sect. 3.5). In contrast, the equivalent mass loss for the mCAST black and the lamp black soots is significantly lower, measured as less than 1 % up to temperatures of 100 °C. A water content of at most 1 % indicates only a limited number of hydrophilic active sites or pores where water can be adsorbed. While most soot types show strong mass loss below $T = 100$ °C the mass of mCAST brown is relatively constant within this temperature range but then decreases for temperatures between approximately 140 °C and 200 °C. Up to $T = 200$ °C the mass loss of the mCAST brown soot is likely associated with highly

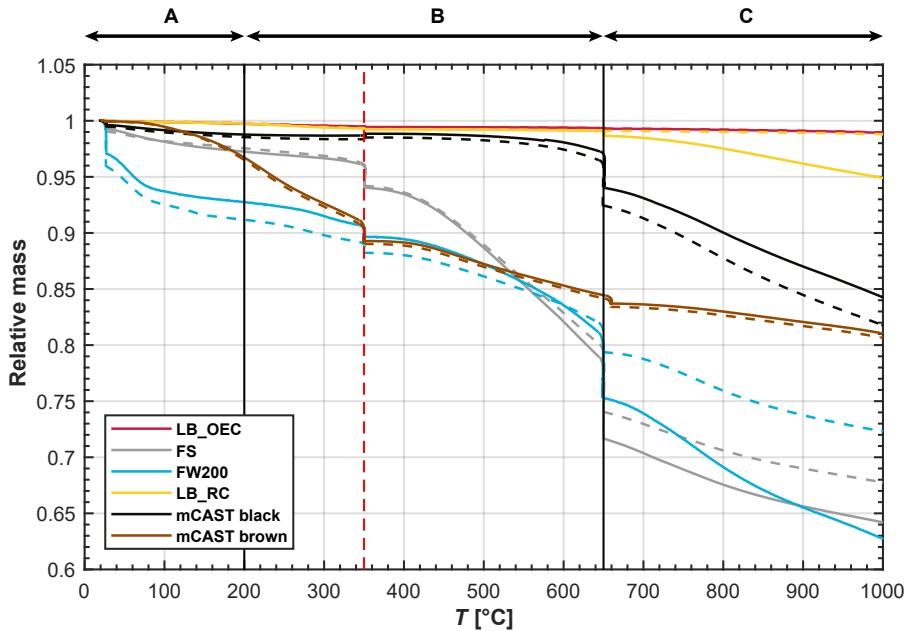

**Figure 6.** Results of the TGA analysis of the investigated soot types in a nitrogen atmosphere, showing the relative sample mass as a function of sample temperature. Solid and dashed lines indicate the results of two individual TGA runs. Indicated regions correspond to different ranges of volatilization, namely adsorbed water and highly heat sensitive material (A), medium heat sensitive material (B) and low heat sensitive material (C). Pyrolysis is expected to contribute to mass loss at temperatures above the vertical red dashed line.

volatile, low molecular mass polyaromatic hydrocarbons (PAH), that have previously been associated with brown carbon from the miniCAST burner (Mueller et al., 2015) and which have been reported to volatilize between approximately 100 °C and 200 °C (Portet-Koltalo and Machour, 2013). Conversely, the mCAST black samples show a smaller mass reduction, indicating a lower proportion of volatile OM, illustrating the chemical difference to the mCAST brown sample. The increased OM content

on brown carbon results from a reduced combustion temperature and a less complete combustion compared to mCAST black as a consequence of an increased amount of $N_2$ premixed to the fuel for this test point (see Appendix B). This supports the interpretation from above that the mass loss observed for mCAST brown is mainly due to OM and not due to sorbed water vapor, which above $T = 100$ °C should have already been desorbed. The lack of sorbed water also suggests that the OM is likely hydrophobic in nature. Considering our ice nucleation results reported in Fig. 3, the shift of the onset of mCAST brown

to higher supersaturation compared to the mCAST black (and the other soots) suggests that these hydrophobic hydrocarbons suppress the freezing abilities of the soot. This interpretation is in agreement with the results from Crawford et al. (2011) and Möhler et al. (2005b) who find freezing for propane soots with higher organic carbon content required higher supersaturation for ice nucleation.

The FS sample demonstrates a strong mass loss above 350 °C, indicating a relatively large abundance of material with medium

volatility compared to the other soots. While some portion of this mass loss is attributable to OM with medium volatility (see

Fig. 5), it is likely that thermal decomposition in the form of pyrolysis also contributes to mass loss (Bredin et al., 2011; Song and Peng, 2010). Besides, we cannot exclude combustion of carbon in the presence of trace amounts of oxygen (trapped within the sample, instrument parts or from gas impurities), which has been shown to contribute to mass loss between $430 - 650$ °C, even for $N_2$ prepurged samples. Bredin et al. (2011) found combustion (in the presence of oxygen) contributing to a mass change of a Printex U soot (gas black) in TGA measurements performed in a $N_2$ atmosphere. They reported combustion starting around $400$ °C and reaching a maximum rate around approximately $500$ °C, similar to the behavior of our FS soot. In conclusion, both pyrolysis and combustion can occur in this temperature range, depending on the presence of $O_2$ in the reaction chamber. The consistent mass loss in this temperature range for both independent TGA runs with FS (solid and dashed line), combined with the negligible mass loss of the lamp blacks and the mCAST black in this temperature range, suggests that most of the mass loss is caused by the evaporation of low-medium volatility OM. For mCAST brown, this likely includes larger, higher molecular weight PAHs which have been reported to volatilize between approximately $125$ °C and $670$ °C (Wang et al., 2012). If the observed mass loss of the FS aggregates was due to water-soluble material, this material could take up water and lead to the formation of aqueous solution droplets which subsequently freeze homogeneously. This process could account for the freezing reported in Fig. 2, however the lack of any change in the slope in the $AF$ curves of FS at homogeneous freezing conditions (see SI Fig. S10) similar to the change in slope of the $AF$ curves observed for homogeneous freezing of ammonium nitrate (see SI Fig. S16), render this interpretation unlikely.

Around $400$ °C the organic fraction should be completely vaporized and desorbed (Stratakis and Stamatelos, 2003). Thus soot samples which exhibit a strong mass loss beyond this temperature, namely FS, FW200, mCAST brown and to some extent also mCAST black, undergo continuous pyrolysis or contain inorganic heat sensitive material.

Altogether, TGA analysis confirms that mCAST black and the lamp blacks can be considered as pure carbons, with minimal impurities in the form of OM, while it illustrates the presence of organic material on the mCAST brown and on the FS. Overall the only sample that shows a significant mass loss for $T < 100$ °C (likely adsorbed or condensed water) is FW200, which corroborates its high ice nucleation ability compared to all the soot samples investigated.

### 3.5   The role of soot-water vapor interaction

An important parameter for the ice nucleation ability of the different soot types is their potential to take up water (Dubinin, 1980). The water uptake capacity is controlled by both the soot morphology (surface area and microstructure, Dubinin and Stoeckli, 1980) and the chemical composition, i.e. the availability of soluble material on the soot surface and/or the presence of hydrophilic functional groups, such as carboxyl, carbonyl and hydroxyl groups (Ferry et al., 2002). We used DVS to measure the water vapor sorption and desorption isotherms of the soots, allowing to infer the particle porosity and hydrophilicity at the same time. Figure 7 shows an overview of the water uptake of the different soots, measured by DVS at $T = 298$ K and covering the range of $0-98$ % $RH_w$. The water uptake is calculated from gravimetric data, expressed in percent as relative mass change difference between the initial dry mass and the quasi-equilibrated, moist sample mass at the probed $RH$ (Fig. 7a), and expressed in terms of equivalent adsorbed water monolayers (Fig. 7b). We use the term *water uptake* to encompass both effects of adsorption and absorption. Adsorption describes the enrichment of water molecules on the solid soot surface through binding

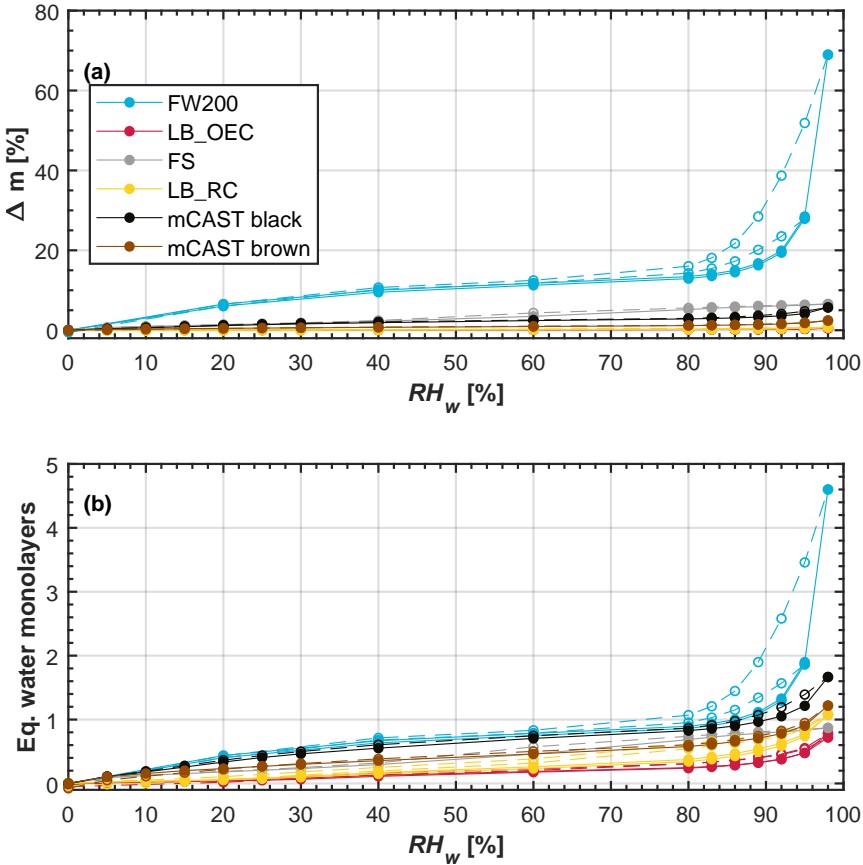

**Figure 7.** Water uptake (solid lines, filled symbols) and loss (dashed lines, open symbols) isotherms, given as (a) relative sample mass change and (b) equivalent water monolayers, as a function of $RH_w$, as measured by DVS at $T = 298$ K. Each set of curves corresponds to an individual, independent soot sample. The data points represent water uptake/loss at quasi-equilibrated $RH_w$ conditions. Each soot type was probed for two independent experiments in the DVS and the curves of the two independent runs lie on top of each other. Exceptions are the mCAST samples, which were only probed once in the DVS. Curves are linear interpolations between quasi-equilibrated sample points to guide the eye. The curves of the LB_OEC and LB_RC samples lie on top of each other in panel (a). All experiments were performed to a maximum $RH_w$ of 98 %, with the exception of one FW200 sample showing a maximum relative mass change of approximately 30 %, which was only probed up to $RH_w = 95$ %.

of water molecules to hydrophilic sites and the formation of water molecule clusters or equivalent monolayers. Absorption, on the other hand, refers to water molecules dissolving into soluble material associated with the particles (Popovicheva et al., 2008; Thommes et al., 2015). In Fig. 7, water uptake curves are shown by solid lines along with filled symbols, whereas water loss curves are shown by dashed lines and open symbols. Each set of (uptake/loss) curves corresponds to an individual,

5   independent experiment, using a fresh sample of the corresponding soot type.

The uptake curves of FW200 soot rise strongly at low $RH$ ($RH_w \leq 20$ %), exhibiting a convex shape, followed by a a slow

rise at intermediate $RH$ (20 % $\leq RH_w \leq$ 80 %), and a sharp increase in water mass taken up at high $RH$ ($\geq$ 80 %), typical for a type IV(a) isotherm (following classification recommendations according to IUPAC, see Sing et al., 1985; Thommes et al., 2015). In the FW200 aggregates water is initially taken up by *equivalent* monolayer adsorption on the soot surface (see Fig. 7b). The increase in mass at higher $RH$ marks the transition to multilayer adsorption followed by the transition to capillary

(pore) condensation at $RH_w \geq$ 89 %, signified by the sudden rise in mass. This is particularly clear from the FW200 sample probed until a maximum value of $RH_w =$ 98 % (water uptake curve with maximum mass change $\Delta m_{max} \approx$ 70 % in Fig. 7a). Due to the strong mass increase of the FW200 soot when probed until $RH_w =$ 98 % and the long time scales required to equilibrate the sample at these high $RH$ (around 800 min) we restricted further measurements of FW200 to $RH_w =$ 95 %. However, all other soots were tested up to $RH_w =$ 98 %. We attribute this strong water uptake at high $RH_w$ to mesopores

(2 − 50 nm, IUPAC, Thommes et al., 2015) formed by inter-particle cavities of the soot aggregates which create the porous structure of soot already discussed above (see Sect. 3.3). The high water affinity of FW200 revealed by the DVS measurements is consistent with the observed ice nucleation at relatively low ice supersaturations. We interpret this as a low soot-water contact angle for this sample. Besides, the availability of mesopores inferred from DVS supports our hypothesis that the mechanism of ice formation on the FW200 soot is indeed PCF. The steep AF curves of FW200 compared to the other soots, as discussed

in Sect. 3.2, this can be thought of as an overlap of pore size distribution and associated contact angle distribution that favours condensing water in pores, resulting in particles with properties suitable to nucleate ice via PCF, whereas those soot types with less steep AF curves indicate a more heterogeneous distribution of particle properties (contact angles and pore sizes).

This further corroborates our discussion above that the mass loss of FW200 observed in the TGA at temperatures $T \leq$ 100 °C is associated with desorbing water. Water uptake by OM could also contribute to the mass increase observed in our DVS

experiments. Taking citric acid (CA) and glutaric acid (GA) as surrogates for typical atmospheric organics and using the mass growth factors reported in Zardini et al. (2008), we find a mass fraction of 13.6 % (CA) and 20 % (GA), respectively, to be required to explain the observed mass change of almost 30 % at $RH_w =$ 95 %. However, our TGA results only indicate a mass change of approximately 8 % and 10 % at the boiling points of CA (310 °C) and GA (200 °C, see Fig. 6). Thus, we confidently attribute the water uptake of FW200 to pores present on the soot aggregates, and not only due to absorption of

water vapor by hydrophilic OM, which is also present, as suggested by our TGA results.

The water uptake on FW200 is associated with a hysteresis as the water uptake and loss curves do not overlap over the complete $RH$ range. The presence of a hysteresis loop further indicates that the mechanism of water uptake cannot be attributed to absorption by hygroscopic OM, but is indeed caused by the availability of pores and capillary condensation. The form of the hysteresis loop can be used to infer information on the pore structure. For the FW200 sample probed until a maximum $RH$

of $RH_w =$ 98 % the hysteresis looks like a H3-type hysteresis loop that is relatively narrow, where adsorption and desorption curves are almost parallel (Thommes et al., 2015), but associated with a very steep desorption branch. In contrast, the FW200 sample probed until $RH_w =$ 95 % ($\Delta m_{max} \approx$ 30 % in Fig. 7a) resembles an H4-type hysteresis with a shallow hysteresis loop, observed for aggregated, non-rigid particles such as mesoporous soots (Thommes et al., 2015). Contrary to typical H3 or H4-type hysteresis loops, the FW200 soot shows hysteresis down to very low $RH_w$, reflecting that either micropores are

developed or present to some extent on FW200 or that some water vapor is taken up through chemisorption on active sites of the

OM. While most of the hysteresis values of the FW200 soot lie within the DVS uncertainty of approximately $\Delta m = 0.75$ %, micropore filling is supported by the values of the $C$ constant in the classical BET equation, describing the adsorbate-adsorbent interaction strength. The $C$-value of FW200 was found to be $> 650$ (see Table A1), indicating filling of narrow micropores (Thommes et al., 2015). The shape of the hysteresis loop indicates involvement of mainly conical and slit-like pores as the majority of the water is lost already at high $RH_\mathrm{w}$. Pores with narrow necks, so-called ink-bottle shaped pores, which remain filled during desorption until low $RH$ (Kittaka et al., 2011), would cause a much stronger hysteresis at lower $RH_\mathrm{w}$ and are thus likely absent. This is consistent with the different hysteresis types, namely H3 and H4, associated with the FW200 probed up to different maximal $RH_\mathrm{w}$. The H3 hysteresis loop indicates *narrow* slit-like pores formed through sintering of carbon spherules. The H4 hysteresis includes also (normal) slit like pores formed in between spherules that are not adjacent (but positioned along different branches of the soot aggregate, see Fig. 5). From these observations we conclude that particle morphology and contact angle determine the water uptake by soot particles, confirming a PCF mechanism for the observed ice nucleation on FW200. This interpretation is also consistent with the observed ice nucleation onsets reported in Fig. 2. Therefore, our interpretation that the mass loss of FW200 during the TGA analysis is consistent with the evaporation of adsorbed water vapor and that the ice nucleation proceeds via a PCF mechanism can further explain the difference in freezing behavior between FW200 and the *hygroscopic* soot tested by Koehler et al. (2009). Their Aircraft Engine Combustor soot showed freezing only close to homogeneous freezing conditions for $T = 233$ K and 221.5 K, both for testing $d_\mathrm{m} = 250$ nm and polydisperse aerosol. Our ice nucleation results for FW200 reveal that the conditions for 1 % of the particles being activated in the cirrus regime are met at roughly the same $RH$ across all temperatures, for a given particle size. The constant onset $S_\mathrm{i}$ indicates that the freezing is in fact determined by pore size and soot-water contact angle, which determine water filling of the pores and subsequent homogeneous freezing. The different onsets reported for the different particle sizes (see SI Fig. S1) suggest that the pore size distribution is a function of the overall soot aggregate size and the associated number density of spherules, forming these cavities (see Fig. 5). An exception are the results at $T = 233$ K, where the nucleation rate is possibly too small for the volume of the pore water to freeze within the 16 s residence time of the particles in HINC (David et al., submitted). This causes the significant increase in $S_\mathrm{i}$ required for ice to to nucleate via homogeneous freezing of bulk solution droplets, as can most clearly be seen for both 300 nm and 400 nm particles of the FW200 soot, depicted in Fig. 2c and d, respectively.

The isotherms of both lamp black soots, LB_OEC and LB_RC, resemble a type-III isotherm. Taking the (specific $N_2$) surface area into account, the isotherms are characterized by a small amount of sorbed water vapor at (nearly) saturated conditions of approximately an equivalent monolayer coverage (see Fig. 7b). Such a weak interaction of the water molecules with the soot is also reflected in our ice nucleation results, where we only observe a modest freezing ability on these soots, below $S_\mathrm{i}$ conditions of homogeneous freezing of solution droplets. This suggest a more hydrophobic character, i.e. a high soot-water contact angle, and/or limited availability of pores of the correct size, which would inhibit pore-filling well below water saturation conditions (David et al., in prep.). The lower affinity to water as compared to FW200 indicates a higher soot-water contact angle and is further supported by the relatively lower values of the $C$ parameter (see Table A1). Furthermore, our TGA experiments indicate that these samples are mainly composed of pure carbon, so we can exclude absorption of water by OM for these soots. This is further supported by the absence of a steep increase in $AF$ at the homogeneous freezing line, indicating a lack in the formation

of solution droplets, which has been observed (e.g. by Koehler et al., 2009). However, care must be taken when comparing equivalent monolayer coverages, as they neglect the effect of water being taken up in patches into the porous structure of the soot aggregates. Our DVS runs reveal a slight hysteresis of the lamp black soots (see Fig. 7b) suggesting the presence of some mesopores, but again lie within the uncertainty of approximately $\Delta m = 0.75$ %. The availability of some slit-like

pores was also inferred from TEM images (SI Fig. S18). At the same time our TEM images reveal that both lamp black soots have significantly broader primary particle size distributions with overall larger mean spherules compared to e.g. FW200 as discussed above (see Sect. 3.3). Thus, for a given mobility diameter selected in the DMA, lamp black aggregates are composed of fewer spherules and thus likely contain less cavities formed by sintered spherules compared to soots with smaller spherules, lowering the probability that a sufficient number of pores of the right size (and contact angle) are present (David et al., in prep.).

The presence of some mesopores is further supported by the shape of the water uptake curves of LB_OEC and LB_RC, which both show a non-linear increase for $RH_{\rm w} \geq 90$ %, related to capillary condensation. Thus, we believe that the ice formation associated with LB_OEC and LB_RC is indeed caused by a PCF mechanism, but that the more hydrophobic character of these soots (higher water-soot contact angles) decreases the efficiency of this process compared with the more hydrophilic FW200. This interpretation is consistent with the slightly larger water uptake of the LB_RC compared to the LB_OEC (see Fig. 7b),

resulting in a slightly better ice nucleation ability of the LB_RC compared to the LB_OEC (see Fig. 3).

The water uptake of the FS sample is roughly six times larger than that of the lamp black samples at the highest $RH$, but does not show a strong increase at $RH > 80$ %(see Fig. 7a). We believe that this reduced water uptake of the FS is caused by a lack of mesopores and a prevalence of narrow micropores, which take up water as observed by the mass increase at relatively low $RH_{\rm w}$. Micropore water uptake has been reported for fullerenes (Hantal et al., 2010) and is further supported by

the absence of any hysteresis of the desorption isotherm. Additionally, the presence of micropores on the FS is also supported by the rather large surface area compared to the lamp blacks (see Table A1) and is consistent with Ferry et al. (2002), who found liquid water to exist in micropores of soot particles produced by a kerosene burner down to temperatures of $T = 200$ K. Also the value of the $C$ constant in the classical BET equation, describing the adsorbate-adsorbent interaction strength, was found to be $C = 478$ (see Table A1). Such a high value of C is usually associated with narrow micropores (Thommes

et al., 2015). Nonetheless, water within micropores is not able to nucleate ice via the PCF mechanism, as these pores are too small to allow formation of a critical ice embryo (e.g. Marcolli, 2014). The absence of any sharp increase along the water uptake isotherm of FS at high $RH_{\rm w}$ further suggests that the mass loss during TGA cannot be associated with hygroscopic material responsible for water sorption, as a similar (non linear) increase in water mass would be expected for OM (e.g. Zardini et al., 2008). Altogether, the ice nucleation mechanism of FS remains inconclusive and requires further studies since the results

from TGA and DVS do not give a consistent picture favoring either PCF or homogeneous freezing of solution droplets as the prevailing mechanism for the observed ice nucleation activity of this soot in the cirrus regime. Finally, the water sorption isotherms for the miniCAST samples reveal their general hydrophobic character, consistent with our ice nucleation results. In particular, it is worthwhile to note that the mCAST brown shows a reduced water uptake capacity compared to the mCAST black. This supports our hypothesis from above, that the OM associated with mCAST brown is water-insoluble, suppressing

the ice nucleation activity possibly by filling the mesopores thus blocking water uptake in them. The larger water uptake of the

mCAST black compared to the lamp blacks could result from a larger number of mesopores on the mCAST black aggregates, which are generally composed of more and smaller primary particles compared to the lamp blacks. This is consistent with the slightly larger hysteresis observed for the mCAST black sample compared to the lamp blacks. Mesopores are indeed present on most tested samples (with the exception of FS), as can be inferred from the DVS experiments and are consistent with ice nucleation below homogeneous freezing conditions of solution droplets for large enough soot aggregates (see Fig. 3). Combining our ice nucleation results with data obtained from TGA and DVS, we identify PCF as the dominant mechanism to cause the freezing of the tested soot particles. Nevertheless, FW200 is the only sample out of the six probed soots that shows significant ice nucleation well below conditions for homogeneous freezing of solutions. Thus, the presence of pores (mechanical active sites) is insufficient for PCF, but the sites need to have a hydrophilic character in the form of hydrophilic surface functional groups to initiate interaction of the soot surface with water vapor within a pore, i.e. a sufficiently low contact angle (David et al., in prep.).

## 4 Atmospheric implications

Our results are direct evidence that bare, large hydrophilic soot aggregates can nucleate ice via PCF. To what extent our results are transferable to atmospheric soot is discussed here. The absence of freezing below water saturation for $T > 233$ K suggests that the impact of soot on ice nucleation in MPCs is negligible, at least for unprocessed soot particles. For temperatures above 233 K our results show activation only above water saturation, indicating droplet formation for all of the soot types and sizes tested, confirming the results of Friedman et al. (2011), who found no heterogeneous ice formation at water subsaturated conditions at 233 K.

Our ice nucleation measurements of 400 nm size selected particles demonstrate that soot particles can nucleate ice in the cirrus regime, below $RH$ for the homogeneous freezing of solution droplets, if the particles have the required physicochemical properties. Specifically, our DVS results reveal that both pore size distribution and contact angle distribution determine the ice nucleation ability of the soot particles studied. In the case of extremely hydrophobic soot (very high soot-water contact angles), such as mCAST brown, the lack of ice nucleation below homogeneous freezing conditions of solution droplets, even for the largest particle sizes tested here, suggests that such soot does not play a role in atmospheric ice nucleation, at least for the cirrus temperatures covered by our experiments. However, slightly less hydrophobic soot (lamp blacks), still characterized by high soot-water contact angles, can promote ice nucleation heterogeneouosly for $d_m \geq 200$ nm. This limits the relevance of the majority of freshly emitted soots for atmospheric ice nucleation at cirrus conditions, which are generally believed to be hydrophobic. In contrast, hydrophilic soot (FW200) has the capability to act as effective INP ($AF > 10^{-3}$ for $RH_w \approx 80$ %, see SI Fig. S1h) even at sizes $d_m > 100$ nm, making it potentially important for the anthropogenic forcing on global climate. Nevertheless, most of the combustion aerosols emitted into the atmosphere have significantly smaller diameters usually below 100 nm (e.g. Rose et al., 2006; Kim et al., 2001; Moore et al., 2017) and can represent more complex internal mixtures. Since none of the investigated soot types was ice nucleation active when particles of 100 nm mobility diameter were selected, it is unlikely that such small soot particles with properties similar to those investigated here will act as INP unless they are internally

mixed with other ice active material. Thus, the role of bare small soot particles would be limited to a condensation sink for semivolatile water-soluble species. This finding is important for instance for the fate of soot particles from aviation emissions, which are generally found to be even smaller than $100$ nm in diameter (Moore et al., 2017; Yu et al., 2017). Such particles are often internally mixed with sulfuric acid (Kärcher, 2018) and can contain metallic compounds (Abegglen et al., 2016) or other

residues such as lubrication oil (Yu et al., 2012) and organics (e.g. Yu et al., 2017). These factors can cause atmospheric soot particles to differ in physicochemical properties, e.g. contact angle (surface properties), from the particles types investigated here, which in turn influences their ice nucleation abilities. Still, there is increasing evidence for larger soot aggregates up to $1$ µm in diameter in the atmosphere (Posfai et al., 2003; Chakrabarty et al., 2014), mainly sourced from biomass burning and wildfires. At the same time such solid fuels can also produce ash particles, which can also contribute to ice formation

(Umo et al., 2015; Grawe et al., 2016, 2018). In addition, atmospheric aerosol particles have been reported to contain large black carbon inclusions (up to $1$ µm in diameter, Moffet et al., 2016). In regions of the atmosphere where these large soot particles are found, with the required physicochemical properties, these soot particles will be able to nucleate ice similar to the $400$ nm particles discussed here. Finally, the results reported here only encompass soot particles that activate within the residence time of HINC ($\tau \approx 16$ s) and within a single cloud cycle. At the same time, the atmospheric life time of soot is

around $5 - 7$ days (Jiao et al., 2014; Reddy and Boucher, 2007). During this time the soot aggregates undergo atmospheric processing, encompassing any chemical and/or physical change of the particle properties (e.g. Zhang et al., 2008), for instance through photochemical processes (Li et al., 2018) or by acquiring of a coating due to condensation of semivolatile species or compaction of the soot agglomerate. Such processing can alter the physicochemical properties, such as fractal dimension or hygroscopicity (contact angle, e.g. Wei et al., 2017). Of particular interest here is the cloud processing of soot particles,

i.e. the change in physicochemical properties as the particles are involved in cloud microphysical processes such as cloud droplet or ice crystal formation. In case of ice nucleation via a PCF mechanism, pore ice can remain trapped within the cavities (microscopic pore ice) between subsequent cloud cycles for certain conditions of $T$ and $RH$, even though the macroscopic ice crystal is sublimated. For instance, an ice crystal formed on a soot particle leaving the cloud will experience an ice subsaturated environment and thus sublimate. However, given that $RH$ conditions outside the cloud are high enough (at $T < 273$ K), the

ice within the pores can survive, due to the reduced saturation vapor pressure of ice within the cavity. This pore ice can then grow into macroscopic ice in subsequent cloud cycles when $RH_\text{i} = 100$ % is exceeded. Such a pre-activation effect has been discussed before (D' Albe, 1949; Fukuta, 1966; Mossop, 1956; Marcolli, 2016) and should be addressed in future work, as this may have implications for the enhanced freezing ability of soots in both the cirrus and MPC temperature regime. Moreover, the wettability (contact angle) of soot particles can increase (decrease), as these particles undergo atmospheric processing. Even

though the more hydrophobic soots tested here reveal only a weak ability to nucleate ice via PCF below homogeneous freezing conditions, this likely changes during atmospheric processing and subsequent cloud cycles, with significant implications for the role of soot in atmospheric ice nucleation. It is clear from this laboratory study that the physicochemical properties of soot aerosol determine their ice nucleation potential, with wettability being particularly important. At the same time our conclusions drawn here are limited by using bulk particle properties to explain ice nucleation taking place on individual particles (and at

a molecular level). Further investigation of other factors, especially elaboration of a more quantitative pore size distribution

and detailed chemical characterization of size selected particles, or at least individual particles, would be desirable to increase our understanding of soot ice nucleation abilities. Certainly, atmospheric soot particles can be more complex than the particle types investigated here. At the same time laboratory studies are needed to provide a more fundamental understanding of the properties relevant for ice nucleation.

## 5    Conclusions

The ice nucleation ability of a variety of soot types has been systematically evaluated in controlled laboratory measurements. The soot types investigated cover a wide range of physicochemical properties as proxies of atmospheric soot particles, including different commercially available black carbons such as gas blacks (FW200), lamp blacks (LB_RC and LB_OEC) and fullerene soots (FS), as well as propane flame soot with different OM content derived from a miniCAST burner (mCAST black and mCAST brown). Ice nucleation was probed on dry, size selected aerosol particles, for four different sizes of 100, 200, 300 and 400 nm (mobility diameter), covering a temperature range between 253 and 218 K, using HINC. Ice nucleation activity was investigated in relation to particle morphology deduced from TEM and coupled DMA-CPMA measurements, temperature dependent mass loss obtained from TGA analysis, as well as water vapor uptake capacity derived from DVS experiments.

The results discussed in this paper show that soot particles can contribute to ice formation below homogeneous freezing conditions of solution droplets only if particle diameters exceed $d_{\mathrm{m}} > 100$ nm. One distinct finding of our work is that there is a marked dependence on the HNT for all of the investigated soot types. In the MPC temperature regime no ice nucleation was observed below water saturation, while for some of the probed soot samples ice nucleation was observed below the RH required for homogeneous freezing of solution droplets in the cirrus regime. The absence of heterogeneous freezing in the MPC regime below water saturation suggests that deposition nucleation does not take place on the tested particles. While water can be taken up into the pores of the soot aggregates also at MPC conditions, the absence of any ice formation below water saturation indicates that there is a lack of active sites that could trigger heterogeneous ice nucleation at these temperatures. The observed ice nucleation in the cirrus regime could theoretically be caused through a (surface area dependent) deposition nucleation mechanism. However, the strong dependence of the ice nucleation efficiency on the HNT implies that it is the liquid water within the soot pores that freezes homogeneously, since particle properties considered relevant for deposition nucleation (if present) should be available for ice nucleation in both the cirrus and MPC regime. Such a dependence on the HNT relevant for liquid water freezing for ice nucleation onto soot particles investigated here is in-line with a PCF process and in contrast to classical deposition nucleation, where the liquid water phase is absent. Overall, we conclude that the ice formation process on the soots is best described by a PCF mechanism and not deposition nucleation. Auxiliary measurements performed along with our ice nucleation experiments indicate that physical and chemical properties of combustion aerosol varied markedly and influenced the ice nucleation ability of the particles. The TEM results revealed the presence of cavities and pores on all soot types investigated. Nevertheless, the pore number density likely depends on primary soot spherule size. More importantly, the potential of pores to take up water through capillary condensation strongly depends on their size and the soot-water contact angle, i.e. the hydrophilicity of the soot, as revealed by our TGA and DVS measurements. The presence of hydrophobic matter

on the soot aggregates impedes ice nucleation probably due to blocking of pores. Our TGA, however, is limited to an overall assessment of the presence of heat sensitive matter and conclusions about the chemical nature of this material is not possible and should be investigated in future studies. In addition, the water affinity obtained from the DVS measurements alone cannot be used to draw direct conclusions about the ice nucleation efficiency and/or freezing mechanism, but has to be related to surface area and spherule size. Our TEM evaluation indicates that spherule size influences the availability of pores. Future studies should focus on how atmospheric processes change the hydrophilicity (contact angle) of a given aggregate and spherule size in order to further our understanding of the interplay between pore size and number density and contact angle.

**Appendix A: Auxiliary measurements for soot sample characterization**

**A1    BET surface area**

Particle surface area was determined with the Autosorb-1MP surface area analyzer (Quantachrome, Odelzhausen, Germany) by $N_2$ adsorption using the BET method. Specific surface area values were determined from the linear BET plot of the $N_2$ isotherm up to $p/p° \approx 0.28$, where $p$ is the equilirbium pressure and $p°$ denotes the saturation vapor pressure, assuming a $N_2$ molecular cross sectional area of $\sigma_{N2} = 16.2$ $\mathring{A}^2$, and a minimum of three adsorption points, using the linear form of the BET equation (Thommes et al., 2015):

$$\frac{p/p°}{n(1-p/p°)} = \frac{C-1}{n_m C}(p/p°) + \frac{1}{n_m C}, \tag{A1}$$

where $n$ is the total amount of nitrogen adsorbed on the particle system and $n_m$ is the specific monolayer capacity. The constant $C$, reported in Table A1 for the investigated soots, gives information about the shape of the isotherm (Gregg and Sing, 1982; Thommes et al., 2015). $N_2$ adsorption was conducted at the temperature of liquid $N_2$ ($\approx 77$ K). The primary factor determining the specific surface area is the primary particle size, with smaller primary particle sizes resulting in higher specific surface areas. Prior to determination of specific surface areas, soot samples were outgassed and dried in a desiccator. First, soot samples where outgassed at 323 K until pressure dropped below 300 mbar. Temperature was subsequently increased step-wise to 353 K and the samples were dried for a period of 15 h at this temperature, under nearly vacuum conditions, before the surface area was determined.

**Table A1.** Detailed results of the specific surface area measurements for the soot samples as determined by $N_2$ adsorption following the BET-method. All specific surface area values, $a_{BET,N2}$, are given in units of $[m^2/g]$. The mean values are reported in Table 1. Besides, we report the values of the $C$ parameter used in Eq. A1 for the two independent experiments labeled #1 and #2, respectively.

|  | $a_{BET,N2}$ (#1) | $C$ (#1) | $a_{BET,N2}$ (#2) | $C$ (#2) |
|---|---|---|---|---|
| FW200 | 524.3 | 896.3 | 528.2 | 650.9 |
| LB_OEC | 24.8 | 132.8 | 22.5 | 218.5 |
| FS | 271.6 | 455.8 | 258.3 | 500.0 |
| LB_RC | 23.2 | 135.3 | 22.8 | 181.4 |
| mCAST black | 121.0 | 118.3 | 118.8 | 115.1 |
| mCAST brown | 127.6 | 59.9 | 122.9 | 50.4 |

**A2    Temperature dependent mass loss: Thermogravimetric analysis**

The temperature dependent mass loss of the soot samples was determined using a thermogravimetric analysis. Soot powder was deposited into a platinum crucible and subsequently exposed to three different temperature stages in a Thermogravimetric

Analyzer. During thermogravimetric analysis, mass changes are continuously monitored while the heating program reported in Table A2 was applied to all samples. Time, sample weight, temperature and gas flows are continuously recorded during the heating program. For TGA analysis performed within this work, a sample purge flow of $30$ mLmin$^{-1}$ and a balance purge flow of $40$ mLmin$^{-1}$ was used. $N_2$ was used for both flows so that the sample only reacts to temperature due to thermal decomposition. For every experiment an equilibration time of approximately $1-2$ min was used before starting the TGA program. This is done in order to allow the (sample-) pan to come to complete rest, as any swing movement of the pan can cause inaccurate mass measurements (Lapuerta et al., 2007). Lapuerta et al. (2007) reported a non-linear increase in loss of material with increasing heating rates, but found the difference to be below $10$ % when heating rates were constrained to a maximum of $20$ °Cmin$^{-1}$. Hence, our samples should not be affected, by using the maximum heating rate of $20$ °Cmin$-1$ (see Table A2).

**Table A2.** Heating program used for thermogravimetric analysis of soots in a pure $N_2$ atmosphere.

| Step | Specification |
|------|---------------|
| 1 | Isothermal at $T = 30$ °Cmin$^{-1}$ for $5$ min |
| 2 | Ramp $15$ °Cmin$^{-1}$ to $T = 350$ °C |
| 3 | Isothermal at $T = 350$ °C for $30$ min |
| 4 | Ramp $15$ °Cmin$^{-1}$ to $T = 650$ °C |
| 5 | Isothermal at $T = 650$ °C for $15$ min |
| 6 | Ramp $20$ °Cmin$^{-1}$ to $T = 1000$ °C |
| 7 | Isothermal at $T = 1000$ °C for $10$ min |
| 8 | Ramp $30$ °Cmin$^{-1}$ to $T = 30$ °C |

**Table A3.** Relative mass loss as determined by TGA analysis in a pure $N_2$ atmosphere. Mass loss (ML) for soot samples is expressed as cumulative, relative weight loss at a given reference temperature, and are given as percentage mass loss from initial weight. Reported values indicate mean values of two TGA experiments performed for a given soot type.

| | $ML_{100}$ °C | $ML_{150}$ °C | $ML_{200}$ °C | $ML_{250}$ °C | $ML_{300}$ °C | $ML_{350}$ °C | $ML_{650}$ °C | $ML_{950}$ °C |
|---|---|---|---|---|---|---|---|---|
| LB_OEC | 0.10 | 0.16 | 0.25 | 0.36 | 0.44 | 0.53 | 0.68 | 0.97 |
| FS | 1.85 | 2.27 | 2.60 | 2.92 | 3.24 | 3.97 | 22.95 | 33.37 |
| FW200 | 6.88 | 7.64 | 8.04 | 8.52 | 9.35 | 10.67 | 22.66 | 31.43 |
| LB_RC | 0.090 | 0.16 | 0.24 | 0.41 | 0.57 | 0.75 | 0.93 | 2.80 |
| mCAST black | 0.94 | 1.18 | 1.34 | 1.44 | 1.48 | 1.48 | 5.27 | 15.62 |
| mCAST brown | 0.55 | 1.61 | 3.39 | 5.71 | 7.52 | 10.25 | 15.68 | 18.58 |

## A3  Water uptake and loss: Dynamic Vapor Sorption

During DVS experiments, the sample is constantly weighed (against a reference) as the sample is exposed to changing $RH$ in so-called $RH_w$-scans. Samples in powder form are placed in a metal pan to prevent static electrical loading and the total gas flow ($H_2O$ and $N_2$) was held at a constant flow rate of $200$ smLmin$^{-1}$ throughout the experiment. $RH_w$ is automatically
scanned in predefined steps and the sample mass at quasi-equilibrium conditions is measured whenever the temporal change in sample weight falls below a defined threshold. Since a *constant* mass is not reached during DVS, we refer to these conditions as *quasi*-equilibrium. The time needed to reach quasi-equilibrium conditions is dependent on the slow water adsorption kinetics (Popovitcheva et al., 2000) and is proportional to the surface area available for uptake. Here we considered a mass change rate at/or below $0.0005$ %min$^{-1}$ to correspond to quasi-equilibrium conditions, for an approximate sample weight of $10$ mg. Each
soot sample was dried at $298$ K for a period of $1000$ min prior to the measurement of a sorption isotherm, in order to remove any sorbed water vapor, by flushing the soot samples with only $N_2$ at atmospheric pressure. DVS runs were performed at $T = 298$ K. Mass change was monitored in intervals of $5$ % for $0$ % $< RH_w < 30$ %, at $40$ %, $60$ % and $80$ % in the medium $RH_w$ range and in steps of $3$ % between $80$ % and $98$ % to investigate the presence of mesopores. Note that the FW200 was only probed in intervals of $20$ % below $RH_w = 80$ %. Each soot type was probed for two independent experiments in the
DVS and good agreement was found. Exceptions are the mCAST samples, which were only probed once in the DVS. The desorption was probed at the same $RH_w$ values so that a hysteresis could be derived. Similar to the TGA, no out gassing at elevated temperature or high vacuum was performed, in order to preserve the original surface properties.

## A4  Soot aggregate morphology and effective density: DMA-CPMA measurements

The Centrifugal Particle Mass Analyser uses two nested, rotating, charged metal cylinders between which injected particles
can be selected based on mass. The two opposing electric and centrifugal forces, arising from the potential difference and the rotation, respectively, are used to classify particles by mass to electrical charge ratio ($m/q$). Details on its operation may be found in Olfert and Collings (2005). In conjunction with a DMA, density and mass mobility exponent ($D_{fm}$) of aerosol particles can be determined. For this purpose, DMA-CPMA are coupled in series in such a way that aerosol particles are first sent into the DMA, classifying the particles by electrical mobility (selecting by drag:charge), resulting in a narrow size distribution. It
should be noted that this distribution contains predominantly singly charged particles of the mobility diameter chosen, but is not perfectly monodisperse, with some fraction of larger, multiply charged particles also being present (see SI Fig. S15). The particles are subsequently passed through the CPMA, selecting monodisperse aerosol by mass to charge, and counted by a CPC (Model 3776, TSI Inc.) downstream of the CPMA, operated at a flow rate of $0.3$ Lmin$^{-1}$, yielding the number concentration of size and mass selected aerosol. For a given mobility size, the CPMA is operated in scanning mode, measuring the number
concentration of particles at discrete mass set points. Due to the fractal like morphology of soot aggregates a range of different masses can correspond with a selected mobility diameter, such that soot particles with a more uniform morphology would have a narrower mass distribution (Dickau et al., 2016). The DMA-CPMA output function can be used to derive aerosol properties, such as the fractal dimension and the effective density. The effective density ($\rho_{eff}$, e.g. Dickau et al., 2016; Olfert et al., 2017;

**Table A4.** Mass-mobility pre-factor, $C$, and fractal dimension, $D_{\text{fm}}$, derived from power-law fits of the form of Eq. (A4) to mass-mobility data shown in Fig. 4a. A $D_{\text{fm}}$ value of 1 corresponds to a straight chain-like structure, whereas $D_{\text{fm}} = 3$ indicates a compact sphere like structure. The value in brackets indicates the standard error.

|  | $C$ [kgm$^{-D_{\text{fm}}}$] | $D_{\text{fm}}$ [-] |
|---|---|---|
| FW200 | 0.02 (0.2) | 2.35 (0.07) |
| FS | 0.17 (0.19) | 2.50 (0.07) |
| LB_OEC | 3.65 (4.6) | 2.64 (0.09) |
| LB_RC | 0.07 (0.12) | 2.38 (0.13) |
| mCAST black | 0.01 ($3.8e^{-6}$) | 1.86 (0.05) |
| mCAST brown | 0.01 (0.01) | 2.31 (0.07) |

Abegglen et al., 2015) is defined as the density particles would have if they were spherical, i.e. by the ratio of particle mass to particle spherical equivalent volume based on the selected mobility diameter in the DMA, $d_{\text{m}}$, given by (McMurry et al., 2002):

$$\rho_{eff} = \frac{6}{\pi} \frac{m_p}{d_{\text{m}}^3}. \tag{A2}$$

5 Here, $m_p$ denotes the median mass obtained by fitting the mean mass spectral number density distribution from the CPMA scan of the DMA-classified particles with a log normal distribution. Typically $\rho_{eff}$ is expressed as a power law fit of the form:

$$\rho_{eff} = C d_{\text{m}}^{(D_{\text{fm}}-3)}, \tag{A3}$$

where $D_{\text{fm}}$ is the mass-mobility exponent and $C$ is a constant pre-factor with units of kgm$^{-D_{\text{fm}}}$. The fractal dimension $D_{\text{fm}}$ is a useful quantity to describe aerosol properties. $D_{\text{fm}}$ can be inferred from the power-law relationships relating particle mobility 10 and mass (Schmidt-Ott, 1988; Schmidt-Ott et al., 1990):

$$m_p = C d_{\text{m}}^{D_{\text{fm}}}, \tag{A4}$$

where $C$ is a constant called the mass-mobility pre-factor. Here, the fractal dimension of the soot particles was obtained by performing a least-square fit, using a power-law relationship of the form of Eq. (A4) applied to the mass-mobility data, where the median mass was used.

15 **Appendix B: miniCAST**

Within the miniCAST, the combustion process is interrupted by quenching the flame with $N_2$. While the quenching of the flame within the burner takes place at a fixed height, the flame size can be varied by changing the absolute flow of fuel gas and oxidation air, holding the C:O ratio, i.e. the flame chemistry constant. At the same time, changing the ratio of the fuel gas and

the oxidation air, or premixing the fuel gas with $N_2$ allows for a controlled adjustment of the C:O ratio. For a high absolute flow of fuel and oxidation air, the flame is large and thus is quenched relatively low within the flame. On the other hand, a small absolute flow of fuel and oxidation air causes the quenching to take place close to the flame tip. We aimed at comparing the ice nucleation abilities of combustion generated soot by varying organic carbon mass. We therefore altered both, the position of
flame quenching and the ratio of fuel to oxidation air. The miniCAST was operated at two different test points (TP). We use the fuel-air flame equivalence ratio, $\Phi$, its inverse, the air-fuel ratio, $\lambda$, and the atomic carbon-to-oxygen ratio, C:O, to characterize the flame chemistry at the two TP.

The flame C:O ratio can be calculated based on the fuel and air densities at normal temperature and pressure (NTP) according to:

$$C:O = \frac{Q_f 10^{-3}}{Q_{ox} 10^{-3}} \cdot \frac{\rho_{f,NTP} F_{Cinfuel}}{\rho_{ox,NTP} F_{O2inair}}, \tag{B1}$$

where $Q_f$ and $Q_{ox}$ denote the fuel and oxidation air flow rates, respectively (see Table B1), $F_{Cinfuel}$ denotes the C-fraction in $C_3H_8$ (0.818), $F_{O2inair}$ the fraction of $O_2$ in air (0.232) and $\rho_{f,NTP}$ and $\rho_{ox,NTP}$ denote the mass densities of $C_3H_8$ (1.882 kgmol$^{-1}$) and air (1.205 kgmol$^{-1}$).

Assuming the combustion reaction to be:

$$C_3H_8 + 5O_2 \rightarrow 3CO_2 + 4H_2O, \tag{BR1}$$

we calculate $\Phi$, as fuel-air mass ratio divided by the stoichiometric fuel-air ratio, given by:

$$\Phi = \frac{(m_f/m_{ox})}{(m_f/m_{ox})_{st}} = \frac{1}{\lambda}. \tag{B2}$$

Here $m_f$ denotes the mass flow of $C_3H_8$, $m_{ox}$ the mass flow of oxidation air and the subscript $st$, denotes the stoichiometric fuel-air ratio. Using reaction BR1 and a mass percentage of $O_2$ of 23.2 % the stoichiometric fuel-air ratio is given by:

$$(m_f/m_{ox})_{st} = \frac{44.096\,\text{gmol}^{-1}}{5 \cdot 31.998\,\text{gmol}^{-1}} \cdot \frac{23.2}{100} = 0.0639. \tag{B3}$$

Thus the combustion process is *fuel rich* for $\Phi > 1$ and $\lambda < 1$ and *fuel lean* combustion occurs for $\Phi < 1$ and $\lambda > 1$, where *lean* refers to conditions where fuel is the limiting reactant.

It should be noted that the parameters described above characterizing the flame chemistry, allow only for limited comparison with other studies, since they do not include additional factors such as the $N_2$ quench gas flow rate, $Q_{N2}$, the $N_2$ flow rate
premixed to the fuel, $Q_{N2,mix}$, the dilution air flow $Q_{dil}$ or any differences in burner geometries (Durdina et al., 2016), which influence the soot characteristics. We use these parameters to characterize our miniCAST soot samples termed mCAST black and mCAST brown, as shown in Table B1.

## Appendix C:  TEM sample preparation and evaluation

We used TEM to visualize the dimensions and morphology of the soot particles. Size selected soot particles were collected
on standard Cu-TEM grids having 400 mesh with a single coated continuous carbon film (Quantifoil Micro Tools GmbH,

**Table B1.** Summary of the settings for miniCAST Series 4200, used to produce mCAST black and mCAST brown presented in this study. $Q_f$ denotes the volumetric $C_3H_8$ (fuel) flow[c], $Q_{ox}$ the oxidation air[a] flow, $Q_{N2,mix}$ the $N_2$[b] flow premixed with the fuel, $Q_{N2}$ the quench $N_2$[b] and $Q_{dil}$ the dilution air[a] flow, respectively. All flows are given in units of standard $[Lmin^{-1}]$. $\lambda$, $\Phi$ and C:O indicate the air-fuel equivalence ratio, the fuel-air ratio and the atomic carbon:oxygen ratio, respectively, as described in the text.

| Soot type | $Q_f$ | $Q_{ox}$ | $Q_{N2,mix}$ | $Q_{N2}$ | $Q_{dil}$ | $\lambda$ | $\Phi$ | C:O |
|---|---|---|---|---|---|---|---|---|
| mCAST black | 0.06 | 1.55 | 0.0 | 7.5 | 20.0 | 1.0576 | 0.9455 | 0.2132 |
| mCAST brown | 0.06 | 1.42 | 0.25 | 7.5 | 20.0 | 0.9689 | 1.0321 | 0.2327 |

[a] VOC filtered (wall) synthetic air, [b] Carbagas 5.0 grade, [c] Carbagas quality 25

Großlöbichau, Germany), using the Zurich Electron Microscope Impactor (ZEMI). ZEMI is a semi-automated, custom built rotating drum impactor, holding a total of 24 TEM grids. The aerosol containing air stream is directed onto the TEM grid. Soot particles with inertia are impacted onto the TEM grid, while the surrounding gas molecules are laterally sucked away. A step motor (R208 Microstepping Driver, RMS Technologies, Carson City, USA), controlled over a LabView program allows to

adjust the time period that each grid is exposed to the aerosol flow and the to control the sampling flow rate using a MFC (G-Series, MKS Instruments, Andover, USA). The distance between the exit of the nozzle and the impaction plate (TEM grid) can mechanically be adjusted. Here, we used a constant flow rate of $1$ $Lmin^{-1}$ and varied the sampling time between $100 - 600$ s, depending on the number concentration of the size selected aerosol, in order to ensure a high enough loading of the TEM grid. Images of the aggregates were obtained from a JEOL TEM-1400+ (JEOL Ltd., Tokyo, Japan) with a LaB6 filament, operating

at $120$ kV. Finally, particle morphology and primary particle size were obtained upon image analysis using the MATLAB (R2017b, MathWorks Inc., Natick, USA) image analysis tool box. Therefore, a custom developed program in MATLAB was used which reads in the intensity images obtained from TEM. The program then allows to select regions of interest (ROIs), where individual primary spherules were sized using MATLAB's *imellipse* function, using the average of the minor and major axis describing the ellipse. For each aggregate a minimum of 3 different ROIs were selected to ensure that more weight was not

given to certain regions of the soot aggregates under investigation. A minmimum of around 100 primary particles were sized for each soot type.

## Appendix D: Horizontal Ice Nucleation Chamber

### D1 Uncertainty in temperature and relative humidity

The main uncertainty of the ice nucleation measurements results from uncertainties in temperature control and flow changes that alter the ratio of aerosol to sheath flow ratio, resulting in variations of the humidity profile across the aerosol lamina (located at the chamber center). We report uncertainties in $S_w/S_i$ arising from the variability of humidities across the aerosol lamina and temperature uncertainty, thus capturing the possible range of humidity conditions experienced by any aerosol particles that are confined within the lamina. We first calculate the aerosol lamina coordinates and in a second step the supersaturation at these coordinates. Due to the horizontal orientation of the chamber walls and the absence of any internal buoyancy (convection), the velocity profile is simply given by (Rogers, 1988):

$$v(z) = \frac{3}{2}\bar{v}[1 - \frac{z^2}{(d/2)^2}], \tag{D1}$$

where $z$ denotes the vertical height coordinate, $d/2 = 1$ cm the half width between the cold and warm walls and $\bar{v}$ the mean flow velocity given by the ratio of the total volumetric flow rate, $Q_{tot} = \int_{z=0}^{z=d} v(z)dz$, and the cross-sectional area of the chamber. Assuming that $50$ % of the volumetric aerosol flow is on either side of the chamber center line (position of aerosol injector), the coordinates of the lamina limits, $a$ and $b$, are found by integrating the flow profile symmetrically around the center line until the integrated flow matches the given aerosol flow. Hence, $a$ and $b$ depend on the *ratio* of aerosol to sheath flow ($F_{AP}/F_{sheath}$).

The temperature at any location $z$ within the chamber can be calculated from the linear temperature profile Rogers (1988):

$$T(z, \Delta T) = T_c + \frac{\Delta T}{D}z, \tag{D2}$$

where $T_c$ and $T_w$ denote the temperature of the cold and warm plates, respectively, and $\Delta T = \frac{T_w - T_c}{d}$. Similarly, the linear profile of vapor pressure with respect to ice, $e_i$ is given by:

$$e_i(z) = e_{s,i}(T_c) + \frac{e_{s,i}(T_w) - e_{s,i}(T_c)}{d}z, \tag{D3}$$

where $e_{s,i}$ denotes the saturation vapor pressure with respect to ice after Murphy and Koop (2005).

Overall, the range of saturation ratio experienced by particles in the aerosol lamina is a function of the ratio of aerosol to sheath flow, $F_{AP}/F_{sheath}$, center temperature, $T_{center}$, and temperature difference between the two walls, $\Delta T$, i.e.:

$$\Delta S_{x,lam} = \Delta S_{x,lam}(F_{AP}/F_{sheath}, T_{center}, \Delta T), \tag{D4}$$

where the subscript $x$ denotes evaulation with respect to water ($S_w$) or with respect to ice ($S_i$). For a given $F_{AP}/F_{sheath}$ and $T_{center}$, activation mostly took place for high $RH_w$, i.e. large $\Delta T$ values. Large values of $\Delta T$ (i.e. large values of $RH$) also result in larger uncertainties in $RH$, thus for a variety of $F_{AP}/F_{sheath}$ and $T_{center}$, we report the calculated uncertainty and variation in $S_{x,lam}$ for a $S_w$,center value of $1.05$ as an upper limit of RH uncertainty.

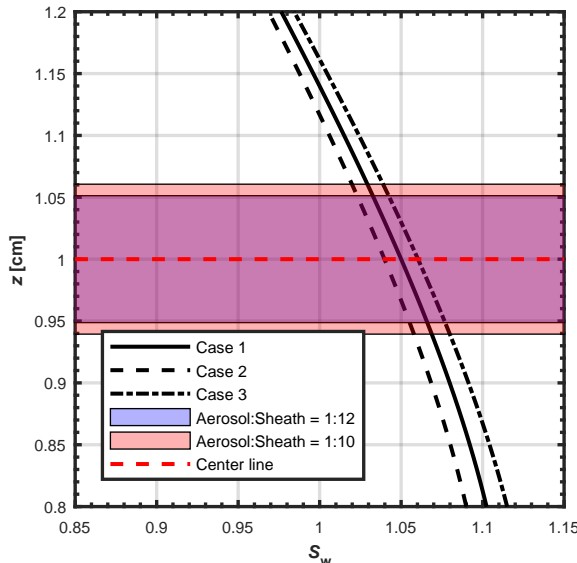

**Figure D1.** Example of $S_w$ profile for the mid-portion of the vertical distance between the cold (at $z = 0$ cm, bottom) and warm (at $z = 2$ cm, top) wall of the HINC chamber, illustrating the variations in $S_w$ across the aerosol lamina (black solid line) for exemplary conditions of $T_{center} = 218$ K, $S_{w,center} = 1.05$ and different $F_{AP}/F_{sheath}$ flow ratios (red and blue shading). The $S_w$ profiles given by the dashed and dash-dotted lines take into into account uncertainty in temperature control arising from the thermocouples. The red dashed line indicates the position where particles are injected into HINC.

The temperatures can ultimately only be controlled within the uncertainty range of the thermocouples used for temperature control and monitoring, having an uncertainty of $\pm 0.1$ K. Hence, for given conditions of $F_{AP}/F_{sheath}$, $T_{center}$ and $S_{w,center}$, $\Delta S_{x,lam}(F_{AP}/F_{sheath}, T_{center}, \Delta T)$ was calculated for three different cases, covering the possible range of thermocouple uncertainties, but all leading to the same center conditions ($T_{center}$, $S_{w,center}$), namely:

- Case 1: $T_w = T_{w,set}$, $T_c = T_{c,set}$

- Case 2: $T_w = T_{w,set} + 0.1$, $T_c = T_{c,set} - 0.1$

- Case 3: $T_w = T_{w,set} - 0.1$, $T_c = T_{c,set} + 0.1$

The variation and uncertainty of the saturation profile across the aerosol lamina for these three cases is shown exemplarily in Fig. D1 for $T_{center} = 218$ K, $S_{w,center} = 1.05$ and different ratios of $F_{AP}/F_{sheath}$. In this study we use a flow ratio
10   $F_{AP}/F_{sheath} = 1 : 10$ to calculate our aerosol lamina boundaries and uncertainties associated with our ice nucleation results. Figure D2 finally illustrates the effect of flow ratio, i.e. aerosol lamina coordinates, on the overall uncertainty in $RH$ across the aerosol lamina.

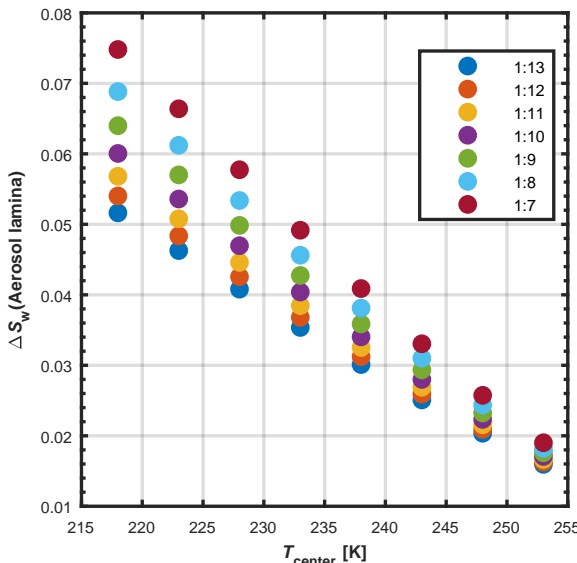

**Figure D2.** Relative humidity variation and uncertainty across the aerosol lamina, $\Delta S_{\text{w,lam}}$ as a function of $T_{center}$ for different *ratios* of $F_{AP}/F_{sheath}$ within HINC. Each reported value denotes the maximum spread in $S_w$ across the aerosol lamina considering all three cases of TC-temperature uncertainty (see text) and a center $S_w$ value of 1.05. The work presented here used flow ratios between 1:10 and 1:12.

## D2 Hydrometeor growth and settling

Growth of ice crystals and cloud droplets is a function of the $T$ and $RH$ conditions within the chamber as well as the residence time of the particles within HINC. The final size the hydrometeors grow to, can be estimated using theoretical diffusional growth equations. In Figs. D3 and D4 we show examples of expected diffusional hydrometeor growth (orange lines) and the
correspondning distance from gravitational settling (blue lines) for ice crystals (solid lines) and cloud droplets (dashed lines) within HINC for cirrus and MPC temperatures, respectively. The examples shown assume an initial hydrometeor diameter of 200 nm and the complete residence time to be available for diffusional growth, i.e. neglecting any time dependence on ice nucleation, thus likely overestimating hydrometeor growth. Diffusional growth was calculated according to Rogers and Yau (1989). The temperature dependency of the water vapor diffusion coefficient in air was taken from Hall and Pruppacher (1976)
and that of the thermal conductivity coefficient from Beard and Pruppacher (1971). Finally, the temperature dependencies of the latent heat of condensation and sublimation were parametrized according to Murphy and Koop (2005). For simplicity we assumed spherical ice crystals, where the capacitance is equal to the ice crystal radius, and calculated growth for a mass accommodation coefficient, $\alpha$, of 0.1 (Skrotzki et al., 2013; Magee et al., 2006). Diffusional growth was calculated in accordance to the vertical profiles of $T$ and $RH$ between the chamber walls. Therefore the (vertical) hydrometeor position was
calculated at time steps of 0.01 s, again assuming particle sphericity, and the $T$ and $RH$ conditions at this (vertical) position were used to calculate the diffusional growth. All hydrometeors were assumed to start growing and falling from the center line

of HINC ($z = 1$ cm), i.e. neglecting an aerosol layer thickness, and the vertical distance covered per time step was calculated using a sedimentation velocity derived from a balance of gravitational and drag force (Lohmann et al., 2016), where the factor $C_{\mathrm{D}}Re/24$ was approximated as 1 justified for particles $r < 30$ μm (Rogers and Yau, 1989). For growth and settling we used the temperature dependent density of supercooled liquid water given in Marcolli (2016) and of ice given in Pruppacher and Klett

(1997). The vertical settling distance experienced due to gravity by the ice crystals and cloud droplets (blue lines), indicate potential loss of the hydrometeors. Particle impaction on the bottom wall is indicated by the blue lines (vertical hydrometeor position) reaching a value of 1 cm, as indicated by the horizontal, gray solid line. Calculation of growth and gravitational settling are aborted if a cloud particle has settled 1 cm. From the theoretical growth curves in Figs. D3 and D4 it becomes evident that cloud particles are not lost due to gravitational settling during the residence time used herein ($\tau = 16$ s), even for

high humidities of $RH_{\mathrm{w}} = 101$ %, with the exception of the 253 K case, when assuming a mass accommodation coefficient of $\alpha = 0.1$ (magenta lines).

While ice crystal impaction onto the bottom plate of HINC could explain an $AF < 1$ as observed for most of our experiments (see SI S2) we would expect a peak in the $AF$ curves, followed by a gradual decrease in $AF$ due to gravitation loss of the particles, as the $RH$ is increased within the experiment. From the absence of any such signal in our soot $AF$ curves we

conclude that our calculations represent an upper limit of settling and diffusional growth values. This can be explained by a potential high time dependency of the soot aerosols to nucleate ice crystals or due to an underestimation of the time needed for the aerosols to equilibrate to the chamber conditions, with both processes reducing the diffusional growth time available for particles within HINC.

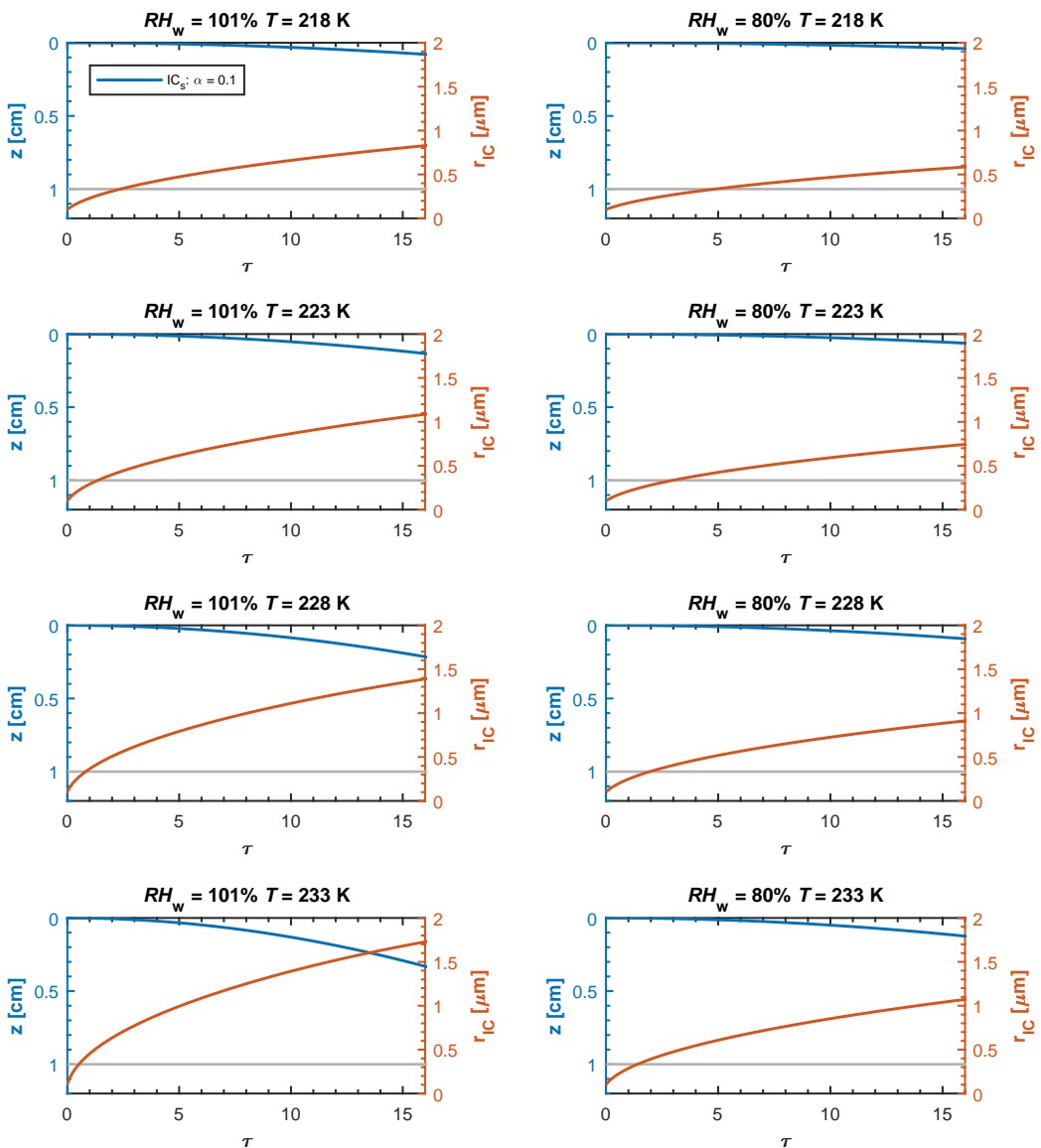

**Figure D3.** Expected diffusional growth of spherical ice crystals, having an initial *diameter* of 200 nm, within HINC for different cirrus regime temperatures, $T_{center}$, (rows) and $RH_w$ along the centerline of HINC (columns), as a function of residence time. Orange lines indicate the ice crystal radius (right hand ordinate). Blue lines indicate the vertical distance covered by the ice crystal (left hand ordinate). Calculations are aborted, once a particle has traveled the half-distance of the two copper plates (1 cm), i.e. has hit the bottom plate of HINC, indicated by the horizontal, gray line (left hand ordinate), i.e. wherever any blue line crosses the horizontal gray line.

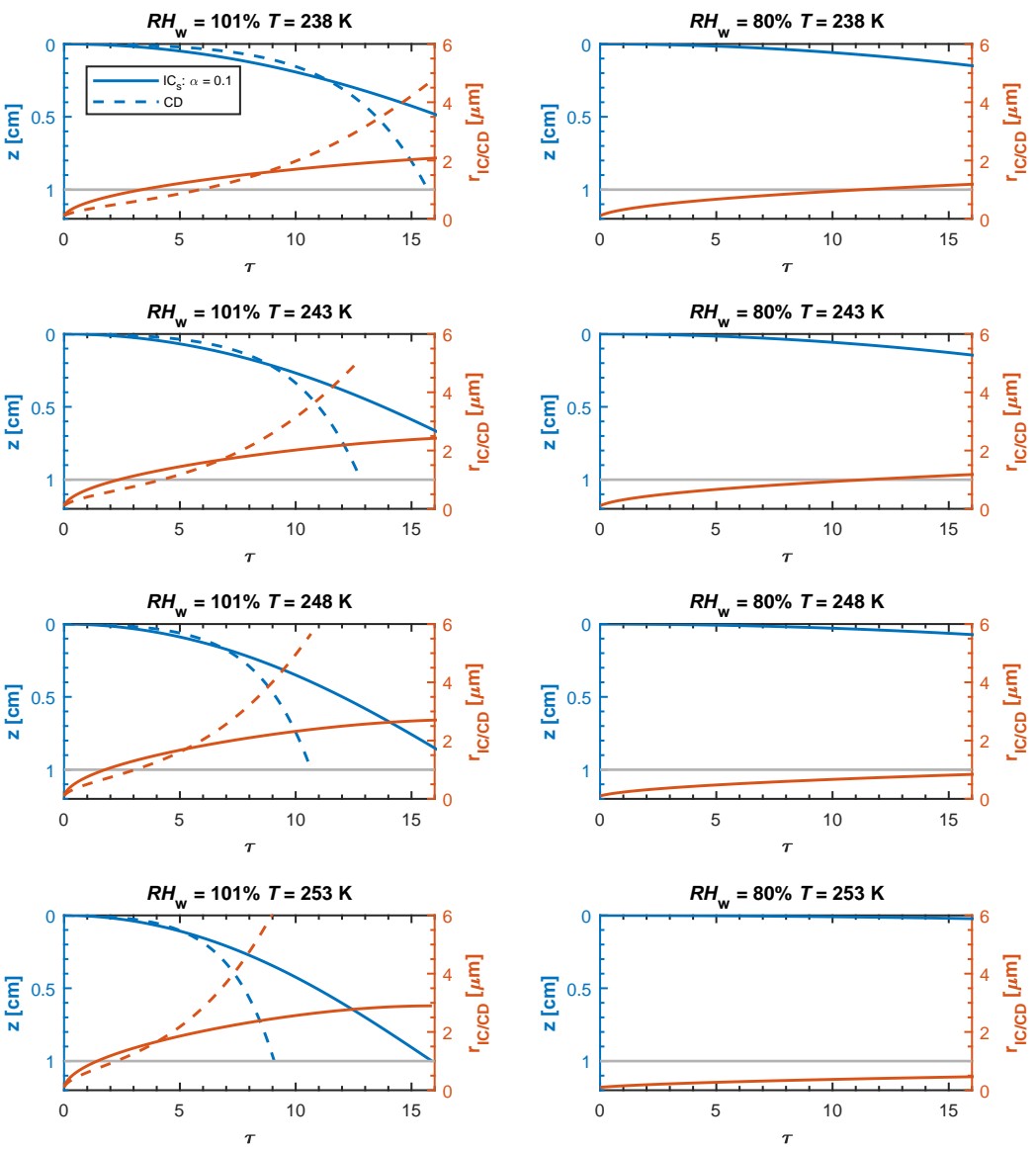

**Figure D4.** Expected diffusional growth and gravitational settling of spherical ice crystals (solid) and cloud droplets (dashed) for MPC temperatures, having an initial *diameter* of 200 nm. Symbols and reference lines as in Fig. D3.

*Author contributions.* FM prepared the manuscript with contributions from ZAK, CM, ROD, UL, EBM and PG. FM and ZAK designed the experiment. FM conducted and analyzed HINC measurements and prepared all figures of the manuscript. PG ran DVS experiments and interpreted data. EBM took TEM images and helped with interpretation. FM, CM, ROD, UL and ZAK interpreted ice nucleation data. ZAK conceived the idea and supervised the overall project.

5   *Competing interests.* The authors declare that they have no conflict of interest.

*Acknowledgements.* F. Mahrt and Z. A. Kanji acknowledge funding from ETH Grant ETH-25-15-1 that supported this work. The authors gratefully acknowledge the great technical support of H. Wydler from the Institute for Atmosphere and Climate Science in overseeing the maintenance of HINC, improving both the CFDC and associated software as well as with help in developing ZEMI. We also thank M. Vecellio and P. Isler from the Institute for Atmosphere and Climate Science for machining parts of our set up. The authors would like to

10   thank J.C.H. Wong from the Laboratory of Composite Materials and Adaptive Structures for her training and help with the TGA and A. Röthlisberger, M. Rothaupt and M. Plötze from the Institute for Geotechnical Engineering for help with BET. The authors are grateful to I. Burgert from the Institute of Structural Engineering for providing access to the DVS facility. We also thank U. Krieger and S. Brunamonti from the Institute for Atmosphere and Climate Science for providing some of the soot samples and L. Lacher (KIT) and J. Atkinson for many helpful discussions on HINC and associated ice nucleation experiments. We thank J. Aerni for help with preparation of the TEM samples.

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
