# Peer review of "Ice nucleation abilities of soot particles determined with the Horizontal Ice Nucleation Chamber"

_Atmospheric Chemistry and Physics, 2018_

## Referee Comment (RC1) · Anonymous Referee #1 · 11 Jul 2018

Review for "Ice nucleation abilities of soot particles determined with the Horizontal Ice Nucleation Chamber" by F. Mahrt et al., submitted to ACPD

In the manuscript, the ice nucleation activity of 6 different soot samples is examined with the CFDC-type ice nucleation chamber HINC at the ETH (Zurich). Additionally, a thorough analysis of physical properties of the soot particles is presented. All data are jointly discussed. It is concluded that the examined soot particles do not act as ice nucleating particles in the mixed cloud regime. In the cirrus cloud regime, some soot types were more ice active than others, which was traced back to pore condensation and freezing.

The manuscript is well written. Data is nicely presented. The overall message is clear. Sometimes I felt that the information was a bit too much, but then it is all interesting and

well researched, so I will not suggest so shorten anything. Below, I give a number of (mostly small) remarks / suggestions for improvements. The number of remarks makes me choose "major revisions", but it should all be easily done. Overall I recommend this manuscript for publication in ACP once these suggestions will have been addressed.

Specific comments:

page 2, line 8: Vali et al. (2015) do not introduce any research or their own, but rather summarize knowledge, thus this citation here is a bit misleading. They don't show that soot particles can act as INPs. I suggest removing this citation here.

page 4, line 25: You claim that your sample "LB_RC" is "directly comparable to the lamp black soot purchased from OEC, due to their very similar physical properties" – however, the BET surface areas of these two soot samples is an order of magnitude apart, so I suggest to delete this statement about their comparability, or adjust it accordingly.

page 5, Table 1: It would be interesting to know how many particles were counted in the TEM analysis. This could be given in an additional line.

page 6, line 8: What were these filters used for? It would be nice to get this information here in the text.

page 6, line 10: Does this mean that the mini-CAST samples on the filters were sampled BEFORE this additional coagulation took place? Please give the answer to this question in the text.

page 6, line 18 and 19: You use HINCNE and HINCNE (i.e., once is in italic, the other isn't). Be consistent throughout the text. My personal preference is the one with italic letters.

page 7, line 7-9: It is not so clear to me if you really can assure that the aerosol flow through the DMA is stable – did you check the exhaust air from the mixing chamber regularly? Was it constant? And were the flows in the DMA stable?

page 7, line 22: The phrase "Cloud particles can be formed" is not correct. Ice may or may not form, as you say, and water droplets will only form at RH_w > 100% and therewith at much higher humidities. Therefore, it would be better to say "Ice crystals may be formed".

page 8, line 1-2: "The residence time is the sum of a nucleation and water droplet or ice crystal growth time in the chamber." – This is strangely formulated. Please reformulate. Maybe "The residence time is the sum of the times . . . "

page 8, line 14-15: How long does it take, until the particles reach the destined temperature after entering HINC? If both sheath and aerosol flow are at laboratory temperature upon entering HINC, there will certainly be a non-negligible delay. This needs to be mentioned and shortly discussed. Also, on the opposite side, could settling of the particles to the lower plate be an issue?

page 8, line 31: Certainly you mean for sizes from 1 ïA■m onwards, and not only in the 1 ïA■m channel? Correct this.

Chapter 2.4: This chapter is an interesting and impressive addition to the ice nucleation measurements - but it would be good to know which of these measurements were performed on the bulk samples and which on the size segregated particles, and for the latter, if then the same set-up described above was used. Please add information on this to the text.

page 9, line 13: Better replace "aid" by "contributed to".

Figure 2: The "blue dashed line" – is that the one that is almost at the RH_w = 1 - line? This is almost impossible to see - use a different color, e.g., red, or small circles or something else.

page 11, line 11-13: Would this then not mean that every possible ice active material would be outcompeted by homogeneous freezing? Still, heterogeneous freezing is considered to play a role even for the cirrus cloud regime. And if this (second) part of

the sentence can be debated, I wonder if the first part (about 100 nm particles being not relevant) is not a trade of between the HINC detection limit and the abundance of these particles in the atmosphere?

page 11, line 25: Missing space between y and 2.

page 11, line 32-33: Another reason could be that it takes up water better – could be worthwhile mentioning.

page 12, line 4: A new sentence should start between "particles" and "they".

page 14, line 7: Add a "," between "increases" and "the" (it took me a while until I got the sentence).

page 14, line 10: Tiny remark, but why "in prep.b" first (before "a")?

page 15, line 7: There is a space too much before the first letter.

Figure 6: One of the screens I used while working at this review did not show this graph well. The "yellow" color shading looked rather pinkish, and the shading of the two other panels was almost the same - maybe there is a better (or additional) way to refer to the panels? (Using a), b) and c) for example?)

page 17, line 19-20: Are you really referring to S16 in the SI, here, so should this rather be S10? But in any case, I have difficulties following your argument – why does the lack of a change in the slope render this interpretation unlikely?

page 17, line 33: Change "of" to "the"

Figure 7: If you used a log-scale for one of the two panels, it would be easier to see the hysteresis and the absolute values for the curves at the lower end. The way it is now, both panels look quite similar, anyway.

page 19, line 21: Are citric acid and glutaric acid organics typical for the atmosphere? Any citation on that? Others might play a more important role (oxalic acid, formic acid,

acetic acid, succinic acid, ...). But there must be a reason why you choose those. Explain that in a few words.

page 20, line 18-21: The last few words "rather than the bulk aggregate size" puzzled me a bit. There is a difference in the freezing curves for FW200 particles of different sizes, and you say, quite correctly, that you observe 1% of particles ice active at similar RHs for a given particle size (BTW: exchange "aerosol size" with "particle size" here). But then why would you refer to "bulk aggregate size" in the end. Would anyone really assume that this plays role? I suggest to delete these few last words unless you feel they are needed.

page 21, line 7: I suggest to tune down "confidently identify" a little.

page 22, line 22: Concerning the use of the word "contact angle", I realize that you talk about the interaction between the soot surface and water vapor. But on the other hand, it has been shown for other materials that it never only is one contact angle describing the ice nucleation behavior of one substance, but a contact angle distribution instead. This might need to be mentioned.

page 23, line 5: It should not be forgotten that solid fuels (from either biomass burning or wildfires) also produce ashes, so for observed atmospheric ice activities of respective particles, also these ashes could be responsible (Umo et al., 2015, Grawe et al., 2016).

page 23, line 15: In your argument about the pore ice, it was not clear to me right away how it should have gotten there in the first place – after rereading the sentence I now assume that likely you imagine that ice nucleated in a previous cloud cycle, and in the following one, ice crystals can form now more readily? Maybe describe this more explicitly?

page 24, line 2-4: In the framework of "pore condensation and freezing", indeed liquid water is part of the concept. However, the fact that ice nucleation was observed below

RH required for homogeneous freezing does not necessarily suggest that liquid water is required: Deposition ice nucleation could also happen! I agree with you that maybe pore freezing is the process rather at work here, but still, the reasoning between these two sentences is somewhat flawed. The observation described in the first sentence does not result in the conclusion you draw in the second sentence. Reformulate.

page 25: Why does this start with Appendix B? What happened to A? (Probably an issue for typesetting, anyway.)

Table B1: What do #1 and #2 stand for? And why does "C (#1)" occur twice?

Figure E1: z-values do not fit to what is shown on y-axis - is this only a fraction of the profile that you show here? Please clarify.

Chapter B2: The here presented analysis, and others, were done for bulk samples, while ice nucleation was measured for size segregated particles - possible implications for this should be discussed somewhere (not here, but up in the main text, where you draw conclusions about the connection between soot characteristics and their ice activity.

Table B4: Is there a reason why you give the factor C, but not (or not also) the effective density? The latter could be of interest for the readers.

My only remark for the supplement: Fig. S18: Please mention in the caption, that the length of the bar in the picture is 800nm for all pictures in the left column and 200nm for all others - otherwise this needs blowing up quite a bit before these numbers can be seen, which is a bit annoying.

Literature: Grawe, S., S. Augustin-Bauditz, S. Hartmann, L. Hellner, J. B. C. Pettersson, A. Prager, F. Stratmann, and H. Wex (2016), The immersion freezing behavior of ash particles from wood and brown coal burning, Atmos. Chem. Phys., 16, 13911–13928, doi:10.5194/acp-16-13911-2016.

Umo, N. S., B. J. Murray, T. M. Baeza-Romero, J. M. Jones, A. R. Lea-Langton, T. L.

Malkin, D. O'Sullivan, L. Neve, J. M. C. Plane, and A. Williams (2015), Ice nucleation by combustion ash particles at conditions relevant to mixed-phase clouds, Atmos. Chem. Phys., 15, 5195–5210, doi:10.5194/acp-15-5195-2015.
* * *

---

## Referee Comment (RC2) · Anonymous Referee #2 · 15 Jul 2018

This manuscript describes the ice nucleation activities of different types of laboratory generated soot particles using the horizontal ice nucleation chamber (HINC). They performed ice nucleation experiments for four different size (mobility diameter) selected particles using DMA. Furthermore, they investigated particle morphology using TEM, and DMA-CPMA; temperature induced mass loss using thermogravimetric analysis; and water uptake using dynamic vapor sorption measurements. They attempted to link all these measurements with the ice nucleation activities of different types of soot.

Overall, the authors found that soot particles are not active in the mixed-phase cloud condition but some of the soot types are active in the cirrus cloud regime. The authors suggested that pore condensation and freezing (PCF) mechanism may be responsible for ice nucleation. Overall, the paper is clearly written and quite detailed. I appreciate

all the details provided by the authors. Some suggested clarifications are listed below. I recommend this paper for publication after the comments outlined below are taken into account.

General comments:

1) One of the main concerns is the size distribution of the particles investigated here. Even though the authors size selected the particles using DMA, but due to fractal morphology of soot the physical diameters are quite different than mobility diameter. Depending on the soot morphology and flow, the difference between geometric diameter and mobility diameter varies. I suggest to add the size distribution of soot particles if available. Authors provided SMPS size distribution for FS soot in the supplementary material but it will be useful to provide size distribution for all the soot types, especially for FW200 soot. If SMPS data is not available, then the authors can use TEM images to provide size distribution of soot particles (like they provided the size distribution of soot monomers). For example, looking at the TEM images of FW200 (the most effective INP investigated here), it seems like these particles are quite bigger in size compared to other soot particles investigated here. Authors should also discuss about the multiply charged particles in DMA.

2) It seems like the differences in onset saturation ratio between 228K and 233 are significant for FW200 soot for both 300 nm and 400 nm size. Can you explain why?

3) LB-RC soot nucleated ice in the circus cloud regime below homogeneous freezing and second most efficient INP investigated here for 300 nm size selected soot even though the surface area of LB_RC soot is less than an order of magnitude lower compared to FW200 soot (Table B1). Why LB_RC soot is relatively active even with low surface area? Overall size of the LB_RC soot aggregate is smaller compared to FW soot but monomers of LB_RC is too large (152 nm) compared to typically monomer size of soot in the atmosphere.

4) Overall, the discussion of soot aggregate porosity is rather qualitative and they tried

to make a link with PCF freezing. Perhaps authors can use the BET surface area measurements to make conclusions or make an attempt to provide more quantitative information on the porosity.

Minor comments:

1) Page2, line6: typically in the atmosphere primary particle diameter of soot particles ranges from 15-60 nm. Several hundreds of nanometer sounds too large to me.

2) Page 13, line 34: please provide the number of aggregates and monomers analyzed.

3) Page 14, line 1: "the most ice active FW200 soot shows particularly densely clustered aggregates" – why the DMA-CPMA derived fractal dimension is low then compared to other soot investigated? May be it's related to coating that added to mass. For example, FS particles seem more coated and has higher Dfm compared to other soot samples.

4) Page 14, lines 5-10: "...soot particles with smaller spherules are more likely to nucleate ice via a PCF mechanism"- I didn't follow this part. How did you come to this conclusion?

5) Figure 4: perhaps the authors can consider to plot using log-log scale, then show the power fit. Then it will be easier to read the fractal dimension.

6) Table B4: please provide the error for pre-factor and fractal dimension.

7) It is interesting that FW200 soot samples show significant mass loss from TGA experiments and also show highest ice nucleation ability. Significant mass loss below 200C suggest that there were volatile material. I didn't follow why authors refer this observation as presence of hydrophilic sites? Why there were condense water? May be I missed something how the experiments were performed. Also, I'm surprise by the amount of mass loss. It suggest that there were quite a bit of volatile material in the soot sample. Information about the chemical composition of soot samples would have been helpful.

8) Page 23, line 15: May be add some examples of atmospheric processing of soot after long-range transport when soot particles become more compact (change contact angle) or coated with other materials.

---

## Referee Comment (RC3) · Anonymous Referee #3 · 7 Aug 2018

In this paper, authors measure the ice nucleation properties of various soot samples and further interpret these measurements by characterizing the particle properties and conclude that PCF kind of mechanism can explain the diverse INP properties observed for soot particles. My major concern is properties of realistic (emitted directly into the atmosphere through natural and anthropogenic processes) soot aerosol could be different than soot generated in the lab through dry-dispersion method, and this could affect the conclusions. Lab generated particles may not be atmospherically relevant particles. It is necessary to discuss how this connection can be made. An attempt is made in section 4, but not enough. The surface properties of soot samples from both sources are very different. Detailed discussion including these limitations needs to be explained and acknowledged. In addition, I have few following minor comments that I

suggest to address before publication.

Page 2-3: Not all historical studies are discussed. It sounds like this is the first paper to study a majority of the soot samples (section 2.1 or Table 1). It may be best to discuss these studies briefly.

Section 2.2: What is the size distribution of soot samples? What size was selected (page 6, line 9)? How was multiple charge correction applied?

Page 8: For ice crystal detection 1 um threshold was used. How it is ensured that soot particles are not larger than 1 um. Figure 5 shows the images of these soot samples, and it looks particles are greater than 1 um. Therefore, ice particles will grow larger than 1 um. Please clarify. The implications are briefly explained on page 9 (line 5), but details regarding how 'aerosol-correct' is needed.

Page 22, line 31-: If the soot properties of combustion aerosols are different than lab-generated, how it is possible that ice nucleation properties of 100 nm will be similar? The sentence saying '...it is unlikely that such small soot particles ....' is not true.

Section 4: Discussions such as comparison of residence time (timescales) within HINC and aircraft plume is irrelevant. Soot properties in both cases are not similar. Also, the assumption regarding cloud formation cycles is not supported.
* * *

---

## Author Comment (AC1) · 23 Aug 2018

Reviewer comments are reproduced in **bold** and our responses in normal typeface; extracts from the originally submitted manuscript are presented in *red italic*, and from the revised manuscript in *blue italic*.

We have numbered the reviewer comments for ease of cross-reference within the other reviews.

**(1) In the manuscript, the ice nucleation activity of 6 different soot samples is examined with the CFDC-type ice nucleation chamber HINC at the ETH (Zurich). Additionally, a thorough analysis of physical properties of the soot particles is presented. All data are jointly discussed. It is concluded that the examined soot particles do not act as ice nucleating particles in the mixed cloud regime. In the cirrus cloud regime, some soot types were more ice active than others, which was traced back to pore condensation and freezing. The manuscript is well written. Data is nicely presented. The overall message is clear. Sometimes I felt that the information was a bit too much, but then it is all interesting and well researched, so I will not suggest so shorten anything. Below, I give a number of (mostly small) remarks / suggestions for improvements. The number of remarks makes me choose "major revisions", but it should all be easily done. Overall I recommend this manuscript for publication in ACP once these suggestions will have been addressed.**

We thank the reviewer for carefully reading the manuscript and the overall constructive comments on it. We appreciate the detail of the comments provided and hope that the responses below satisfactorily address the reviewer concerns.

**Specific comments:**

**(2) page 2, line 8: Vali et al. (2015) do not introduce any research or their own, but rather summarize knowledge, thus this citation here is a bit misleading. They don't show that soot particles can act as INPs. I suggest removing this citation here.**

We have removed this reference, as suggested (page 2 line 8 in revised manuscript).

**(3) page 4, line 25: You claim that your sample "LB_RC" is "directly comparable to the lamp black soot purchased from OEC, due to their very similar physical properties" – however, the BET surface areas of these two soot samples is an order of magnitude apart, so I suggest to delete this statement about their comparability, or adjust it accordingly.**

We are grateful for the reviewer to have realized this. There is a typo in the numeric value given for the surface area of LB_RC on p.4, l.24:

*"Its specific surface area was measured to be 233 $m^2 g^{-1}$"*

We changed it to (page 4 line 24 in revised manuscript):

*"Its specific surface area was measured to be 23 $m^2 g^{-1}$"*

This is also in accordance with the value reported in Tab.1.

**(4) page 5, Table 1: It would be interesting to know how many particles were counted in the TEM analysis. This could be given in an additional line.**

We believe you are referring to the number of primary particles counted to derive the mean primary particle diameter indicated in Tab. 1. The corresponding figure and number is shown in Fig. S17. We have now added information to Tab. 1 to show the number of particles counted.

**(5) page 6, line 8: What were these filters used for? It would be nice to get this information here in the text.**

We primarily used the mCAST filters to obtain bulk soot aerosols of the mCAST samples for TGA, DVS and BET analysis. We agree with the reviewer that the use of the filters should be more clearly specified in the main text. We therefore deleted the following statement on p.26 l.1.

*"For the miniCAST samples, not available in powder form, soot was collected on 47 mm diameter quartz fiber filters (Tissuquartz Filters, Type 2500QAT-UP, Pall Inc.), using a 47 mm aluminum in-line filter holder. The filter holder was mounted at a distance of 10 cm downstream of the miniCAST exhaust pipe, using an air-cooled stainless-steel pipe. Soot aerosols were then carefully removed from the filter with a metal spatula prior to TGA analysis."*

And instead extended the statement on page 6 line 11 in the revised manuscript to read:

*"mCAST samples collected for analysis were directly sampled from the miniCAST outlet, upstream of the VKL10, on 47 mm diameter quartz fiber filters (Tissuquartz Filters, Type 2500QAT-UP, Pall Inc.), using a 47 mm aluminum in-line filter holder. The filter holder was mounted at a distance of 10 cm downstream of the miniCAST exhaust pipe, using an air-cooled stainless-steel pipe and connected to a vacuum pump, operated at a constant flow rate of 20 Lmin$^{-1}$. Soot aerosols were then carefully removed from the filters with a metal spatula for bulk particle analysis as specified in Sect. 2.4."*

**(6) page 6, line 10: Does this mean that the mini-CAST samples on the filters were sampled BEFORE this additional coagulation took place? Please give the answer to this question in the text.**

Yes, collection on quartz fibre filters took place upstream of the mixing chamber (where coagulation occurred). This is now clarified on page 6 line 11 in the revised manuscript. We also added the filter collection stage to Fig. 1 for clarity.

**(7) page 6, line 18 and 19: You use HINCNE and HINCNE (i.e., once is in italic, the other isn't). Be consistent throughout the text. My personal preference is the one with italic letters.**

We have changed this to italics *NE* and *ST*, respectively. We also changed this in the caption of Fig. S16 to italics.

**(8) page 7, line 7-9: It is not so clear to me if you really can assure that the aerosol flow through the DMA is stable – did you check the exhaust air from the mixing chamber regularly? Was it constant? And were the flows in the DMA stable?**

Yes, the flow through the DMA column was stable. The DMA output flow was regularly checked between experiments with a flow meter mounted inline between the DMA outlet and the flow splitter depicted in Fig. 1 and deviations were found to be well below 10% over a period of 3h. Besides the pressure within the mixing chamber was continuously monitored using a pressure sensor. We refer to this on page 7 line 7 of the original manuscript:

*"The over-pressure was regulated with a needle-valve controlled exhaust, mounted to the mixing volume."*

But we further clarify this now on page 7 line 19 of the revised manuscript

*"The over-pressure was regulated with a needle-valve controlled exhaust, mounted to the mixing volume, where the pressure was continuously monitored using a pressure sensor."*

We also added on page 7 line 21 of the revised manuscript

*"In addition, a stable correct flow through the DMA was ensured by checking the flow between the DMA and the flow splitter over regular intervals of approximately 3 h."*

**(9) page 7, line 22: The phrase "Cloud particles can be formed" is not correct. Ice may or may not form, as you say, and water droplets will only form at RH_w > 100% and therewith at much higher humidities. Therefore, it would be better to say "Ice crystals may be formed".**

We intended to use the term cloud particles to encompass both ice crystals and cloud droplets. You are right that the sentence as written is incomplete and with that incorrect, as it only holds for T < 273 K (no ice saturation for higher T). As in our experiments both ice crystals and cloud droplets can be formed at the respective RH conditions we would like to make the reader aware of this possibility at this stage, also it is more detailed when discussing the WDS (p.8 l.10). We thus revised the sentence on p.7 l.22 from:

*"Cloud particles can be formed within the chamber by exposing the injected aerosol particles to RH conditions $RH_i$ > 100 %, where the subscript I denotes evaluation with respect to ice."*

To (page 8 line 1 in the revised manuscript)

*"Ice particles can be formed within the chamber by exposing the injected aerosol particles to conditions of $RH_i$ > 100 %, where the subscript i denotes evaluation with respect to ice, and cloud droplets can be formed for conditions of $RH_w$ > 100%, where the subscript w denotes evaluation with respect to water."*

**(10) page 8, line 1-2: "The residence time is the sum of a nucleation and water droplet or ice crystal growth time in the chamber." – This is strangely formulated. Please reformulate. Maybe "The residence time is the sum of the times..."**

*"The residence time is the sum of a nucleation and water droplet or ice crystal growth time in the chamber."*

Is now changed to (page 8 line 18 in the revised manuscript):

*"Assuming a perfectly parabolic velocity profile across the chamber, the aerosol particles are assumed to travel at the maximal velocity at the center of the profile, which is used to derive the particle residence time in the chamber, i.e. the time it takes a particle to cross the chamber. The residence time can be divided into the time it takes to nucleate an ice crystal (or activate a water droplet) and the subsequent growth time of the particle within the chamber."*

**(11) page 8, line 14-15: How long does it take, until the particles reach the destined temperature after entering HINC? If both sheath and aerosol flow are at laboratory temperature upon entering HINC, there will certainly be a non-negligible delay. This needs to be mentioned and shortly discussed.**

The reviewer raises a valid point here. The time needed to equilibrate the aerosol and sheath flow from room temperature to the temperature conditions within HINC is on the order of 0.3-2 seconds (depending on temperature) before steady state is achieved, as detailed in Kanji and Abbatt (2009), and thus only a small fraction of the residence time used here. However, we note that the sheath flow has a longer residence time in the chamber before it joins the aerosol flow. The sheath flow enters HINC upstream of the aerosol flow and as such has approximately 8-9 seconds in HINC before meeting the aerosol flow, thus should already be equilibrated to the chamber conditions (T and RH). Furthermore, the aerosol flow also traverses through the injector inside HINC before joining the sheath flow at the injector exit. Thus when the aerosol flow meets the sheath flow at the centre temperature of HINC, only the aerosol flow (~1/10th of the total flow) needs to equilibrate to the HINC RH conditions which should be instantaneous given the aerosol would be temperature conditioned in the injector.

We included a discussion of this point on (page 8 line 12 in the revised manuscript)::

*"Both $F_{AP}$ and $F_{sheath}$ are introduced into HINC at approximately room temperature conditions, however, $F_{sheath}$ is introduced at the beginning of HINC (prior to $F_{AP}$) and thus will reach steady*

*state conditions of temperature and water vapor upon entering the chamber prior to joining the aerosol flow. $F_{AP}$ (~$1/10^{th}$ of $F_{OPC}$) should equilibrate with the temperature and to the saturation conditions in HINC within 0.2 - 2 s, as described by Kanji and Abbatt (2009) and Lacher et al. (2017) depending on the temperature in HINC."*

We also changed on p.8, l.2 (initial manuscript):

*"For all experiments presented here a particle residence time of τ ≈ 16 s was chosen, allowing the nucleated ice crystals to grow to sizes of a couple of micrometers in diameter within the chamber."*

To (page 8 line 22, revised manuscript)

*"For all experiments presented here a particle residence time of τ ≈ 16 s was chosen. This is well above the maximum time needed for the airstream to reach steady state conditions within HINC (0.2 – 2 s) and allows the nucleated ice crystals to grow to sizes > 1 $\mu m$ (ice detection threshold size) in diameter within the chamber."*

**Also, on the opposite side, could settling of the particles to the lower plate be an issue?**

Particle settling in HINC is a valid concern. This issue has been addressed in Kanji and Abbatt (2009) and Lacher et al. (2017). Owing to the different residence times used in this work than in the previously mentioned studies, we have added another section (D2) to the appendix of our manuscript, to discuss hydrometeor growth and settling.

**(12)    page 8, line 31: Certainly you mean for sizes from 1 µm onwards, and not only in the 1 µm channel? Correct this.**

This is correct; the AF is defined as the ratio of all particles larger than 1 µm to the total number of particles entering HINC. This is consistent with our statement on p.8 l.5 of the original manuscript:

*"The OPC can count and size particles in the size range between 0.3 µm and 10 µm (optical diameter) and can be operated at six different, customizable size bins within this range. However, it does not have phase discrimination capability, as such discrimination between interstitial aerosol particles, cloud droplets and ice crystals is based purely on optical particle size. Here, we choose the 1 µm size bin as the threshold to detect ice crystals in HINC, i.e. particles with optical diameters >1 µm."*

For clarification, we have added the following statement on page 8 line 28 of the revised manuscript:

*"The OPC was operated in normal (cumulative) mode such that the number counts within each channel correspond to particles of that optical size and larger."*

Besides we have changed our nomenclature from $n_{ice,CH1\mu m}$ to $n_{ice,CH>1\mu m}$ for clarification (page 9 line 17 in revised manuscript).

**(13)    Chapter 2.4: This chapter is an interesting and impressive addition to the ice nucleation measurements - but it would be good to know which of these measurements were performed on the bulk samples and which on the size segregated particles, and for the latter, if then the same set-up described above was used. Please add information on this to the text.**

TEM and DMA-CPMA measurements were performed on size segregated particles. TGA, BET and DVS measurements were performed on bulk soot samples.

For clarification, we now added the following (page 10 line 6 of revised manuscript):

*"Therefore, the TEM sampler and the CPMA were operated directly downstream of the flow splitter depicted in Fig. 1, i.e. on (mobility) size selected aerosol particles."*

For the case of the DMA-CPMA measurements this is already described in more detail on p.27 l.15-21 (initial manuscript, now at page 29, line 23):

*"For this purpose, DMA-CPMA are coupled in series in such a way that aerosol particles are first sent into the DMA, classifying the particles by electrical mobility (selecting by drag:charge), resulting in a narrow size distribution. […] The particles are subsequently passed through the CPMA, selecting monodisperse aerosol by mass to charge, and counted by a CPC (Model 3776, TSI Inc.) downstream of the CPMA, operated at a flow rate of 0.3 Lmin$^{-1}$, yielding the number concentration of size and mass selected aerosol."*

In addition we changed p.9 l.16 (initial manuscript) from:

*"In addition, all soot types were investigated by means of a thermogravimetric analyzer […]"*

To (page 10 line 7 in revised manuscript)

*"In addition, bulk soot properties were investigated by means of a thermogravimetric analyzer […]"*

Further, we changed p.9 l.20 from:

*"[…] water vapor sorption isotherms of the soot particles were measured by Dynamic Vapor Sorption (DVS, Model Advantage ET 1, Surface Measurement Systems Ltd., London, UK)."*

To (page 10 line 11 in revised manuscript)

*"[…] water vapor sorption isotherms of the bulk soot samples were measured by Dynamic Vapor Sorption (DVS, Model Advantage ET 1, Surface Measurement Systems Ltd., London, UK)."*

Accordingly, we changed p.9., l. 21 from:

"Finally, the BET specific surface area of the samples was determined from additional $N_2$ adsorption measurements ($a_{BET,N2}$)."

To (page 10 line 12)

"Finally, the BET specific surface area of the bulk soot samples was determined from additional $N_2$ adsorption measurements ($a_{BET,N2}$)."

**(14)    page 9, line 13: Better replace "aid" by "contributed to".**

Change made (page 10 line 2 in revised manuscript).

**(15)    Figure 2: The "blue dashed line" – is that the one that is almost at the RH_w = 1 - line? This is almost impossible to see - use a different color, e.g., red, or small circles or something else.**

We changed the color to red and slightly increased the line width for visibility. We changed relevant parts of the manuscript to refer to the red WDS line.

**(16)    page 11, line 11-13: Would this then not mean that every possible ice active material would be outcompeted by homogeneous freezing?**

No, this would not be the case. It depends on how ice active the other types of INPs are that the reviewer refers to. For example, if a particle is only active as an INP at very low temperatures and high RH say close to the conditions of homogeneous freezing, then yes, homogenous freezing rates begin to get very high at such conditions and heterogeneous nucleation would not contribute to ice crystal formation in the atmosphere.

On the other hand if the ice active material was an INP at warmer temperatures like mineral dust e.g. for temperatures as warm as -20 °C, then of course homogeneous nucleation rates at these temperatures are negligible for volumes of droplets in the troposphere, as such heterogeneous freezing would outcompete homogeneous freezing.

**Still, heterogeneous freezing is considered to play a role even for the cirrus cloud regime.**

This is true, we agree with the reviewer, that heterogeneous freezing plays a role in cirrus clouds and is even believed to be the dominant freezing process (e.g. Cziczo et al., 2013) if dust aerosol and other species are present. However, this should be true only for cases where the heterogeneous freezing is observed to occur below the RH required for homogeneous freezing (see Kuebbeler et al., 2014). Again, if RHs as high as those required for homogeneous freezing are also required for heterogeneous freezing, then homogeneous freezing will outcompete heterogeneous freezing simply because there are orders of magnitude higher droplets in the atmosphere than INPs.

**And if this (second) part of the sentence can be debated, I wonder if the first part (about 100 nm particles being not relevant) is not a trade of between the HINC detection limit and the abundance of these particles in the atmosphere?**

We do not believe this statement has anything to do with the HINC detection limit. If a particle of a given aerosol type and size is only ice active at or above homogeneous freezing conditions, as our 100 nm soot particles tested here, then we can no longer state that freezing occurred heterogeneously, i.e. when water condensed on soot particles because of high excessive amounts of humidity like in a contrail. Since at these temperatures, ice germs can form within the bulk volume of water (i.e. an interface with the INP is not required, nor necessary to lower the energy barrier of nucleation of a new phase), the freezing is considered homogeneous, because the surface of the soot plays a role for water condensation, but is not necessary for the freezing process since this can occur in a bulk water droplet without any solid interface in it. In the atmosphere, if soot does not nucleate ice via deposition nucleation or PCF, even up to *RH* required for homogeneous freezing ($T < \sim$ -36 °C), then solution or water droplets will begin to freeze since their nucleation rates are would be significant at these conditions rendering any nucleation from soot particles negligible.

**(17)    page 11, line 25: Missing space between y and 2.**

We added a space.

**(18)    page 11, line 32-33: Another reason could be that it takes up water – could be worthwhile mentioning.**

We agree with the reviewer that the lower ice nucleation onset of FW200 soot is caused by its enhanced ability to take up water (lower contact angle) compared to the other soots tested here. However, we see no need to discuss this at this point of the manuscript, as this is discussed in great detail in Section 3.5 and should ultimately become clear by the following statement on p.19, l.15 (initial manuscript, page 20 line 5 in revised manuscript):

*"The high water affinity of FW200 revealed by the DVS measurements is consistent with the observed ice nucleation at relatively low ice supersaturations."*

**(19)    page 12, line 4: A new sentence should start between "particles" and "they".**

Change made (page 12, line 28 in revised manuscript).

**(20)    page 14, line 7: Add a "," between "increases" and "the" (it took me a while until I got the sentence).**

Suggested change made (page 16, line 3 in revised manuscript).

**(21)    page 14, line 10: Tiny remark, but why "in prep.b" first (before "a")?**

This reference is now updated to "submitted" and should resolve this issue (page 16, line 7 in revised manuscript).

**(22)    page 15, line 7: There is a space too much before the first letter.**

We have amended that and indented the whole paragraph (added more space in front of "Highly"), in order to mark the change in paragraph (page 16 and line 21 in revised manuscript)

**(23)  Figure 6: One of the screens I used while working at this review did not show this graph well. The "yellow" color shading looked rather pinkish, and the shading of the two other panels was almost the same - maybe there is a better (or additional) way to refer to the panels? (Using a), b) and c) for example?)**

We removed the shading and added arrows and labels (as suggested) for each temperature range to the figure.

For consistency, we also revised Fig. S13 and its caption accordingly.

**(24)  page 17, line 19-20: Are you really referring to S16 in the SI, here, so should this rather be S10? But in any case, I have difficulties following your argument – why does the lack of a change in the slope render this interpretation unlikely?**

We are in fact referring to Fig. S16, as stated in the manuscript, however, we agree with the reviewer that the way it is written in the current version is confusing. We will adjust the manuscript to clarify our argument, which is as follows:

Fig. S16 shows an ice nucleation experiment using $NH_4NO_3$, i.e. a salt aerosol, at 233 K which should freeze homogeneously at this temperature at RH conditions predicted by the Koop et al. (2000) parametrization (vertical black dashed line).

Our Fig. S16 shows how the injected (dry) salt aerosol particles deliquesce and grow by hygroscopic growth and ultimately form solution droplets between approximately 95% and 99% $RH_w$. This growth is associated with a moderate slope of the AF curves shown. Upon reaching the critical $RH_w$, homogeneous freezing of the solution droplets occurs (between 99% and 100% $RH_w$, i.e. within instrumental uncertainty), the slope of the AF curves becomes steeper ("step-like"), indicating the formation of ice crystals of nearly all solution droplets. This change in slope of the AF curves at homogeneous freezing conditions (within the uncertainty range given by the gray shaded region), indicates a change in mechanism occurring (i.e. the nucleation of ice crystals through homogeneous freezing). In other words, we expect this steepness in the slope of the AF curves, since we expect a large change in nucleation rate once homogeneous freezing conditions are reached.

Given that we do not observe any such change in the slope of the AF curves of our FS sample shown in Fig. S10, we believe that the FS samples do not form droplets that then freeze homogeneously in this case.

We now clarify this aspect in the manuscript on page 18 lines 13-16.

**(25)  page 17, line 33: Change "of" to "the"**

Done! (page 18 line 29 in revised manuscript).

**(26)  Figure 7: If you used a log-scale for one of the two panels, it would be easier to see the hysteresis and the absolute values for the curves at the lower end. The way it is now, both panels look quite similar, anyway.**

The pores relevant for an ice nucleation mechanism via PCF are mesopores (> 2 nm) start filling at approximately $RH_w$ > 60-70%, depending on the pore type (shape). Thus, our figure aims at showing the difference in water uptake of the soot types mainly at these high relative humidities, where also the resolution of our DVS scans is largest. Therfore we prefer the linear axes. However, we include the figure using log axis below for the reviewer's information. Mircoporosity on the contrary, which can be infered from the DVS scans at lower RH values, does not contribute to ice nucleation via PCF, as these pores are too small to accommodate an ice germ. They are, however, relevant for soot particle restructuring due to water uptake, which is not part of the current manuscript.

[Figure]

[Figure]

**(27)    page 19, line 21: Are citric acid and glutaric acid organics typical for the atmosphere? Any citation on that? Others might play a more important role (oxalic acid, formic acid, acetic acid, succinic acid, ...). But there must be a reason why you choose those. Explain that in a few words.**

Glutaric acid is a common dicarboxylic acids in the atmosphere (Winterhalter et al., 2009, Kawamura and Ikushima, 1993) and its effects on soot have been studied in the past (Xue et al., 2009). Citric acid is less common but has also been identified in atmospheric aerosols (e.g. Kawamura and Yasui, 2005). Their hygroscopic growth is typical and a proxy for that of water-soluble organic aerosol fraction. Formic and acetic acid do not partition to the condensed phase in atmospheric aerosols while oxalic and succinic acid effloresce and deliquesce when RH is decreased and increased, respectively. Deliquescence/efflorescence is not expected for organic aerosols which are mixtures of many compounds (Marcolli et al., 2004) and therefore does not represent typical hygroscopic growth of atmospheric aerosols.

**(28)    page 20, line 18-21: The last few words "rather than the bulk aggregate size" puzzled me a bit. There is a difference in the freezing curves for FW200 particles of different sizes, and you say, quite correctly, that you observe 1% of particles ice active at similar RHs for a given particle size (BTW: exchange "aerosol size" with "particle size" here). But then why would you refer to "bulk aggregate size" in the end. Would anyone really assume that this plays role? I suggest to delete these few last words unless you feel they are needed.**

The "bulk aggregate size" was meant in terms of an "overall" aggregate size. We agree that there is no need for this (repetition of sentence before) and changed the sentence accordingly to avoid confusion (page 21 line 18 in revised manuscript):

*"The constant onset $S_i$ indicates that the freezing is in fact determined by pore size and soot-water contact angle, which determine water filling of the pores and subsequent homogeneous freezing."*

We also changed *"aerosol size"* to *"particle size" (page 21 line 18 in revised manuscript)*

**(29)    page 21, line 7: I suggest to tune down "confidently identify" a little.**

We assume that the referee refers to p.22 l.7 instead. We changed it to (page 23 line 6 in revised manuscript) to:

*"Combining our ice nucleation results with data obtained from TGA and DVS, we identify PCF as the dominant mechanism to cause the freezing of the tested soot particles."*

**(30)    page 22, line 22: Concerning the use of the word "contact angle", I realize that you talk about the interaction between the soot surface and water vapor. But on the other hand, it has been shown for other materials that it never only is one contact angle describing the ice nucleation behavior of one substance, but a contact angle distribution instead. This might need to be mentioned.**

The reviewer raises a good point here and we agree that it is more meaningful to think of a contact angle distribution, rather than one absolute value.

On p.13, l.4-10 we discuss the steepness of the AF curves of the FW200 sample ("step-like") and argue that none of the other soots reveal a similarly clear/steep AF curve. This is likely caused by the FW200 particles having the right properties (pore size and contact angle), but also more homogeneous physicochemical properties compared to the other soot types. We argue, for instance, that there is a freezing attributable to PCF for both lamp black samples in the cirrus regime (Fig. 3). One can think of the absence of any step-like activation for the lamp blacks to be caused by more heterogeneous particle properties, where not every soot aggregate has the "right match" of contact angle and pore size to fill and freeze within a narrow defined $RH_w$ range, causing the AF curve to be less steep, i.e. activation over a broader range of RH conditions at a given temperature. In other words, the contact angle distribution and pore size distribution have a stronger interplay in case of the FW200 particles, compared to the

other soot types. We hint at this early in the manuscript and revisit it in a bit more detail later. We revised the manuscript as follows:

p.13, l.10: *"[…] the observed difference must be related to the physical and chemical properties of the particles."*

Is changed to (page 16 line 12 in revised manuscript):

*"Thus, the observed difference must be related to other physical and chemical properties of the particles, in addition to the morphology, which might be more heterogeneous for the other soot types"*

p.19, l.18: We added (page 20 line 14 in revised manuscript):

*"The steep AF curves of FW200 compared to the other soots, as discussed in Sect. 3.2, can be thought of as an overlap of pore size distribution and associated contact angle distribution that favours condensing water in pores, resulting in particles with properties suitable to nucleate ice via PCF, whereas those soot types with less steep AF curves indicate a more heterogeneous distribution of particle properties (contact angles and pore sizes)."*

Finally, we changed p.22, l.22 from:

*"Specifically, our DVS results reveal that both pore structure and contact angle determine the ice nucleation ability of the soots studied."*

To (page 23 line 21 in revised manuscript)

*"Specifically, our DVS results reveal that both pore size distribution and contact angle distribution determine the ice nucleation ability of the soot particles studied."*

**(31)  page 23, line 5: It should not be forgotten that solid fuels (from either biomass burning or wildfires) also produce ashes, so for observed atmospheric ice activities of respective particles, also these ashes could be responsible (Umo et al., 2015, Grawe et al., 2016).**

Soot and ash particles differ in their chemical composition, as discussed for instance in Grawe et al. (2016). Nevertheless, we agree with the referee that such ash particles, which have been shown to be ice active, can be sourced from biomass burning and wildfires. We now add this on page 24 line 9 of the revised manuscript:

*"At the same time such solid fuels can also produce ash particles, which can contribute to ice formation (Grawe et al., 2016, Grawe et al., 2018, Umo et al., 2015)."*

**(32)  page 23, line 15: In your argument about the pore ice, it was not clear to me right away how it should have gotten there in the first place – after rereading the sentence I now assume that likely you imagine that ice nucleated in a previous cloud cycle, and in the following one, ice crystals can form now more readily? Maybe describe this more explicitly?**

This is exactly what we mean. In order to clarify this we changed the sentence p.23 l.15 (original manuscript) from:

*"In case of ice nucleation via a PCF mechanism, pore ice can remain trapped within the cavities between the cloud cycles, thus particles could grow into macroscopic ice crystals, as soon as the $RH_i$>100 %."*

To (page 24 line 21, revised manuscript):

*"In case of ice nucleation via a PCF mechanism, pore ice can remain trapped within the cavities (microscopic pore ice) between subsequent cloud cycles for certain conditions of T and RH, even though the macroscopic ice crystal is sublimated. For instance, an ice crystal formed on a soot particle leaving the cloud will experience an ice subsaturated environment and thus sublimate. However, given that RH conditions outside the cloud are high enough (at T < 273*

*K), the ice within the pores can survive, due to the reduced saturation vapor pressure of ice within the cavity. This pore ice can then grow into macroscopic ice in subsequent cloud cycles when RH$_i$ = 100% is exceeded."*

**(33)    page 24, line 2-4: In the framework of "pore condensation and freezing", indeed liquid water is part of the concept. However, the fact that ice nucleation was observed below RH required for homogeneous freezing does not necessarily suggest that liquid water is required: Deposition ice nucleation could also happen! I agree with you that maybe pore freezing is the process rather at work here, but still, the reasoning between these two sentences is somewhat flawed. The observation described in the first sentence does not result in the conclusion you draw in the second sentence. Reformulate.**

We agree with the reviewer and have reformulated the sentences to read (page 25 line 16 revised manuscript):

*"In the MPC temperature regime no ice nucleation was observed below water saturation, while for some of the probed soot samples ice nucleation was observed below the RH required for homogeneous freezing of solution droplets in the cirrus regime. The absence of heterogeneous freezing in the MPC regime below water saturation suggests that deposition nucleation does not take place on the tested particles. While water can be taken up into the pores of the soot aggregates also at MPC conditions, the absence of any ice formation below water saturation indicates that there is a lack of active sites that could trigger heterogeneous ice nucleation at these temperatures. The observed ice nucleation in the cirrus regime could theoretically be caused through a (surface area dependent) deposition nucleation mechanism. However, the strong dependence of the ice nucleation efficiency on the HNT implies that it is the liquid water within the soot pores that freezes homogeneously, since particle properties considered relevant for deposition nucleation (if present) should be available for ice nucleation in both the cirrus and MPC regime. Such a dependence on the HNT relevant for liquid water freezing for ice nucleation onto soot particles investigated here is in-line with a PCF process and in contrast to classical deposition nucleation, where the liquid phase is absent. Overall, we conclude that the ice formation process on the soots is best described by a PCF mechanism and not deposition nucleation."*

**(34)    page 25: Why does this start with Appendix B? What happened to A? (Probably an issue for typesetting, anyway.)**

Thank you for spotting this. It is corrected now such that we start with Appendix A.

**(35)    Table B1: What do #1 and #2 stand for? And why does "C (#1)" occur twice?**

**1 and #2 stand for two independent BET experiments performed on the bulk soot samples. The repeated occurrence of #1 is a typo, which we corrected within the table. We now clarify the two experiments within the table caption by adding:**

*"[…] We report the values of the C parameter used in Eq. B1 for the two independent experiments labeled #1 and #2, respectively."*

**(36)    Figure E1: z-values do not fit to what is shown on y-axis - is this only a fraction of the profile that you show here? Please clarify.**

This is correct, only a fraction of the vertical distance between the two plates is shown in Fig. E1. To avoid confusion we changed the first sentence of the figure caption to:

*"Example of S$_w$ for the mid-portion of the vertical distance between the cold (at z = 0 cm, bottom) and warm (at z = 2 cm, top) wall of HINC chamber […]"*

**(37)    Chapter B2: The here presented analysis, and others, were done for bulk samples, while ice nucleation was measured for size segregated particles – possible implications for this should be discussed somewhere (not here, but up in the main**

**text, where you draw conclusions about the connection between soot characteristics and their ice activity.**

We agree with this limitation. The statement added on page 24 line 32 of the revised manuscript and posted in the reply to question (1) of reviewer #03 accounts for this.

**(38)   Table B4: Is there a reason why you give the factor C, but not (or not also) the effective density? The latter could be of interest for the readers.**

The mass-mobility pre-factor, C, is derived from fitting a power-law of the form of Eq. B4 to all data points of a given soot type shown in Fig. 4, and is given for completeness and allows the calculation of any mass associated with a given mobility diameter for each soot type for the interested reader.

We have added a figure (Fig. S20) to the manuscript, which shows the effective densities corresponding to all our DMA-CPMA data. Effective density values for other soot diameters can be calculated using eq. B3.

**(39)   My only remark for the supplement: Fig. S18: Please mention in the caption, that the length of the bar in the picture is 800nm for all pictures in the left column and 200nm for all others - otherwise this needs blowing up quite a bit before these numbers can be seen, which is a bit annoying.**

We added the following sentence at the end of the figure caption:

*"The lengths of the scale bars correspond to 800 nm for the left column and to 200 nm for the middle and right columns."*

**\*\*\*Additional changes by the authors:**

Fig. 7: There was a "°C" vs. "K" error, so we changed from:

*"[…] as measured by DVS at T = 298 °C. […]"*

To (page 19 in the revised manuscript)

*"[…] as measured by DVS at T = 298 K. […]"*

The flow rate given for the miniCAST on p.7 l.8 should read 30 Lmin$^{-1}$, this value was corrected from:

*"[…] in case of the FBG and 35 Lmin$^{-1}$ in case of the miniCAST."*

To (page 7 line 21 in revised manuscript).

*"[…] in case of the FBG and 30 Lmin$^{-1}$ in case of the miniCAST."*

CZICZO, D. J., FROYD, K. D., HOOSE, C., JENSEN, E. J., DIAO, M., ZONDLO, M. A., SMITH, J. B., TWOHY, C. H. & MURPHY, D. M. 2013. Clarifying the Dominant Sources and Mechanisms of Cirrus Cloud Formation. *Science,* 340**,** 1320-1324.

GRAWE, S., AUGUSTIN-BAUDITZ, S., CLEMEN, H. C., EBERT, M., ERIKSEN HAMMER, S., LUBITZ, J., REICHER, N., RUDICH, Y., SCHNEIDER, J., STAACKE, R., STRATMANN, F., WELTI, A. & WEX, H. 2018. Coal fly ash: Linking immersion freezing behavior and physico-chemical particle properties. *Atmos. Chem. Phys. Discuss.,* 2018**,** 1-32.

GRAWE, S., AUGUSTIN-BAUDITZ, S., HARTMANN, S., HELLNER, L., PETTERSSON, J. B. C., PRAGER, A., STRATMANN, F. & WEX, H. 2016. The immersion freezing behavior of ash particles from wood and brown coal burning. *Atmos. Chem. Phys.,* 16**,** 13911-13928.

KANJI, Z. A. & ABBATT, J. P. D. 2009. The University of Toronto Continuous Flow Diffusion Chamber (UT-CFDC): A Simple Design for Ice Nucleation Studies. *Aerosol Science and Technology,* 43**,** 730-738.

KAWAMURA, K. & IKUSHIMA, K. 1993. SEASONAL-CHANGES IN THE DISTRIBUTION OF DICARBOXYLIC-ACIDS IN THE URBAN ATMOSPHERE. *Environmental Science & Technology,* 27**,** 2227-2235.

KAWAMURA, K. & YASUI, O. 2005. Diurnal changes in the distribution of dicarboxylic acids, ketocarboxylic acids and dicarbonyls in the urban Tokyo atmosphere. *Atmospheric Environment,* 39**,** 1945-1960.

KOOP, T., LUO, B. P., TSIAS, A. & PETER, T. 2000. Water activity as the determinant for homogeneous ice nucleation in aqueous solutions. *Nature,* 406**,** 611-614.

KUEBBELER, M., LOHMANN, U., HENDRICKS, J. & KÄRCHER, B. 2014. Dust ice nuclei effects on cirrus clouds. *Atmos. Chem. Phys.,* 14**,** 3027-3046.

LACHER, L., LOHMANN, U., BOOSE, Y., ZIPORI, A., HERRMANN, E., BUKOWIECKI, N., STEINBACHER, M. & KANJI, Z. A. 2017. The Horizontal Ice Nucleation Chamber (HINC): INP measurements at conditions relevant for mixed-phase clouds at the High Altitude Research Station Jungfraujoch. *Atmospheric Chemistry and Physics,* 17**,** 15199-15224.

MARCOLLI, C., LUO, B. P. & PETER, T. 2004. Mixing of the organic aerosol fractions: Liquids as the thermodynamically stable phases. *Journal of Physical Chemistry A,* 108**,** 2216-2224.

UMO, N. S., MURRAY, B. J., BAEZA-ROMERO, M. T., JONES, J. M., LEA-LANGTON, A. R., MALKIN, T. L., O'SULLIVAN, D., NEVE, L., PLANE, J. M. C. & WILLIAMS, A. 2015. Ice nucleation by combustion ash particles at conditions relevant to mixed-phase clouds. *Atmos. Chem. Phys.,* 15**,** 5195-5210.

WINTERHALTER, R., KIPPENBERGER, M., WILLIAMS, J., FRIES, E., SIEG, K. & MOORTGAT, G. K. 2009. Concentrations of higher dicarboxylic acids C-5-C-13 in fresh snow samples collected at the High Alpine Research Station Jungfraujoch during CLACE 5 and 6. *Atmospheric Chemistry and Physics,* 9**,** 2097-2112.

XUE, H. X., KHALIZOV, A. F., WANG, L., ZHENG, J. & ZHANG, R. Y. 2009. Effects of dicarboxylic acid coating on the optical properties of soot. *Physical Chemistry Chemical Physics,* 11**,** 7869-7875.

---

## Author Comment (AC2) · 23 Aug 2018

Reviewer comments are reproduced in **bold** and our responses in normal typeface; extracts from the originally submitted manuscript are presented in *red italic*, and from the revised manuscript in *blue italic*.

We have renumbered the reviewer comments for ease of cross-reference within the other reviews.

**This manuscript describes the ice nucleation activities of different types of laboratory generated soot particles using the horizontal ice nucleation chamber (HINC). They performed ice nucleation experiments for four different size (mobility diameter) selected particles using DMA. Furthermore, they investigated particle morphology using TEM, and DMA-CPMA; temperature induced mass loss using thermogravimetric analysis; and water uptake using dynamic vapor sorption measurements. They attempted to link all these measurements with the ice nucleation activities of different types of soot.**

**(1) Overall, the authors found that soot particles are not active in the mixed-phase cloud condition but some of the soot types are active in the cirrus cloud regime. The authors suggested that pore condensation and freezing (PCF) mechanism may be responsible for ice nucleation. Overall, the paper is clearly written and quite detailed. I appreciate all the details provided by the authors. Some suggested clarifications are listed below. I recommend this paper for publication after the comments outlined below are taken into account.**

We thank the reviewer for their comments and address the concerns in the answers below and accordingly point out changes to the manuscript.

**General comments:**

**(2) One of the main concerns is the size distribution of the particles investigated here. Even though the authors size selected the particles using DMA, but due to fractal morphology of soot the physical diameters are quite different than mobility diameter. Depending on the soot morphology and flow, the difference between geometric diameter and mobility diameter varies. I suggest to add the size distribution of soot particles if available. Authors provided SMPS size distribution for FS soot in the supplementary material but it will be useful to provide size distribution for all the soot types, especially for FW200 soot. If SMPS data is not available, then the authors can use TEM images to provide size distribution of soot particles (like they provided the size distribution of soot monomers). For example, looking at the TEM images of FW200 (the most effective INP investigated here), it seems like these particles are quite bigger in size compared to other soot particles investigated here. Authors should also discuss about the multiply charged particles in DMA.**

We agree with the reviewer's point that selection of the mobility size can result in soot particles spanning a range of physical sizes. The reviewer further raises concerns that the FW200 sample could contain relatively larger particles compared to the other soots investigated, suggesting that this could contribute to its enhanced ice nucleation ability. We have added the SMPS size distributions of FW200 soot as suggested by the reviewer to the SI, Fig. S15.

As suggested by the reviewer we also made use of our TEM analysis and plotted the distribution of the area equivalent diameters derived from the projected areas, which reveals that the FW200 soot does not contain larger aggregates compared to the other soot types, but in fact is rather dominated by smaller aggregates. Furthermore this is supported by comparing the number size distributions from the SMPS of FS to FW200 (SI Fig. S15). Aware of the fact that the number of analyzed aggregates is limited, we added the following statement to the SI Sect. S8:

Page 17 line 14 of the revised SI:

*"In Fig. S19 we qualitatively show the distribution of area equivalent diameters calculated as:*

$$d_{Aeq} = \sqrt{\frac{4A_a}{\pi}},$$

*where $A_a$ denotes the (2D) projected area of a soot particle derived from TEM analysis. For Fig. S19 we only analyzed TEM grids corresponding to particles that were selected at 400 nm mobility diameter within the DMA. The mode at small equivalent diameters ($d_{Aeq}$ < 200 nm) for the FW200 corresponds to small aggregates that are made of only a few (< 10) primary particles. These are also visible in the exemplary images depicted in Fig. S18a1. We do not believe but cannot exclude whether these small particles are fragments being formed upon impaction of larger aggregates or result from other processes. From Fig. S19 there are strong indication to suggest that the most ice active FW200 soot does not contain a higher fraction of large particles relative to the other soot types, which could account for its enhanced ice nucleation ability."*

[Figure]

*Figure 1: Distribution of area equivalent diameters (eq. 1) corresponding to 400 nm mobility diameter selected soot aggregates at an aerosol to sheath flow ratio of 1/7, derived from TEM analysis. The number in brackets indicates the number of aggregates sized for the corresponding soot type.*

**(3) It seems like the differences in onset saturation ratio between 228K and 233 are significant for FW200 soot for both 300 nm and 400 nm size. Can you explain why?**

The answer to this question is stated on p.20, 23 (initial manuscript):

*"An exception are the results of T = 233 K, where the nucleation rate is likely too small for the volume of the pore water to freeze within the 16 s residence time of the particles in HINC (David et al., in prep.b)."*

In order to make this more explicit, we added a statement on p.21, l.22, in revised manuscript:

*"An exception are the results at T = 233 K, where the nucleation rate is possibly too small for the volume of the pore water to freeze within the 16 s residence time of the particles in HINC (David et al., in prep). This causes the significant increase in $S_i$ required for ice to nucleate via homogeneous freezing of bulk solution droplets, as can clearly be seen for both 300 nm and 400 nm particles of the FW200 soot, depicted in Fig. 2c and d, respectively."*

**(4) LB-RC soot nucleated ice in the circus cloud regime below homogeneous freezing and second most efficient INP investigated here for 300 nm size selected soot even though the surface area of LB_RC soot is less than an order of magnitude lower compared to FW200 soot (Table B1). Why LB_RC soot is relatively active even with low surface area? Overall size of the LB_RC soot aggregate is smaller compared to FW soot but monomers of LB_RC is too large (152 nm) compared to typically monomer size of soot in the atmosphere.**

The reviewer is right that the surface areas of the FW200 and the LB_RC samples are significantly different (see Tab.1). However, we suggest that ice formation on soot particles does not take place via a (surface area) dependent deposition nucleation process, but instead a PCF mechanism prevails. The latter one being dependent on pore size and contact angle, but not directly on the surface area available on the INP. In other words, as long as the LB_RC soot aggregates have pores with the right physicochemical properties, these particles will be able to form ice via PCF. Thus, assuming an identical pore size and contact angle distribution for FW200 and LB_RC we would expect them to nucleate ice within our experiments at the same RH conditions at a given T, independent of the larger surface area of the FW200 sample.

The larger surface area of the FW200 is believed to result from the smaller monomer size (Fig. S17).

We have described the difference in surface area of the soots in context of deposition nucleation and PCF on p.11, l.32-p.13, l.7 (old manuscript).

We have revised the following statement to clarify the independence from particle surface area:

p.12, l.3: *"This suggests a homogeneous freezing mechanism for the FW200 particles below water saturation."*

To (page 12 line 24 in revised manuscript):

*"This suggests a homogeneous freezing mechanism for the FW200 particles below water saturation, most likely not directly related to the particle surface area."*

In addition, we have added a statement (page 25 line 15 revised manuscript) for clarification. See our reply to question (33) of reviewer #01.

The monomer size stated for the LB_RC corresponds to our TEM data (see Fig. S17). We agree that this is rather large compared to atmospheric soot particles. See our reply to point (1) of reviewer #03. Besides, the LB_RC sample is very heterogeneous in primary particle size, as shown in our Fig. 5, resulting in the rather large mean primary particle size (152 nm) reported. This does not preclude LB_RC aggregates with smaller primary particles to nucleate ice via PCF.

**(5) Overall, the discussion of soot aggregate porosity is rather qualitative and they tried to make a link with PCF freezing. Perhaps authors can use the BET surface area**

**measurements to make conclusions or make an attempt to provide more quantitative information on the porosity.**

We acknowledge the point raised by the author that the presented porosity is rather qualitative and mainly based on our BET and DVS data. Deriving a pore size distribution requires some a priori assumptions to be made on the particle system that is probed, for instance the pore structure (Dubinin and Stoeckli, 1980). Next, different adsorbents are used to probe these different pore types/structures (Popovitcheva et al., 2000). The uncertainty associated with the resulting (bulk) pore distribution for a complex fractal aerosol such as soot, where a variety of different pore structures are likely present, along with the instrumental cost/requirements are beyond the scope of this paper. To really control the pore size and structure, experiments with carbon nanotubes (Alstadt et al., 2017) or synthesized mesoporous particles (David et al., submitted) would be needed, but these are less atmospherically relevant and focus on a different research question.

**Minor comments:**

**(1) Page2, line6: typically in the atmosphere primary particle diameter of soot particles ranges from 15-60 nm. Several hundreds of nanometer sounds too large to me.**

We agree that most of the atmospheric soot particles have primary particle diameters below 100 nm and that significantly larger primary particles are an extreme case, but still found for some of our particles, as shown in Fig. S17. Nevertheless, we tuned down "several hundred" so that this introductory statement covers the case for the majority of the soot aerosols.

Accordingly, we changed p.2, l.6:

"The primary particle diameter itself can vary from around 10 nm to several hundred nanometers, depending on the combustion source."

To (page 2 line 5 in revised manuscript)

"The primary particle diameter itself can vary from around 10 nm to several tens of nanometers, depending on the combustion source."

**(2) Page 13, line 34: please provide the number of aggregates and monomers analyzed.**

We revised the sentence and added the number of aggregates and monomers to be more explicit as follows on page 15 line 9 of the revised manuscript:

*"This is further supported by the size distribution of primary particles shown in SI Fig. S17, for which between 10 to 50 aggregates and a minimum of 122 primary particles have been evaluated for each soot type."*

**(3) Page 14, line 1: "the most ice active FW200 soot shows particularly densely clustered aggregates" – why the DMA-CPMA derived fractal dimension is low then compared to other soot investigated? May be it's related to coating that added to mass. For example, FS particles seem more coated and has higher Dfm compared to other soot samples.**

It should be noted that the TEM images only show a 2D projection of the particles. Thus the FW200 particles could be more branched/fractal than they appear on the TEM images shown, which would be in-line with our DMA-CPMA derived fractal dimension.

We now also report the uncertainties of the derived fractal dimensions (see our reply to point (5) of reviewer #02). Considering the standard error of the fractal dimensions, the FW200 and FS sample are very similar.

**(4) Page 14, lines 5-10: ". . .soot particles with smaller spherules are more likely to nucleate ice via a PCF mechanism"- I didn't follow this part. How did you come to this conclusion?**

In case of a relatively smaller primary particle diameter an aggregate of a given mobility size is composed of a larger number of primary spherules (monomers) compared to a soot type

with a relatively larger primary particle diameter. In the aggregate with the higher number density of primary particles (smaller primary particle diameter) the chances for pores are higher and therefore the probability of ice formation via PCF is higher.

This is supported by our ice nucleation results (Fig. 3) and the primary particle size distribution (Fig. S17). The step-like AF curves of the FW200 reflects very homogeneous pore properties of this sample, which is supported by the narrow primary particle size distribution and also suggest a narrow distribution of contact angles on this sample. Other soot samples that nucleate ice via PCF have more heterogeneous distributions of primary particle sizes (pores) and/or contact angles (also see reply to point (30) of reviewer #01).

This point is already explained in the original manuscript (p.14 l 3-10) and we revised the statement further down in the text (page 16 line 1 in the revised manuscript) to read:.

*"Nevertheless, it is likely that the clear difference in primary particle size determines and/or strongly influences overall aggregate porosity. Soot aggregates of a given mobility size are composed of an increasing number of carbon spherules for decreasing primary particle sizes. As the number of spherules increases, the propensity for pores in an aggregate also increases due to the potential for pores between sintered spherules and/or through intra-aggregate cavities between the branches of the aggregate. As such, soot particles with smaller spherules are more likely to nucleate ice via a PCF mechanism due to the higher concentration of pores, resulting from the increased number of primary particles in these aggregates."*

**Figure 4: perhaps the authors can consider to plot using log-log scale, then show the power fit. Then it will be easier to read the fractal dimension.**

This is a very good idea. We replotted the figure using log-log scale and also included the fit lines along with the $R^2$ of the fits in the legend.

[Figure]

Accordingly, we added the following statement to the caption of Fig. 4:

*"Error bars correspond to standard deviations of the individual measurements and dashed lines to the power law fits using eq. A4. The values in parenthesis give the $R^2$ of the fit."*

**(5) Table B4: please provide the error for pre-factor and fractal dimension.**

We have added the standard error of the fit values for the pre-factor and the fractal dimension to the table, and added the following statement to the caption of Table A4 (former B4).

*"The value in brackets indicates the standard error."*

**(6) It is interesting that FW200 soot samples show significant mass loss from TGA experiments and also show highest ice nucleation ability. Significant mass loss below 200C suggest that there were volatile material. I didn't follow why authors refer this observation as presence of hydrophilic sites? Why there were condense water?**

As correctly pointed out by the reviewer, the significant mass loss of the FW200 sample in the TGA (Fig. 6) indicates the presence of *"highly volatile compounds"* (p.16, l.21, revised manuscript). This term encompasses any material that volatilizes at temperatures below 200 °C. Now the question arises, whether the observed mass loss is due to *"adsorbed/condensed water and/or low molecular weight organic substances"* (p.16, l.24, revised manuscript). We interpret the majority of the mass loss of the FW200 sample to be associated through evaporation of adsorbed water but do not fully exclude the presence of any other highly volatile material. This conclusion is drawn by combining our TGA results with our ice nucleation and DVS experiments. We refer to the FW200 sample having hydrophilic sites, as it demonstrates the highest water uptake in the DVS already at very low $RH_w$ (< 30%).

To include the simultaneous presence of OM, we changed the sentence on p.19, l.25 (initial manuscript):

*"[…] attribute the water uptake of FW200 to pores present on the soot aggregates, and not only due to absorption of water vapor by hydrophilic OM."*

To (page 20 line 24, revised manuscript)

*"[…] attribute the water uptake of FW200 to pores present on the soot aggregates, and not only due to absorption of water vapor by hydrophilic OM, which is also present, as suggested by our TGA results."*

**May be I missed something how the experiments were performed. Also, I'm surprise by the amount of mass loss. It suggest that there were quite a bit of volatile material in the soot sample. Information about the chemical composition of soot samples would have been helpful.**

Chemical composition data is unfortunately not available within the auxiliary measurements performed within the presented study, except what is given within the discussion/interpretation of the TGA data in Section 3.4. However, we acknowledge the point made by the referee and believe that using more detailed chemical data (along with physical properties as for instance presented herein) will help to further our understanding of soot ice nucleation in further studies, as we suggest in the added statement (page 24 line 32 in the revised manuscript), given in the reply to point (1) of reviewer #03.

**(7) Page 23, line 15: May be add some examples of atmospheric processing of soot after long-range transport when soot particles become more compact (change contact angle) or coated with other materials.**

We agree that it is a good idea to add a brief description of possible processing mechanism.

Therefore, we changed the sentence from:

*"During this time the soot aggregates undergo atmospheric processing, for instance in the form of multiple cloud formation cycles."*

To (page 24 line 15, revised manuscript)

*"During this time the soot aggregates undergo atmospheric processing, encompassing any chemical and/or physical change of the particle properties (e.g. Zhang et al., 2008), for instance through photochemical processes (Li et al., 2018) or by acquiring of a coating due to condensation of semivolatile species or compaction of the soot agglomerate. Such processing can alter the physicochemical properties, such as fractal dimension or hygroscopicity (contact angle). Of particular interest here is the cloud processing of soot particles, i.e. the change in*

*physicochemical properties as the particles are involved in cloud microphysical processes such as cloud droplet or ice crystal formation."*

ALSTADT, V. J., DAWSON, J. N., LOSEY, D. J., SIHVONEN, S. K. & FREEDMAN, M. A. 2017. Heterogeneous Freezing of Carbon Nanotubes: A Model System for Pore Condensation and Freezing in the Atmosphere. *The Journal of Physical Chemistry A,* 121**,** 8166-8175.

DAVID, R. O., MARCOLLI, C., FAHRNI, J., QIU, Y., PEREZ SIRKIN, Y. A., MOLINERO, V., MÜNCH, S., MAHRT, F., BRÜHWILER, D., LOHMANN, U. & KANJI, Z. A. submitted. Is Deposition Ice Nucleation Real? The Role of Pore Condensation and Freezing on Atmospheric Ice Nucleation.

DUBININ, M. M. & STOECKLI, H. F. 1980. Homogeneous and heterogeneous micropore structures in carbonaceous adsorbents. *Journal of Colloid and Interface Science,* 75**,** 34-42.

LI, M., BAO, F., ZHANG, Y., SONG, W., CHEN, C. & ZHAO, J. 2018. Role of elemental carbon in the photochemical aging of soot. *Proceedings of the National Academy of Sciences*.

POPOVITCHEVA, O. B., PERSIANTSEVA, N. M., TRUKHIN, M. E., RULEV, G. B., SHONIJA, N. K., BURIKO, Y. Y., STARIK, A. M., DEMIRDJIAN, B., FERRY, D. & SUZANNE, J. 2000. Experimental characterization of aircraft combustor soot: Microstructure, surface area, porosity and water adsorption. *Physical Chemistry Chemical Physics,* 2**,** 4421-4426.

ZHANG, R., KHALIZOV, A. F., PAGELS, J., ZHANG, D., XUE, H. & MCMURRY, P. H. 2008. Variability in morphology, hygroscopicity, and optical properties of soot aerosols during atmospheric processing. *Proceedings of the National Academy of Sciences of the United States of America,* 105**,** 10291-10296.

---

## Author Comment (AC3) · 23 Aug 2018

Reviewer comments are reproduced in **bold** and our responses in normal typeface; extracts from the originally submitted manuscript are presented in *red italic*, and from the revised manuscript in *blue italic*.

We have numbered the reviewer comments for ease of cross-reference within the other reviews.
* * *
**(1) In this paper, authors measure the ice nucleation properties of various soot samples and further interpret these measurements by characterizing the particle properties and conclude that PCF kind of mechanism can explain the diverse INP properties observed for soot particles. My major concern is properties of realistic (emitted directly into the atmosphere through natural and anthropogenic processes) soot aerosol could be different than soot generated in the lab through dry-dispersion method, and this could affect the conclusions. Lab generated particles may not be atmospherically relevant particles. It is necessary to discuss how this connection can be made. An attempt is made in section 4, but not enough. The surface properties of soot samples from both sources are very different. Detailed discussion including these limitations needs to be explained and acknowledged. In addition, I have few following minor comments that I suggest to address before publication.**

We thank the reviewer for his/her comments. Further, we agree with the reviewer's major comment, that the particle types investigated herein are not necessarily identical to atmospheric soot particles. In Section 2 we describe the sample and their (implicit laboratory) origin in great detail and state that these soot types cover *"a wide range of combustion aerosol physicochemical properties as proxies of atmospheric soot"* (p.4, l.3 in old manuscript), i.e. are taken only as surrogates of atmospheric particles.

We revised this statement to more carefully acknowledge the difference to realistic soot particles to page 4 line 3 in revised manuscript:

*"Soot samples were chosen to represent a wide range of combustion aerosol physicochemical properties as proxies of atmospheric soot which could still differ from those particles studied here"*

Additionally, to address the reviewer's concern we clarify the differences between the laboratory generated soot particles and atmospheric soot particles more clearly, by adding the following statement (page 24 line 2 in revised manuscript):

*"This finding is important for instance for the fate of soot particles from aviation emissions, which are generally found to be even smaller than 100 nm in diameter (Moore et al., 2017, Yu et al., 2017). Such particles are often internally mixed with sulfuric acid (Kärcher, 2018) and can contain metallic compounds (Abegglen et al., 2016) or other residues such as lubrication oil (Yu et al., 2012) and organics (e.g.Yu et al., 2017). These factors can cause atmospheric soot particles to differ in physicochemical properties, e.g. contact angle (surface properties), from the particles types investigated here, which in turn influences their ice nucleation abilities. Still, there is increasing evidence [...]"*

We added a reference for the change in particle wettability as the soot particles undergo atmospheric processing, i.e. interact with atmospheric gases on page 24 line 19 of the revised manuscript.

We added the following statement at the end of Section 4 (page 24 line 32):

*"It is clear from this laboratory study that the physicochemical properties of soot aerosol determine their ice nucleation potential, with wettability being particularly important. At the same time our conclusions drawn here are limited by using bulk particle properties to explain ice nucleation taking place on individual particles (and at a molecular level). Further investigation of other factors, especially elaboration of a more quantitative pore size distribution and detailed chemical characterization of size selected particles, or at least individual particles, would be desirable to increase our understanding of soot ice nucleation abilities. Certainly, atmospheric soot particles can be more complex than the particle types investigated here. At the same time laboratory studies provide a more fundamental understanding of the properties relevant for ice nucleation"*

**(2) Page 2-3: Not all historical studies are discussed. It sounds like this is the first paper to study a majority of the soot samples (section 2.1 or Table 1). It may be best to discuss these studies briefly.**

We agree with the reviewer and have revised the manuscript introduction (page 2, line 21-26) to include and discuss more previous studies on soot ice nucleation, namely:

Häusler et al. (2018), Ullrich et al. (2017), Gorbunov et al. (1998), Garten and Head (1964).

We hope that the added references in the introduction section on p.2-3 provide a sufficient background to put our work in context.

**(3) Section 2.2: What is the size distribution of soot samples? What size was selected (page 6, line 9)? How was multiple charge correction applied?**

We clarified the size distribution of the soot samples in the revised manuscript. Please see our reply to point (2) of reviewer #02.

Details of the mobility sizes and corresponding flow ratios of aerosol to sheath flow, as well as an exemplary size distributions of the size selected aerosol, where multiple charge correction was applied, are given in the SI Section S6 (Table S1 and Fig. S15). Note aerosol to sheath flow ratios in the DMA and SMPS system, with the higher aerosol to sheath flow generally yielding a wider size distribution (broader transfer function).

We refer to the details of aerosol size selection a little bit further down in the main text on p.6, l.14 (initial manuscript):

*"[…] where they were size selected based on their dry electrical mobility diameter, $d_m$ (see Supplementary Information (SI) Table S1 for flow settings)."*

For clarification that the SI also contains information on the electrical mobility size selected for ice nucleation experiments, we revised this statement to (page 7 line 2 in revised manuscript:
*"[…] where they were size-selected based on their dry electrical mobility diameter, $d_m$ (see Supplementary Information (SI) Table S1 for details of mobility sizes selected and associated flow settings within the DMA)."*

We also added a cross-reference to the SI Section S6 in the manuscript on page 6 line 18 in the revised manuscript as well as to the caption of page 11, Fig. 2.

Our presented AF curves are not corrected for multiple-charged particles. We clarified this by adding the following statement to page 7 line 4 in the revised manuscript.

*"The ice nucleation experiments presented below include multiple-charged particles, i.e. particles that are larger than the electrical mobility size selected. The amount of double-charged particles is approximately 6 and 12 % for the 100 and 200 nm particles respectively and 15 % for the case of selecting mobility diameters 300 and 400 nm (Wiedensohler, 1988)."*

**(4) Page 8: For ice crystal detection 1 um threshold was used. How it is ensured that soot particles are not larger than 1 um. Figure 5 shows the images of these soot samples, and it looks particles are greater than 1 um. Therefore, ice particles will grow larger than 1 um. Please clarify. The implications are briefly explained on page 9 (line 5), but details regarding how 'aerosol-correct' is needed.**

The "aerosol correction" of the AF curves is applied to separate the signals from the particles from that corresponding to ice crystals. In Figure 1 below we illustrate the effect of this correction. Aerosol correction is applied when the size selected aerosol particles are detected in the OPC channel used for ice detection (in our case the 1 μm channel). This is the case in the example below, where an AF of approximately $8*10^{-2}$ is detected in the OPC for 70% < $RH_w$ < 85%. In order to remove this bias and more clearly show the real ice signal, we calculated the mean AF value over the RH range, where no ice is detected (in this case 70-84%) and subtracted this value from every data point of the entire AF curve. This correction makes the ice signal more accessible, as it reduces the false AF signal from aerosol particles increasing our detection ability of ice crystals at these low AFs.

For clarification, we added the following statement to p.9, l.24 in the revised manuscript:

*"In these cases the mean AF value detected by the OPC (channel) at low RH values, where no ice crystals are formed, was subtracted from each data point of the AF curve, correcting for any false signal in AF arising from large unactivated aerosol particles."*

Such an aerosol correction was only applied to the 300 nm and 400 nm samples of the LB_OEC and LB_RC samples, but not to the other soot samples. For the other soot samples no counts were detected in the 1 μm OPC channel prior to ice formation. This is why we write on p. 9, l. 5 (initial manuscript):

*"[…] to aerosol-correct some of the AF curves […]"*

The reason why aerosol correction was only necessary for the lamp black soots might be related to their wider primary particle size distribution (Fig. S17), but we have no direct evidence to support this. Nevertheless, this gives us confidence that the majority of the soot particles are (optically) smaller than 1 μm, which is further supported by our new Fig. S18, showing the distribution of spherical equivalent diameters, as derived from TEM analysis. See our reply to (2) of reviewer #02.

[Figure]

*Figure 1: Exemplary AF curves of the LB_RC soot sample for 400 nm soot particles, which have been aerosol corrected (black border) and which have not been aerosol corrected (yellow outline), to illustrate the effect of the aerosol correction applied to our AF curves. AF curves shown correspond to the mean over six independent ice nucleation experiments, evaluated using the 1 μm channel of the OPC.*

Finally, it is true that larger OPC channels could have been used for analysis, which is e.g. done by Lacher et al. (2017) when considering MPCs where different hydrometeor types are separated by optical particle size. However, this comes at the cost of detecting the ice nucleation at higher RH, as particles need to grow larger, especially for the cold temperatures investigated here.

**(5) Page 22, line 31-: If the soot properties of combustion aerosols are different than labgenerated, how it is possible that ice nucleation properties of 100 nm will be similar? The sentence saying '. . .it is unlikely that such small soot particles . . ..' is not true.**

We agree, and therefore have clarified this in the manuscript by stating that 100 nm particles with properties similar to the ones investigated here will not act as INP in the MPC regime. However, this could be different for combustion aerosols emitted in the atmosphere.

We have now modified the sentence to acknowledge that (see page 23 line 32 in revised manuscript):

*"Since none of the investigated soot types was ice nucleation active when particles of 100 nm mobility diameter were selected, it is unlikely that such small soot particles with properties similar to those investigated here will act as INP unless they are internally mixed with other ice active material."*

**(6) Section 4: Discussions such as comparison of residence time (timescales) within HINC and aircraft plume is irrelevant. Soot properties in both cases are not similar. Also, the assumption regarding cloud formation cycles is not supported.**

We agree that a comparison of timescales between aircraft plume and chamber residence time is irrelevant. We therefore deleted the following statement from the manuscript, p.23, l.10 (initial manuscript):

*"However, ice nucleation in aircraft plumes happens rapidly as contrail formation becomes visible within approximately one wingspan behind the air plane due to the high local supersaturations. This corresponds to timescales of a few seconds after particle emission (Schumann and Heymsfield, 2017; Schumann, 2012) similar to the residence time in HINC and the high saturation ratios achieved towards the end of our RH scans."*

We note that atmospheric soot particles can undergo multiple cloud formation cycles (e.g. Huang et al., 1994). Involvement of a soot particle to form a cloud droplet or an ice crystal resulting in altered physicochemical properties of the residual aerosol particle, is broadly referred to as "cloud processing". For instance, China et al. (2015) have recently shown that such processing can lead to a compaction of the soot aerosol, with implication for its optical properties. However, a change in particle morphology likely also alters the surface and hence pore distribution of the aggregate. From our conclusion of a PCF mechanism triggering ice nucleation, this would ultimately also affect/change the ice nucleation ability of the aerosol. It is thus our purpose to clearly state that the ice nucleation results as presented, are only valid for the first ice nucleation process (first cloud cycle) these particles are exposed to, but not necessarily any further ice nucleation cycles. At the same time once ice has been nucleated on a soot particle, the ice within the pores can be preserved even within environments that are subsaturated with respect to ice, due to the lower saturation vapor pressure over the concave surface within the cavity. Such "ice pockets" can then trigger detectable, macroscopic ice formation (growth) at lower RH, compared to the ice nucleation results presented herein.

We have clarified the statement regarding cloud formation page 24 line 21 in revised manuscript and reply to point (32) of reviewer #01.

ABEGGLEN, M., BREM, B. T., ELLENRIEDER, M., DURDINA, L., RINDLISBACHER, T., WANG, J., LOHMANN, U. & SIERAU, B. 2016. Chemical characterization of freshly emitted particulate matter from aircraft exhaust using single particle mass spectrometry. *Atmospheric Environment,* 134**,** 181-197.

CHINA, S., KULKARNI, G., SCARNATO, B., V. , SHARMA, N., PEKOUR, M., SHILLING, J., E. , WILSON, J., ZELENYUK, A., CHAND, D., LIU, S., C. AIKEN, A., DUBEY, M. K., LASKIN, A., ZAVERI, R., A. & MAZZOLENI, C. 2015. Morphology of diesel soot residuals from supercooled water droplets and ice crystals: implications for optical properties. *Environmental Research Letters,* 10**,** 114010.

GARTEN, V. A. & HEAD, R. B. 1964. Carbon particles and ice nucleation. *Nature,* 201**,** 1091-&.

GORBUNOV, B., BAKLANOV, A., KAKUTKINA, N., TOUMI, R. & WINDSOR, H. L. 1998. Ice nucleation on soot particles. *Journal of Aerosol Science,* 29**,** S1055-S1056.

HÄUSLER, T., GEBHARDT, P., IGLESIAS, D., RAMESHAN, C., MARCHESAN, S., EDER, D. & GROTHE, H. 2018. Ice Nucleation Activity of Graphene and Graphene Oxides. *Journal of Physical Chemistry C,* 122**,** 8182-8190.

HUANG, P. F., TURPIN, B. J., PIPHO, M. J., KITTELSON, D. B. & MCMURRY, P. H. 1994. EFFECTS OF WATER CONDENSATION AND EVAPORATION ON DIESEL CHAIN-AGGLOMERATE MORPHOLOGY. *Journal of Aerosol Science,* 25**,** 447-459.

KÄRCHER, B. 2018. Formation and radiative forcing of contrail cirrus. *Nature Communications,* 9**,** 1824.

LACHER, L., LOHMANN, U., BOOSE, Y., ZIPORI, A., HERRMANN, E., BUKOWIECKI, N., STEINBACHER, M. & KANJI, Z. A. 2017. The Horizontal Ice Nucleation Chamber (HINC): INP measurements at conditions relevant for mixed-phase clouds at the High Altitude Research Station Jungfraujoch. *Atmospheric Chemistry and Physics,* 17**,** 15199-15224.

MOORE, R. H., THORNHILL, K. L., WEINZIERL, B., SAUER, D., D'ASCOLI, E., KIM, J., LICHTENSTERN, M., SCHEIBE, M., BEATON, B., BEYERSDORF, A. J., BARRICK, J., BULZAN, D., CORR, C. A., CROSBIE, E., JURKAT, T., MARTIN, R., RIDDICK, D., SHOOK, M., SLOVER, G., VOIGT, C., WHITE, R., WINSTEAD, E., YASKY, R., ZIEMBA, L. D., BROWN, A., SCHLAGER, H. & ANDERSON, B. E. 2017. Biofuel blending reduces particle emissions from aircraft engines at cruise conditions. *Nature,* 543**,** 411-+.

ULLRICH, R., HOOSE, C., MÖHLER, O., NIEMAND, M., WAGNER, R., HÖHLER, K., HIRANUMA, N., SAATHOFF, H. & LEISNER, T. 2017. A New Ice Nucleation Active Site Parameterization for Desert Dust and Soot. *Journal of the Atmospheric Sciences,* 74**,** 699-717.

WIEDENSOHLER, A. 1988. An approximation of the bipolar charge-distribution for particles in the sub-micron size range. *Journal of Aerosol Science,* 19**,** 387-389.

YU, Z., HERNDON, S. C., ZIEMBA, L. D., TIMKO, M. T., LISCINSKY, D. S., ANDERSON, B. E. & MIAKE-LYE, R. C. 2012. Identification of Lubrication Oil in the Particulate Matter Emissions from Engine Exhaust of In-Service Commercial Aircraft. *Environmental Science & Technology,* 46**,** 9630-9637.

YU, Z. H., LISCINSKY, D. S., FORTNER, E. C., YACOVITCH, T. I., CROTEAU, P., HERNDON, S. C. & MIAKE-LYE, R. C. 2017. Evaluation of PM emissions from two in-service gas turbine general aviation aircraft engines. *Atmospheric Environment,* 160**,** 9-18.